# MMMG: A Massive, Multidisciplinary, Multi-Tier Generation Benchmark for Text-to-Image Reasoning

**Yuxuan Luo**[1][†], **Yuhui Yuan**[4][‡], **Junwen Chen**[2][†], **Haonan Cai**[1], **Ziyi Yue**[1],
**Yuwei Yang**[3][†], **Fatima Zohra Daha**[5], **Ji Li**[5], **Zhouhui Lian**[1][‡]

[1]Wangxuan Institute of Computer Technology, Peking University, China,
[2]The University of Electro-Communications, [3]Australian National University
[4]Microsoft Research Asia, [5]Microsoft

https://mmmgbench.github.io/

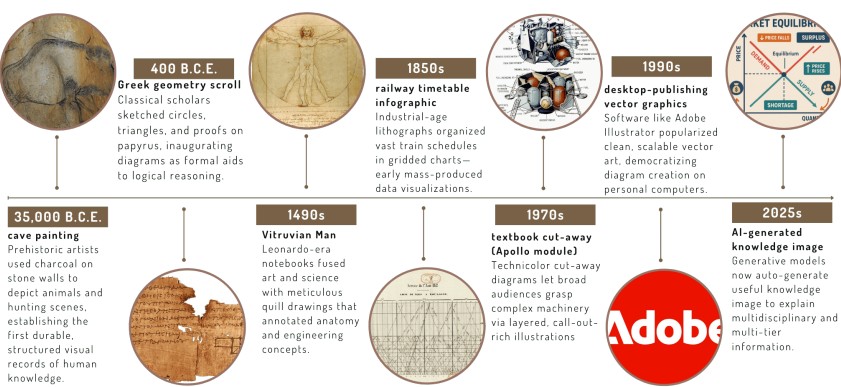

Figure 1: $40,000$ Years of Knowledge Image: From Cave Paintings to Generative AI.

## Abstract

In this paper, we introduce knowledge image generation as a new task, alongside the Massive Multi-Discipline Multi-Tier Knowledge-Image Generation Benchmark (MMMG) to probe the reasoning capability of image generation models. Knowledge images have been central to human civilization and to the mechanisms of human learning—a fact underscored by *dual-coding theory* and the *picture-superiority effect*[2]. Generating such images is challenging, demanding multimodal reasoning that fuses world knowledge with pixel-level grounding into clear explanatory visuals. To enable comprehensive evaluation, MMMG offers $4,456$ expert-validated (knowledge) image-prompt pairs spanning $10$ disciplines, $6$ educational levels, and diverse knowledge formats such as charts, diagrams, and mind maps. To eliminate confounding complexity during evaluation, we adopt a unified Knowledge Graph (KG) representation. Each KG explicitly delineates a target image's core entities and their dependencies. We further introduce MMMG-Score to evaluate generated knowledge images. This metric combines factual fidelity, measured by graph-edit distance between KGs, with visual clarity assessment. Comprehensive evaluations of $21$ state-of-the-art text-to-image generation models expose serious reasoning deficits—low entity fidelity, weak relations, and clutter—with GPT-4o achieving an MMMG-Score of only $50.20$, underscoring the benchmark's difficulty. To spur further progress, we release FLUX-Reason (MMMG-Score of $34.45$), an effective

---

[†]Research Intern at Microsoft. [‡]: ✉ yuhui.yuan@microsoft.com ✉ lianzhouhui@pku.edu.cn

[2]Both dual-coding theory and the picture-superiority effect principles suggest humans remember visuals more effectively than words, partly because visual information can engage multiple cognitive encoding pathways. This work was supported by the National Natural Science Foundation of China (Grant No.: 62372015), Key Laboratory of Intelligent Press Media Technology, and State Key Laboratory of General Artificial Intelligence.

39th Conference on Neural Information Processing Systems (NeurIPS 2025) Track on Datasets and Benchmarks.



Figure 2: Representative knowledge images generated with GPT-4o. Disciplines (L-R): Economics, Oceanography, Environmental Engineering, Astrophysics, Climate Science. Details are in the supplementary.

and open baseline that combines a reasoning LLM with diffusion models and is trained on 16,000 curated knowledge image–prompt pairs.

# 1 Introduction

Reasoning-based large language models (LLMs) such as OpenAI-o1/o3 [33, 36, 35] and DeepSeek-R1 [12] excel on math and coding tests (AIME 2024 [23], Codeforces [39], GPQA Diamond [43], MATH-500 [15], MMLU [14], SWE-bench [25]) thanks to rigorous, reasoning-focused benchmarks. By contrast, widely used text-to-image benchmarks [21, 7] still focus on instruction following and compositionality—e.g., attribute binding—while largely overlooking reasoning. The lack of reasoning-oriented benchmarks has left text-to-image generation models lagging significantly behind reasoning-focused LLMs.

Measuring reasoning in image generation is a non-trivial task. Current evaluations emphasize prompt-following, aesthetics, and visual–text rendering, typically quantified by CLIP [44], FID [45], OCR, and Aesthetic [1] scores. Yet producing such images seldom requires complex logical, domain-specific reasoning. Motivated by how humans leverage visuals to think, we introduce a new task—knowledge image generation: given only a vague user prompt, the model must autonomously infer the pertinent concepts (or entities) and relationships, and render them in a coherent knowledge image—such as a diagram, chart, infographic, or other visual—that faithfully conveys the intended information.

Throughout human history, knowledge images have propelled progress for nearly 40,000 years, serving as a lasting bridge that turns abstract ideas into concrete, shareable visual forms (Figure 1). Cognitive science also supports this direction—the dual-coding theory [53] and the picture-superiority effect [54] suggest humans encode information more robustly when language and imagery are combined. Creating such visuals, however, is intrinsically difficult: a model must fuse broad world knowledge with spatial composition, select salient entities, and faithfully ground relations in pixel space. Figure 2 visualizes several knowledge representations across disciplines.

To advance text-to-image reasoning, we introduce the Massive Multi-Discipline Multi-Tier Knowledge-Image Generation Benchmark (`MMMG`). `MMMG` comprises 4,456 expert-validated prompt–image pairs spanning ten academic disciplines—Biology (850), Chemistry (328), Mathematics (399), Engineering (582), Geography (352), Economics (623), Sociology (479), Philosophy (210), History (327), and Literature (306);—and six educational tiers: pre-school (591), primary school (680), secondary school (693), high school (936), undergraduate (744), and PhD (812). Each sample is annotated with a high-quality knowledge graph that lists the necessary entities and their dependencies, enabling format-agnostic coverage and requiring models to generalize across domains and reasoning levels. Its benefits are twofold: first, it abstracts core concepts into an interpretable graph, reducing the diversity and complexity of knowledge visuals; second, it enables objective fidelity evaluation via graph-edit distance between the ground-truth and generated graphs.

Benchmarking reasoning fidelity in generated images requires more than perceptual metrics. We therefore introduce `MMMG-Score`, which combines the graph-edit distance between knowledge graphs with a visual-clarity score derived from foundational segmentation models. Specifically, we employ the OpenAI-o3 reasoning LLM to analyze each image–prompt pair, and predict an initial knowledge graph. For the visual-clarity component, we run Segment Anything Model v2 (SAM-2) [42] on the generated images and penalize overly cluttered and disorganized outputs that may "hack" the reasoning LLM to extract the unreliable knowledge graphs yet fail to convey the knowledge clearly. The importance of this visual-clarity metric is examined in the experimental section.

We conduct comprehensive evaluations of 21 state-of-the-art text-to-image models—LlamaGen, JanusFlow, Emu-3, SimpleAR, Janus-Pro, CogView, SEED-X, SDXL-1.0, SDXL-1.0-refiner, Infinity, FLUX.1-[dev], FLUX.1-[pro], Ideogram 2.0, HiDream-l1-Full, Qwen-Image, BAGEL, Nano Banana,

| Benchmark | Scale | Focus | Domains | Metrics | World Knowledge | Explanatory |
|---|---|---|---|---|---|---|
| GenEval [11] | 553 | Compositionality | counting, colors, position, attribute binding | Accuracy | ✗ | ✗ |
| T2I-CompBench++ [20] | 6,000 | Compositionality | Object-Attribute Binding | BLIP-VQA, UniDet, CLIP | ✗ | ✗ |
| DPG-Bench [18] | 1,065 | Prompt Adherence | Dense Scene Generation | CLIP, Human Eval | ✗ | ✗ |
| Commonsense-T2I [7] | 1,000+ | Commonsense Reasoning | Everyday Scenarios | Accuracy | ✗ | ✗ |
| Winoground-T2I [58] | 11,000 | Compositionality | 20 Types | Contrastive Accuracy | ✗ | ✗ |
| TIFA [19] | 1,000 | Faithfulness | General Knowledge | VQA-based | ✗ | ✗ |
| TypeScore [46] | 1,000 | Text Fidelity | Scene Text | OCR-based | ✗ | ✗ |
| GenAI-Bench [28] | 1,600 | Compositionality | Scenes, objects, attributes, relations, counting, comparison, etc. | VQAScore | ✗ | ✗ |
| CUBE [26] | 1,000 | Cultural Competence | Cuisine, Landmarks, Art spanning 8 countries | Cultural Awareness, Vendi Scores | ✓ | ✗ |
| WISE [32] | 1,000 | Commonsense Reasoning | Science, Culture, Space-Time | LLM-Judged | ✓ | ✗ |
| MMMG (Ours) | 4,456 | Disciplinary Knowledge | 10+ Academic Fields | Readability, Graph Edit Distance | ✓ | ✓ |

Table 1: Comparison with previous Text-to-Image (T2I) benchmarks.

and GPT-4o image generation—on the MMMG benchmark, reporting their FID, aesthetic, WISE [32], and MMMG-Score metrics. We also conduct human studies to confirm that MMMG-Score aligns best with user judgements, underscoring the value of knowledge-graph–based evaluation. Our MMMG benchmark presents significant challenges: even the GPT-4o image generation achieves only MMMG-Score of 46.66, while the next-best model, the open-source HiDream-I1-Full, reaches just MMMG-Score of 25.72. To catalyze further research, we release FLUX-Reason, a fully reproducible and open-source baseline that pairs a reasoning-oriented LLM (e.g., OpenAI-o3 or DeepSeek-R1) with a diffusion model (FLUX.1-[dev]) trained on 16,000 curated knowledge-image pairs. Although its MMMG-Score of 30.52 still trails that of GPT-4o, FLUX-Reason serves as an open source baseline and underscores the new challenges posed by the MMMG benchmark for next-generation reasoning-oriented text-to-image generation models.

## 2 Related Work

**Benchmarks for Text-to-Image Generation.** Many benchmarks have been developed to assess both the limitations and progress of recent text-to-image models. We summarize the comparison between MMMG and prior benchmarks in Table 1. GenEval [11] introduces object detectors for fine-grained, object-level evaluation, addressing the shortcomings of holistic metrics. T2I-CompBench++ [20] increases compositional difficulty via prompts involving attributes, relationships, numeracy, and complex scenes. Commonsense-T2I [7] uses adversarial prompts to probe visual commonsense reasoning. Winoground-T2I [58] evaluates compositional generalization with contrastive sentence pairs. DPG-Bench [18] targets instruction-following with longer, text-rich prompts. The concurrent WISE benchmark [32] is most related, focusing on world knowledge-based evaluation across cultural, scientific, and temporal domains. However, WISE emphasizes photorealism with implicit knowledge, while MMMG requires models to explicitly visualize structured world knowledge in a semantically grounded and explanatory manner.

**Reasoning in Text-to-Image Generation.** While LLMs have achieved significant progress in reasoning through techniques such as chain-of-thought prompting [52, 27] and large-scale reinforcement learning [12, 33], recent models excel in benchmarks focused on mathematics, coding, and tool use (e.g., MMLU [14], AGIEval [57], LogicBench [38], MathVista [30]). Inspired by this progress, several works have explored injecting reasoning into image generation, including ImageGen-CoT [8, 29], HiDream [16], T2I-R1 [24], Meta-Queries [37], and MINT [51]. However, these methods are typically evaluated on prior benchmarks and rely on caption-based metrics (e.g., CLIPScore [40]) or subjective human preference, both of which lack fidelity in assessing reasoning ability. To address this gap, MMMG introduces a knowledge image generation task requiring advanced multimodal reasoning, along with MMMG-Score, a structured metric that compares extracted and ground-truth knowledge graphs. We further propose FLUX-Reason, a reasoning-enhanced model, and evaluate it on the MMMG benchmark.

## 3 Method

We first illustrate the definition of the knowledge image generation task by leveraging additional knowledge graph annotations. Next, we describe how we build the MMMG benchmark, and provide an overview of key dataset statistics. We then introduce the novel MMMG-Score, a metric we propose for more reliable evaluation. Last, we present details of our strong baseline, FLUX-Reason, which explicitly combines a reasoning LLM with a text-to-image generation model in a cascaded manner.

### 3.1 Knowledge Image Generation

**Formulation.** The knowledge image generation task begins with a concise, question-like prompt $\mathbf{X}$ and employs a generative model $f$ to produce a knowledge image $\mathbf{Y}$ conditioned on $\mathbf{X}$. A knowledge image typically comprises multiple entities and their interrelationships. To capture this structure, we extract an auxiliary knowledge graph $\mathbf{G} = (\mathcal{E}, \mathcal{D})$ using a reasoning LLM (e.g., OpenAI-o3), which takes both the text prompt and the target image (if available) as input. Here, $\mathcal{E} = \{\mathbf{e}_1, \ldots, \mathbf{e}_n\}$

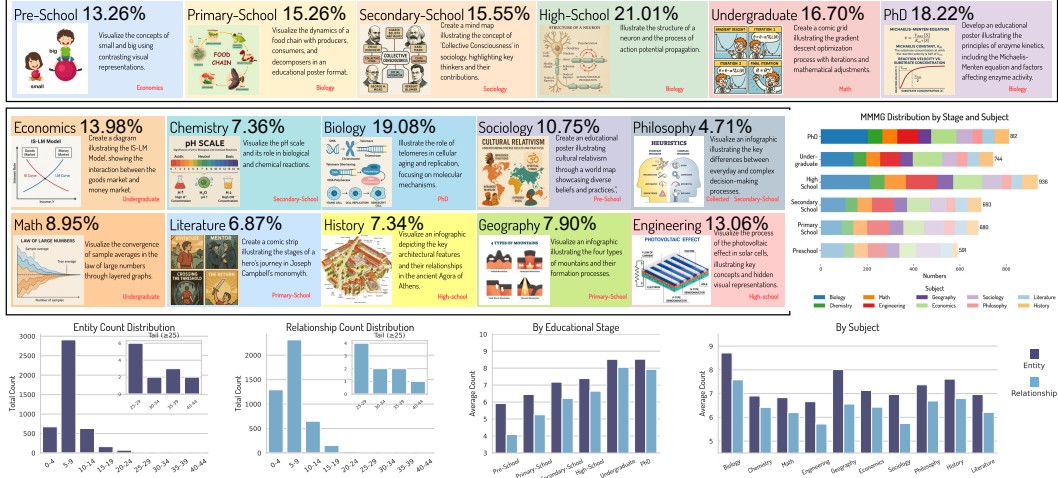

Figure 3: **MMMG Dataset Statistics:** The top panel shows MMMG test dataset distribution across educational levels and disciplines. The bottom panel presents statistics of both train and test sets, while bottom right depicts knowledge graph complexity increase across educational stages and differ among subjects.

denotes the set of graph nodes representing entities, and $\mathcal{D} = \{d_i(\mathbf{e}_j, \mathbf{e}_k)\}_{i=1}^{K}$ denotes the set of edges encoding relationships between entities.

**Importance of Knowledge Graph.** The knowledge graph $\mathbf{G}$ is essential for evaluating whether the generated image faithfully visualizes the domain knowledge implied by the prompt $\mathbf{X}$. Since $\mathbf{X}$ is typically a concise question rather than a descriptive instruction—e.g., "*Illustrate the structure of a neuron and the process of action potential propagation*"—it is not well-aligned with the visual content, making CLIP-based verification unreliable.

**Knowledge Graph Extraction.** Accurate knowledge graph extraction requires inferring world knowledge from the text prompt $\mathbf{X}$ and identifying the corresponding visual entities and relations in the image $\mathbf{Y}$. We follow a two-step process: OpenAI-o3 processes each <$\mathbf{X}$, $\mathbf{Y}$> and extracts knowledge graphs following a defined schema (Supplemental C.3); four expert annotators manually verify the results and filter out low-quality cases.

**Relationship Formalization.** To ensure that the knowledge graph can represent diverse knowledge across six educational levels and ten disciplines, we propose a domain-agnostic set of predicates for relationships: $\text{Defines}(\cdot, \cdot)$, $\text{Entails}(\cdot, \cdot)$, $\text{Causes}(\cdot, \cdot)$, $\text{Contains}(\cdot, \cdot)$, $\text{Requires}(\cdot, \cdot)$, and $\text{TemporalOrder}(\cdot, \cdot)$, with optional dynamic modifiers such as $\text{change}(\cdot)$ to represent trends or shifts. For instance, $\text{Causes}(\text{increase}(e_1), \text{decrease}(e_2))$ may represent a graph edge where an increasing population (denoted as $e_1$) leads to reduced biodiversity (denoted as $e_2$).

In the neuron example, we can extract a non-trivial knowledge graph consisting of 9 entities, $\mathcal{E} = \{$dendrites, cell body, nucleus, axon, myelin sheath, schwann cell, node of ranvier, action potential propagation, depolarization$\}$, and 8 relationships, $\mathcal{D} = \{\text{Contains}(\text{dendrites}, \text{cell body}), \text{Contains}(\text{cell body}, \text{nucleus}), \text{Contains}(\text{cell body}, \text{axon}), \text{Contains}(\text{axon}, \text{myelin sheath}), \text{Contains}(\text{myelin sheath}, \text{schwann cell}), \text{Contains}(\text{axon}, \text{node of Ranvier}), \text{Causes}(\text{depolarization}, \text{action potential propagation}), \text{Requires}(\text{action potential propagation}, \text{axon})\}$. This abstraction provides two key benefits: (i) structural consistency across disciplines and educational levels, and (ii) automated evaluation using normalized Graph Edit Distance (GED) to assess factual alignment with reference graphs.

### 3.2 MMMG Benchmark: Statistics and Curation Process

**Statistics.** Figure 3 provides an overview of the dataset statistics for MMMG, which spans six educational stages and ten academic disciplines. MMMG contains 4,456 expert-collected prompt–image pairs. We analyze the statistics as follows:

• At the top of Figure 3, we present the distribution across six different educational levels and illustrate representative examples to demonstrate how the inherent challenges increase from pre-school to PhD-level knowledge images. We ensured a balanced distribution across educational levels.

• In the middle of Figure 3, we present the distribution across ten academic disciplines. We find that biology, economics, and engineering are the dominant domains that rely more on knowledge images

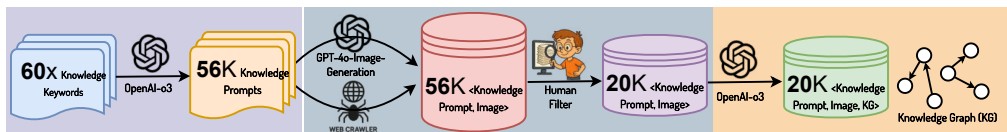

Figure 4: `MMMG` **Dataset Construction Pipeline**: From knowledge keywords across six educational levels and ten disciplines, we generate 56K prompts using OpenAI-o3. These prompts are clustered into semantic groups and then sent to either GPT-4o for image generation or to a web crawler for image retrieval. The resulting 56K knowledge images are filtered down to 20K through a cascade of automated steps and human filter. Last, we use OpenAI-o3 to extract a knowledge graph for each prompt–image pair.

than others—especially philosophy, which accounts for only $4.71\%$ of the dataset. On the right side of the middle, we visualize the distribution of all 10 disciplines across the 6 educational levels.

• At the bottom of Figure 3, we highlight the complexity of knowledge image generation by showing statistics on the number of entities and relationships in the dataset. We find that nearly $3,000$ samples require generating $5 \sim 10$ entities and $5 \sim 10$ relationships. On the right side of the third row, we report the distributions of these statistics across different educational levels and disciplines.

**Curation Pipeline.** Figure 4 illustrates the overall pipeline for curating the `MMMG` dataset. Starting from $60\times$ knowledge keywords, we employ a cascade dataset flywheel comprising several stages: OpenAI-o3 for prompt generation, GPT-4o-Image for synthesizing training images, a web crawler for collecting real-world data, and human expert filtering for quality assurance. Together, these stages yield around $20,000$ curated candidates—$16,000$ high-aesthetic synthetic samples and $3,452$ knowledge-accurate, real-world visuals—from which the `MMMG` benchmark is constructed for comprehensive evaluation.

• Knowledge Keywords ❶ → Knowledge Prompts ❷: In the left of Figure 4, we first apply OpenAI-o3 to generate approximately 56,830 knowledge text prompts by combining two keywords: one specifying the educational level and one specifying the discipline. The educational level keyword is sampled from a seed set of six candidates: [pre-school, primary school, secondary school, high school, undergraduate, PhD], while the disciplinary keyword is sampled from ten candidates: [economics, chemistry, biology, sociology, philosophy, math, literature, history, geography, engineering].

• Knowledge Prompts ❷ → Knowledge Images ❸: We source knowledge images from two complementary domains: web-crawled, factually grounded visuals for benchmark construction and GPT-4o-generated, scalable images for training. To prevent concept overlap, we cluster 56,830 knowledge prompts into 11,732 semantic groups using SentenceTransformer embeddings and DBSCAN. Prompts from larger clusters (common concepts) are used for synthetic data (30K images), while smaller clusters are used for web crawling (26K images). This allocation leverages the generator's advantage on familiar concepts while enhancing the benchmark's conceptual diversity.

• Knowledge Images Filter ❸ → ❹: We apply deduplication[3] and OCR-based filtering to remove duplicates and samples lacking explanatory visual text. During GPT-4o image generation, cropping artifacts often harm visual completeness. To address this, we use OpenAI-o3 to detect severe cropping. Human experts further verify text–image alignment and discard samples with factual errors or visual artifacts. After filtering, we obtain 20K curated knowledge image–text pairs, including 16K for training and 4K for benchmarking.

• Knowledge Prompts, Images ❹ → Knowledge Graphs ❺: We use OpenAI-o3 to generate a structured knowledge graph for each of the 20K prompt–image pairs, following the format described earlier. We also use DeepSeek-R1 to produce step-by-step reasoning over the graph's entities and relations. The 20K samples are split by their topics and concepts to ensure minimal overlap. Human experts then verify all samples and select the most accurate ones, resulting in the `MMMG` benchmark of 4,456 high-quality pairs.

### 3.3 MMMG-Score: Measuring Knowledge Fidelity and Visual Readability

As perceptual metrics like FID or CLIP are insufficient for reasoning evaluation, we propose `MMMG-Score`, a novel metric combining knowledge fidelity and visual readability.

---

[3] https://github.com/idealo/imagededup

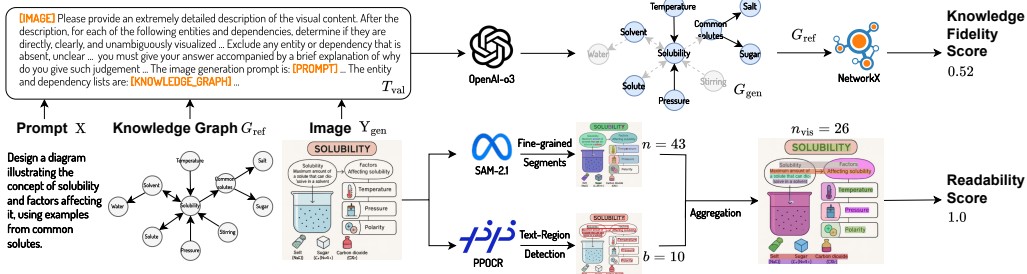

Figure 5: Illustration of `MMMG-Score` computation: We compute the knowledge fidelity score using graph edit distance, and the readability score by counting the number of segments in the generated knowledge image.

**Knowledge Fidelity via Grounded Knowledge Graph Extraction.** Given a generated image $\mathbf{Y}_{gen}$, its knowledge prompt $\mathbf{X}$, and the reference knowledge graph $\mathbf{G}_{ref}$, we employ OpenAI-o3 (high reasoning effort) to ground $\mathbf{G}_{ref}$ onto the pixel space of $\mathbf{Y}_{gen}$. This yields a grounded subgraph $\mathbf{G}_{gen} \subseteq \mathbf{G}_{ref}$, representing the knowledge actually depicted in the image. To ensure fair evaluation of each entity $\mathbf{e}_i$ and relation $d_i(\mathbf{e}j, \mathbf{e}k)$, OpenAI-o3 performs visual reasoning and outputs a detailed chain-of-thought before the final judgement. As illustrated in Figure 5, the knowledge fidelity score is computed as $1 - \text{GED}(\mathbf{G}_{gen}, \mathbf{G}_{ref})$ using NetworkX[4], where smaller graph edit distances indicate higher fidelity.

**Visual Readability via Foundation Segmentation Model.** Readability is critical for knowledge-image generation, ensuring effective information delivery. To assess it, we compute a Readability Score using a segmentation model and a text detector, rewarding coherent regions and penalizing excessive fragmentation. As shown in Figure 5 (bottom right), we use SAM-2.1 [41] for segmentation (seeded with $32 \times 32$ uniform points, NMS threshold $0.6$) and PaddleOCR[5] for text detection. Overlapping masks and text boxes are merged, and the final region count defines the Readability Score:

$$R(n_{vis}) = \begin{cases} 1, & n_{vis} \leq n_{min}, \\ \dfrac{n_{max} - n_{vis}}{n_{max} - n_{min}}, & n_{min} < n_{vis} < n_{max}, \\ 0, & n_{vis} \geq n_{max}. \end{cases} \tag{1}$$

Empirically, we set $n_{min} = 70$ and $n_{max} = 160$ based on segment distributions observed across common text-to-image models (see Appendix D.2). This range excludes overly fragmented or unreadable outputs that are (i) unsuitable for evaluating meaningful reasoning and (ii) prone to exploitation by models that over-generate dense but illegible content. This thresholding approach thus enforces essential legibility for reliable knowledge evaluation.

**MMMG-Score via a Multiplicative Design.** The final `MMMG-Score` is computed by multiplying the above knowledge fidelity and readability score:

$$\texttt{MMMG-Score}(\mathbf{Y}_{gen}) = R(n_{vis}) \cdot [1 - \text{GED}(\mathbf{G}_{gen}, \mathbf{G}_{ref})] \in [0, 1]. \tag{2}$$

This formula enforces the essential "AND" requirement for an effective knowledge image. Poor performance in either fidelity or readability will result in a proportionally low overall score.

Empirical results show that this composite metric correlates more strongly with human ratings than general perceptual scores such as FID, CLIP-Score, or aesthetic measures. Error analysis further reveals that visual clutter and unreadability mainly occur in weaker baselines, reflecting deficiencies in visual reasoning and layout planning. Among alternative designs, the multiplicative formulation remains the most concise and robust for quantifying knowledge-image quality.

### 3.4 FLUX-Reason

We design FLUX-Reason to enhance reasoning capabilities in knowledge image generation by explicitly integrating a reasoning LLM with a diffusion-based generator. As illustrated in Figure 6, it

---

[4] https://github.com/networkx
[5] https://github.com/PaddlePaddle/PaddleOCR

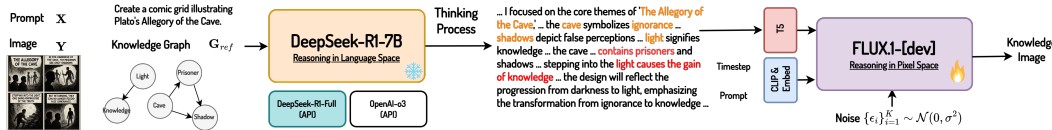

Figure 6: Overview of the FLUX-Reason pipeline. Reasoning LLMs first generate chain-of-thought (CoT) trajectories and visual planning cues conditioned on structured knowledge graphs. In the reasoning trace, entities and relations are highlighted in orange and red, respectively. These traces are then encoded into the diffusion model to guide visual planning in pixel space.

consists of three variants: `FLUX-Reason (R1-7B)` incorporates `DeepSeek-R1-Distill-Qwen-7B` for local inference; `FLUX-Reason (R1)` queries the `DeepSeek-R1-Full` API; `FLUX-Reason (o3)` utilizes `OpenAI-o3` to produce summarized reasoning chains.

To supervise training, we extract chain-of-thought (CoT) reasoning traces from 16K GPT-4o-generated samples, each annotated with a ⟨prompt, image, knowledge graph (KG)⟩ triplet. Conditioned on explicit entities and relations, the extracted reasoning traces provide fine-grained guidance for concept selection, interaction modeling, and spatial arrangement.

To incorporate such long-form structured reasoning into generation, we extend the T5 encoder's input length to 2048 tokens to accommodate the textual reasoning input, which is then transformed into pixel-space representations. The diffusion model is fine-tuned with LoRA over 10K steps, enabling it to learn pixel-level planning aligned with the structured reasoning trajectory.

## 4    Experiments

### 4.1    Main Results

We benchmark 21 state-of-the-art T2I models and 3 FLUX-Reason variants, spanning three paradigms: **autoregressive (AR)** models, including `JanusFlow-1.3B` [31], `Janus-pro-7B` [5], `LlamaGen` [48], `SimpleAR` [49], and `Infinity` [13]; **diffusion-based (DM)** models, including `SDXL-1.0`, `SDXL-1.0-refiner` [47], `Ideogram` [22], `CogView-4` [56], `HiDream-I1-Full` [17], `Qwen-Image` [55], `FLUX.1-[dev]` [2], re-captioned `FLUX.1-[dev]`, `FLUX.1-[pro]` [3]; and **multimodal (MM)** models, including `Emu-3` [50], `BLIP3-o` [4], `Seed-X` [8], `BAGEL` [6], `Gemini 2.0 Flash` [9], `Gemini Nano Banana` [10] and `GPT-4o` [34]. All models are evaluated with a fixed seed of 42. DM models use a classifier-free guidance (CFG) scale of 3.5, while AR and MM models apply default decoding settings. Open-source models are experimented on with 6×A40 ($512^2$) or 4×A100 ($1024^2$), depending on the image resolution. API-based models are queried directly.

**Performance Variation with Educational Level.**    Table 2 shows a clear performance drop as task complexity increases with education level. Most models, regardless of architecture, perform reasonably at pre-school (e.g., 20–30), but fall to low scores (below 10) at the PhD level, exposing their limitations in abstract reasoning and compositional planning. `GPT-4o` and `Nano Banana` stand out with strong, stable performance, showing robust generalization even on underspecified prompts.

**Model Highlights.**    `HiDream-I1-Full` shows competitive scores (28.04) despite being open-source, likely benefiting from structured priors in its Llama-based encoder. Similarly, `FLUX.1-[pro]` (27.14) and `SEED-X` (18.16) outperform many AR and MM models, indicating advantages of diffusion planning. `BAGEL` underperforms due to its overly brief "thinking" trajectories, hindering the delineation of detailed entities and intricate relationships. `Qwen-Image` performs well at lower educational levels but drops sharply in higher ones, likely due to its distilled data and limited conceptual understanding.

**Reasoning vs. Recaptioning.**    To assess the impact of reasoning, we compare `FLUX.1-[dev]`, a recaptioned variant using `OpenAI-o3` (512-token prompts), and our reasoning-guided `FLUX-Reason (R1)`. While recaptioning reduces performance across levels, reasoning traces yield substantial gains—particularly at higher tiers. `FLUX-Reason (R1)` reaches an average score of 34.45, confirming that structured reasoning, not verbosity, is crucial for knowledge-grounded image generation.

**Discipline-Level Observations.**    Figure 7 reveals domain-specific reasoning challenges. Weaker models (e.g., `LlamaGen`, `Emu-3`) perform best in Geography and Literature, where visuals are descriptive and align better with pretraining data. In contrast, Economics, History, and Sociology

Table 2: `MMMG-Score` (×100) across prevalent image generation models, covering Diffusion, AR-based, and Multimodal architectures, evaluated over six educational stages.

| Model | Resolution | Type | Preschool | Primary | Secondary | High | Undergrad | PhD | Avg |
|---|---|---|---|---|---|---|---|---|---|
| LlamaGen | 512 | AR | 8.24 | 3.77 | 2.44 | 1.44 | 1.08 | 1.14 | 3.02 |
| Emu-3 | 720 | MM | 12.44 | 7.12 | 6.41 | 5.28 | 2.65 | 2.74 | 6.11 |
| JanusFlow-1.3B | 384 | AR | 24.11 | 12.72 | 8.81 | 5.56 | 3.57 | 3.82 | 9.77 |
| SimpleAR | 1024 | AR | 23.12 | 11.97 | 8.96 | 6.44 | 4.36 | 3.99 | 9.81 |
| Ideogram | 1024 | DM | 20.39 | 14.14 | 12.90 | 9.68 | 8.41 | 7.73 | 12.21 |
| BLIP3-o | 1024 | MM | 29.59 | 17.43 | 11.52 | 8.32 | 5.75 | 5.21 | 12.97 |
| Janus-pro-7B | 384 | AR | 29.50 | 16.72 | 12.73 | 8.45 | 5.57 | 5.66 | 13.10 |
| CogView-4 | 1024 | DM | 24.61 | 16.02 | 13.91 | 10.02 | 7.30 | 6.73 | 13.10 |
| BAGEL | 1024 | MM | 29.29 | 19.42 | 15.29 | 11.11 | 7.40 | 7.60 | 15.02 |
| SDXL-1.0 | 1024 | DM | 23.41 | 19.12 | 17.41 | 16.26 | 9.92 | 9.29 | 15.90 |
| SDXL-1.0-refiner | 1024 | DM | 24.55 | 19.24 | 18.59 | 16.72 | 9.68 | 8.94 | 16.29 |
| FLUX.1-[dev] (recaption) | 1024 | DM | 28.05 | 20.29 | 20.70 | 15.74 | 12.59 | 11.20 | 18.10 |
| SEED-X | 1024 | MM | 33.41 | 22.67 | 19.49 | 15.74 | 8.88 | 8.76 | 18.16 |
| FLUX.1-[dev] | 1024 | DM | 29.80 | 23.09 | 20.99 | 16.12 | 12.47 | 12.30 | 19.13 |
| Infinity | 1024 | AR | 25.87 | 20.63 | 21.86 | 18.36 | 14.23 | 14.14 | 19.18 |
| Qwen-Image | 1024 | DM | 37.23 | 25.46 | 25.54 | 18.28 | 15.11 | 14.20 | 22.64 |
| FLUX.1-[pro] | 1024 | DM | 42.27 | 30.10 | 29.15 | 23.40 | 19.32 | 18.61 | 27.14 |
| HiDream-I1-Full | 1024 | DM | 42.86 | 31.77 | 30.26 | 23.39 | 19.88 | 20.05 | 28.04 |
| Gemini 2.0 Flash | 1024 | MM | 41.98 | 32.06 | 31.69 | 29.99 | 20.58 | 19.53 | 29.31 |
| Gemini Nano Banana | 1024 | MM | 49.46 | 44.58 | 51.17 | 48.85 | 41.27 | 39.07 | 45.73 |
| GPT-4o | 1024 | MM | **64.78** | **51.94** | **53.04** | **51.29** | **41.52** | **38.60** | **50.20** |
| FLUX-Reason (o3) | 1024 | DM | 37.83 | 29.72 | 29.50 | 23.62 | 20.29 | 18.73 | 26.62 |
| FLUX-Reason (R1-7B) | 1024 | DM | 44.93 | 34.41 | 34.19 | 28.70 | 23.36 | 21.99 | 31.26 |
| FLUX-Reason (R1) | 1024 | DM | 49.10 | 39.39 | 37.00 | 33.65 | 24.96 | 22.57 | 34.45 |

remain difficult even for stronger models due to their reliance on charts, temporal events, and abstract social concepts—structures rarely seen during pretraining.

A notable divergence appears between Chemistry and Biology: while both start similarly, Chemistry performs better with stronger models, likely due to its standardized diagram formats and symbolic representations. In contrast, Biology's visuals are often more irregular and spatially complex, making them less amenable to straightforward interpretation by such models.

Mathematics and Engineering perform well despite textual abstraction, suggesting structured visuals (e.g., geometry, schematics) are more model-friendly than symbolic reasoning. Since 77% of `MMMG` images are human-designed, these trends also reflect real-world preferences. Overall, `MMMG` exposes domain gaps and visual-semantic reasoning challenges overlooked by text-only benchmarks.

## 4.2 Human Alignment and Metric Comparison

To assess alignment with human perception, we collected over 1,200 expert ratings (0–10 on clarity, correctness, accuracy and faithfulness) across six educational levels. We compared four metrics—`MMMG-Score`; an LLM-as-a-judge WIScore [32] with OpenAI-o3 evaluator; FID computed over 3,452 ground-truth images; and AES-2.5 [1]. Figure 8 reports their Pearson correlations against human scores: `MMMG-Score` leads with $r = 0.876$; WISE achieves only $r = 0.701$, even trailing FID's negative correlation ($r = -0.774$) in magnitude, underscoring that knowledge-image evaluation is far from trivial and that direct LLM judgments lack transparency; AES-2.5 performs poorly ($r = 0.215$), capturing only surface aesthetics rather than semantic fidelity. These findings motivate `MMMG`'s knowledge-graph formulation as the only structure-and-knowledge-aware metric that reliably mirrors human judgments on complex, knowledge-dense visualizations.

## 4.3 MMMG-Score Variants

To validate the multiplicative design of the `MMMG-Score`, we compare it against two conventional aggregation forms and further analyze its stability under different readability weightings.

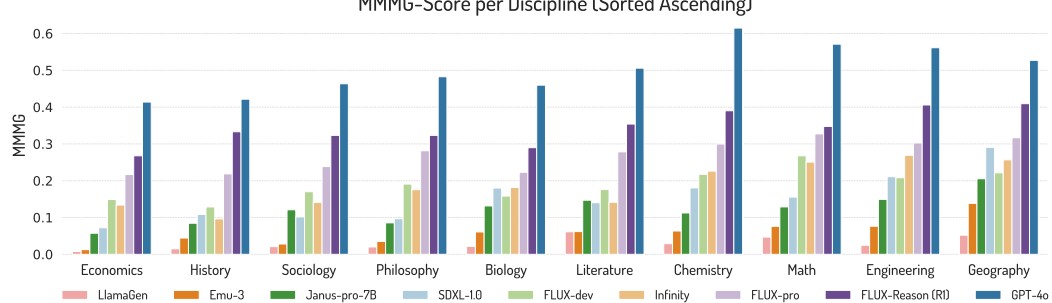

Figure 7: Discipline-level evaluation. Domains are sorted by average `MMMG-Score`.

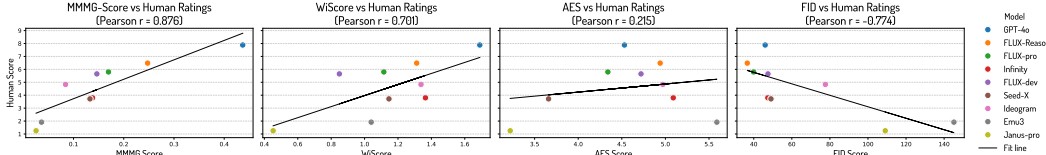

Figure 8: Illustration of Pearson Correlation among `MMMG-Score`, WiSCore, Aes-2.5 and FID.

Table 3: Comparison of average `MMMG-Score` across different aggregation methods and $\alpha$ values.

| Model | Arithmetic Mean | | | Geometric Mean | | | Stability Variant | | | | |
|---|---|---|---|---|---|---|---|---|---|---|---|
| $\alpha$ | 0.5 | 0.75 | 1 | 0.5 | 0.75 | 1 | 0.5 | 0.75 | 1 | 1.25 | 1.5 |
| GPT-4o | 72.36 | 60.12 | 50.49 | 64.30 | 54.55 | 50.49 | 50.28 | 50.25 | 50.20 | 50.17 | 50.15 |
| FLUX-Reason (R1-7B) | 63.37 | 46.81 | 32.10 | 46.52 | 36.63 | 32.10 | 31.41 | 31.32 | 31.26 | 31.14 | 31.07 |
| FLUX.1 [pro] | 59.97 | 44.06 | 29.91 | 43.27 | 32.49 | 29.91 | 27.31 | 27.16 | 27.14 | 26.67 | 26.51 |
| SDXL-1.0 | 47.35 | 36.79 | 25.37 | 29.48 | 25.71 | 25.37 | 17.19 | 16.14 | 15.90 | 14.61 | 14.04 |
| CogView4 | 45.79 | 32.05 | 20.24 | 25.13 | 20.41 | 20.24 | 13.73 | 13.13 | 13.10 | 12.40 | 12.02 |

**Arithmetic and Geometric Forms.** We define the arithmetic and geometric aggregation variants:

$$\texttt{MMMG-Score-A}(\alpha) = \alpha \cdot K_{\text{score}} + (1 - \alpha) \cdot R_{\text{score}}, \tag{3}$$

$$\texttt{MMMG-Score-G}(\alpha) = K_{\text{score}}^{\alpha} \cdot R_{\text{score}}^{(1-\alpha)}, \tag{4}$$

where $\alpha \in \{0.5, 0.75, 1\}$, $K_{\text{score}}$ and $R_{\text{score}}$ denote the knowledge fidelity and visual readability.

**Robustness Analysis.** To further examine robustness, we generalize the multiplicative form as:

$$\texttt{MMMG-Score}(\alpha) = K_{\text{score}} \cdot R_{\text{score}}^{\alpha}, \tag{5}$$

where $\alpha \in \{0.5, 0.75, 1, 1.25, 1.5\}$, interpolating $R_{\text{score}}$ between concave and convex regimes.

Table 3 reports results across these settings. We find that the arithmetic and geometric means are highly sensitive to $\alpha$. Thus, an inappropriate choice ($\alpha = 0.5$) may reduce the benchmark's discrimination. Conversely, the multiplicative form preserves both model rankings and absolute scores across all tested $\alpha$ values, indicating robustness to moderate weighting changes. Based on this analysis, we adopt the minimalist and interpretable multiplicative formulation `MMMG-Score` $= K_{\text{score}} \cdot R_{\text{score}}$.

## 4.4 Error Analysis

To better understand model limitations in structured visual reasoning, we conduct a systematic analysis of failure cases with low `MMMG-Score` ($\leq 0.5$). We categorize these into three types based on thresholds: **Visual Readability Failures** (readability score $\leq 0.5$), **Entity Representation Failures** (entity recall ratio $\leq 0.3$), and **Dependency Structure Failures** (dependency accuracy $\leq 0.4$). Figure 9(a) shows the error distribution across six top models, including `GPT-4o`, `FLUX-Reason` `(R1)`, `HiDream`, `FLUX.1-[pro]`, `SDXL-1.0`, and `Infinity`.

`GPT-4o` shows the fewest errors but struggles with visual dependency nomination. As shown in Figure 9(b, center), its motor diagram is visually coherent but misses key interactions (e.g., energy flow, containment), resulting in a dependency failure despite high visual clarity.

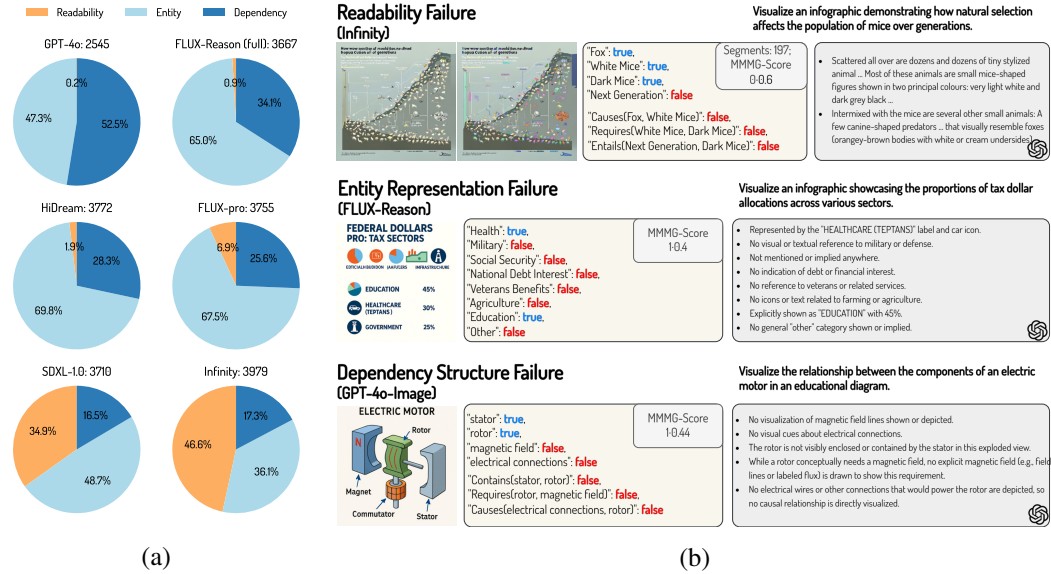

Figure 9: **Error analysis of generated knowledge images across models.** Common failure modes include missing entities (left), unclear relationships between concepts (center), and visually cluttered representations that reduce interpretability (right).

Middle-tier models like `FLUX-Reason` and `HiDream` can also generate clear visuals, but they tend to miss or ambiguously depict critical entities. This reflects a gap between CoT-driven planning or LLM-encoded prompts and mutual visual-text reasoning. For instance, in Figure 9(b), `FLUX-Reason` captures the intent of a tax allocation infographic, but fails to label or visually distinguish specified categories, leading to factual omissions.

Lower-performing models such as `Infinity` and `SDXL-1.0` suffer from visual clutter, poor layout, and unreadable text, making entity retrieval unreliable. Figure 9(b) shows how distorted elements hinder interpretation—an issue overlooked by LLM-only metrics but effectively penalized by our segmentation-aware method with `SAM-2.1`, ensuring fairer and more robust evaluation.

# 5 Conclusion

Knowledge images play a central role in human civilization and learning, and generating useful knowledge images is a fundamentally distinct and challenging task. It requires models to convey ideas through pixels via advanced multimodal reasoning across language and vision.To enable rigorous evaluation, we propose the `MMMG` benchmark, which assesses text-to-image reasoning using `MMMG-Score`—a metric combining graph edit distance and visual readability score based on coherent semantic regions. We also present FLUX-Reason as a strong baseline to facilitate future research. The `MMMG` benchmark has been released to HuggingFace at the time of submission, and the FLUX-Reason's model weights, source code, and training data will be released later.

**Limitations & Future Work.** We pose several important questions for future work. *How can we ensure accurate grounding of knowledge graphs in generated images?* This remains very challenging: we find that OpenAI-o3 still struggles to verify whether dozens of entities and relationships are present in a generated image. *How can we collect more high-quality knowledge images?* Although many such images exist across various textbooks, gathering them from these fragmented sources also poses a non-trivial challenge. *How can we close the performance gap to proprietary systems such as GPT-4o?* Although FLUX-Reason already improves its backbone by 15 points, future work should investigate multimodal pretraining, architectural improvements, and post-training procedures, together with systematic data-scaling experiments.

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

# NeurIPS Paper Checklist

The checklist is designed to encourage best practices for responsible machine learning research, addressing issues of reproducibility, transparency, research ethics, and societal impact. Do not remove the checklist: **The papers not including the checklist will be desk rejected.** The checklist should follow the references and follow the (optional) supplemental material. The checklist does NOT count towards the page limit.

Please read the checklist guidelines carefully for information on how to answer these questions. For each question in the checklist:

- You should answer [Yes] , [No] , or [NA] .
- [NA] means either that the question is Not Applicable for that particular paper or the relevant information is Not Available.
- Please provide a short (1–2 sentence) justification right after your answer (even for NA).

**The checklist answers are an integral part of your paper submission.** They are visible to the reviewers, area chairs, senior area chairs, and ethics reviewers. You will be asked to also include it (after eventual revisions) with the final version of your paper, and its final version will be published with the paper.

The reviewers of your paper will be asked to use the checklist as one of the factors in their evaluation. While "[Yes] " is generally preferable to "[No] ", it is perfectly acceptable to answer "[No] " provided a proper justification is given (e.g., "error bars are not reported because it would be too computationally expensive" or "we were unable to find the license for the dataset we used"). In general, answering "[No] " or "[NA] " is not grounds for rejection. While the questions are phrased in a binary way, we acknowledge that the true answer is often more nuanced, so please just use your best judgment and write a justification to elaborate. All supporting evidence can appear either in the main paper or the supplemental material, provided in appendix. If you answer [Yes] to a question, in the justification please point to the section(s) where related material for the question can be found.

IMPORTANT, please:

- **Delete this instruction block, but keep the section heading "NeurIPS Paper Checklist",**
- **Keep the checklist subsection headings, questions/answers and guidelines below.**
- **Do not modify the questions and only use the provided macros for your answers**.

1. **Claims**

   Question: Do the main claims made in the abstract and introduction accurately reflect the paper's contributions and scope?

   Answer: [Yes]

   Justification: The abstract and introduction explicitly introduce knowledge image generation task, alongside the Massive Multi-Discipline Multi-Tier Knowledge-Image Generation Benchmark (`MMMG`), highlighting the key contribution of this paper and research scope.

   Guidelines:
   - The answer NA means that the abstract and introduction do not include the claims made in the paper.
   - The abstract and/or introduction should clearly state the claims made, including the contributions made in the paper and important assumptions and limitations. A No or NA answer to this question will not be perceived well by the reviewers.
   - The claims made should match theoretical and experimental results, and reflect how much the results can be expected to generalize to other settings.
   - It is fine to include aspirational goals as motivation as long as it is clear that these goals are not attained by the paper.

2. **Limitations**

   Question: Does the paper discuss the limitations of the work performed by the authors?

   Answer: [Yes]

Justification: This paper provides a limitation discussion. We admit that OpenAI-o3 may fail to extract extremely complex knowledge graphs, but we need to clarify that OpenAI-o3 generally performs well on the constructed benchmark, that the impact of failure cases is limited, and that we provide several key statistics to address this concern in the appendix. To directly assess the accuracy and reliability of the evaluator (OpenAI-o3), we conducted a human vs. model comparison study. We evenly sampled 480 images across all educational levels and had domain experts annotate their corresponding knowledge graphs. MMMG-Scores computed by the LLM-based evaluator closely match those derived from expert annotations, with an average deviation of less than 1.5 points across all disciplines and difficulty levels. This confirms that, despite the theoretical limitations of using LLMs for graph extraction, the scoring metric remains reliable, consistent, and aligned with human judgment in practice.

Guidelines:

- The answer NA means that the paper has no limitation while the answer No means that the paper has limitations, but those are not discussed in the paper.
- The authors are encouraged to create a separate "Limitations" section in their paper.
- The paper should point out any strong assumptions and how robust the results are to violations of these assumptions (e.g., independence assumptions, noiseless settings, model well-specification, asymptotic approximations only holding locally). The authors should reflect on how these assumptions might be violated in practice and what the implications would be.
- The authors should reflect on the scope of the claims made, e.g., if the approach was only tested on a few datasets or with a few runs. In general, empirical results often depend on implicit assumptions, which should be articulated.
- The authors should reflect on the factors that influence the performance of the approach. For example, a facial recognition algorithm may perform poorly when image resolution is low or images are taken in low lighting. Or a speech-to-text system might not be used reliably to provide closed captions for online lectures because it fails to handle technical jargon.
- The authors should discuss the computational efficiency of the proposed algorithms and how they scale with dataset size.
- If applicable, the authors should discuss possible limitations of their approach to address problems of privacy and fairness.
- While the authors might fear that complete honesty about limitations might be used by reviewers as grounds for rejection, a worse outcome might be that reviewers discover limitations that aren't acknowledged in the paper. The authors should use their best judgment and recognize that individual actions in favor of transparency play an important role in developing norms that preserve the integrity of the community. Reviewers will be specifically instructed to not penalize honesty concerning limitations.

3. **Theory assumptions and proofs**

   Question: For each theoretical result, does the paper provide the full set of assumptions and a complete (and correct) proof?

   Answer: [Yes]

   Justification: The paper states the modeling assumptions and fully specifies its proposed evaluation framework. Specifically, MMMG-Score integrates a segmentation-based readability metric and a graph edit distance–based knowledge alignment score, both derived from proven algorithmic foundations. These components are designed to reflect the dual nature of knowledge images: clarity in visual presentation and accuracy in semantic structure. We also included an ablation study of different aggregation approaches, where either arithmetic or geometric means are highly sensitive to hyperparameters and lead to less discrimination. Exponential on readability score showcases robustness, thus we apply the minimalist and interpretable multiplicative formulation, enforcing the essential "AND" requirement for an effective knowledge image.

   Guidelines:

   - The answer NA means that the paper does not include theoretical results.

- All the theorems, formulas, and proofs in the paper should be numbered and cross-referenced.
- All assumptions should be clearly stated or referenced in the statement of any theorems.
- The proofs can either appear in the main paper or the supplemental material, but if they appear in the supplemental material, the authors are encouraged to provide a short proof sketch to provide intuition.
- Inversely, any informal proof provided in the core of the paper should be complemented by formal proofs provided in appendix or supplemental material.
- Theorems and Lemmas that the proof relies upon should be properly referenced.

4. **Experimental result reproducibility**

   Question: Does the paper fully disclose all the information needed to reproduce the main experimental results of the paper to the extent that it affects the main claims and/or conclusions of the paper (regardless of whether the code and data are provided or not)?

   Answer: [Yes]

   Justification: This paper gives a detailed experimental configures, and releases dataset, sampled images as well as evaluation code. We provide detailed installation instructions and carefully respond to every issue.

   Guidelines:

   - The answer NA means that the paper does not include experiments.
   - If the paper includes experiments, a No answer to this question will not be perceived well by the reviewers: Making the paper reproducible is important, regardless of whether the code and data are provided or not.
   - If the contribution is a dataset and/or model, the authors should describe the steps taken to make their results reproducible or verifiable.
   - Depending on the contribution, reproducibility can be accomplished in various ways. For example, if the contribution is a novel architecture, describing the architecture fully might suffice, or if the contribution is a specific model and empirical evaluation, it may be necessary to either make it possible for others to replicate the model with the same dataset, or provide access to the model. In general. releasing code and data is often one good way to accomplish this, but reproducibility can also be provided via detailed instructions for how to replicate the results, access to a hosted model (e.g., in the case of a large language model), releasing of a model checkpoint, or other means that are appropriate to the research performed.
   - While NeurIPS does not require releasing code, the conference does require all submissions to provide some reasonable avenue for reproducibility, which may depend on the nature of the contribution. For example
     (a) If the contribution is primarily a new algorithm, the paper should make it clear how to reproduce that algorithm.
     (b) If the contribution is primarily a new model architecture, the paper should describe the architecture clearly and fully.
     (c) If the contribution is a new model (e.g., a large language model), then there should either be a way to access this model for reproducing the results or a way to reproduce the model (e.g., with an open-source dataset or instructions for how to construct the dataset).
     (d) We recognize that reproducibility may be tricky in some cases, in which case authors are welcome to describe the particular way they provide for reproducibility. In the case of closed-source models, it may be that access to the model is limited in some way (e.g., to registered users), but it should be possible for other researchers to have some path to reproducing or verifying the results.

5. **Open access to data and code**

   Question: Does the paper provide open access to the data and code, with sufficient instructions to faithfully reproduce the main experimental results, as described in supplemental material?

   Answer: [Yes]

Justification: The paper provides open access to the data and code, with sufficient instructions to faithfully reproduce the main experiments.

Guidelines:

- The answer NA means that paper does not include experiments requiring code.
- Please see the NeurIPS code and data submission guidelines (`https://nips.cc/public/guides/CodeSubmissionPolicy`) for more details.
- While we encourage the release of code and data, we understand that this might not be possible, so "No" is an acceptable answer. Papers cannot be rejected simply for not including code, unless this is central to the contribution (e.g., for a new open-source benchmark).
- The instructions should contain the exact command and environment needed to run to reproduce the results. See the NeurIPS code and data submission guidelines (`https://nips.cc/public/guides/CodeSubmissionPolicy`) for more details.
- The authors should provide instructions on data access and preparation, including how to access the raw data, preprocessed data, intermediate data, and generated data, etc.
- The authors should provide scripts to reproduce all experimental results for the new proposed method and baselines. If only a subset of experiments are reproducible, they should state which ones are omitted from the script and why.
- At submission time, to preserve anonymity, the authors should release anonymized versions (if applicable).
- Providing as much information as possible in supplemental material (appended to the paper) is recommended, but including URLs to data and code is permitted.

6. **Experimental setting/details**

Question: Does the paper specify all the training and test details (e.g., data splits, hyperparameters, how they were chosen, type of optimizer, etc.) necessary to understand the results?

Answer: [Yes]

Justification: The paper specifies all the training and test details (e.g., data splits, hyperparameters, how they were chosen, type of optimizer, etc.).

Guidelines:

- The answer NA means that the paper does not include experiments.
- The experimental setting should be presented in the core of the paper to a level of detail that is necessary to appreciate the results and make sense of them.
- The full details can be provided either with the code, in appendix, or as supplemental material.

7. **Experiment statistical significance**

Question: Does the paper report error bars suitably and correctly defined or other appropriate information about the statistical significance of the experiments?

Answer: [Yes]

Justification: This paper provides an error analysis and details the statistics method.

Guidelines:

- The answer NA means that the paper does not include experiments.
- The authors should answer "Yes" if the results are accompanied by error bars, confidence intervals, or statistical significance tests, at least for the experiments that support the main claims of the paper.
- The factors of variability that the error bars are capturing should be clearly stated (for example, train/test split, initialization, random drawing of some parameter, or overall run with given experimental conditions).
- The method for calculating the error bars should be explained (closed form formula, call to a library function, bootstrap, etc.)
- The assumptions made should be given (e.g., Normally distributed errors).

- It should be clear whether the error bar is the standard deviation or the standard error of the mean.
- It is OK to report 1-sigma error bars, but one should state it. The authors should preferably report a 2-sigma error bar than state that they have a 96% CI, if the hypothesis of Normality of errors is not verified.
- For asymmetric distributions, the authors should be careful not to show in tables or figures symmetric error bars that would yield results that are out of range (e.g. negative error rates).
- If error bars are reported in tables or plots, The authors should explain in the text how they were calculated and reference the corresponding figures or tables in the text.

8. **Experiments compute resources**

Question: For each experiment, does the paper provide sufficient information on the computer resources (type of compute workers, memory, time of execution) needed to reproduce the experiments?

Answer: [Yes]

Justification: This paper details the compute resources of all experiments, including GPU numbers, hyper-parameters, training steps, etc.

Guidelines:

- The answer NA means that the paper does not include experiments.
- The paper should indicate the type of compute workers CPU or GPU, internal cluster, or cloud provider, including relevant memory and storage.
- The paper should provide the amount of compute required for each of the individual experimental runs as well as estimate the total compute.
- The paper should disclose whether the full research project required more compute than the experiments reported in the paper (e.g., preliminary or failed experiments that didn't make it into the paper).

9. **Code of ethics**

Question: Does the research conducted in the paper conform, in every respect, with the NeurIPS Code of Ethics https://neurips.cc/public/EthicsGuidelines?

Answer: [Yes]

Justification: This the research conducted in the paper conform with the NeurIPS Code of Ethics. All data collection, model training, and evaluation processes follow ethical standards, including safety, fairness, transparency, and respect for community norms. The MMMG-Benchmark data was collected via Bing image search with strict SafeSearch settings, and are limited to watermark-added previews. These thumbnails are intended to use solely for internal graph construction. GPT-4o images were created using OpenAI's public image generation API, which complies with global usage terms and passes strict content and safety filters. These images do not appear in the evaluation benchmark. Our work encodes cultural and information diversity. The training prompts in MMMG were generated using OpenAI-o3 model without explicit specification of race or cultural origin. The resulting distribution reflects the model's knowledge boundaries.

Guidelines:

- The answer NA means that the authors have not reviewed the NeurIPS Code of Ethics.
- If the authors answer No, they should explain the special circumstances that require a deviation from the Code of Ethics.
- The authors should make sure to preserve anonymity (e.g., if there is a special consideration due to laws or regulations in their jurisdiction).

10. **Broader impacts**

Question: Does the paper discuss both potential positive societal impacts and negative societal impacts of the work performed?

Answer: [Yes]

Justification: This paper discusses both the potential positive and negative societal impacts in the Conclusion section. On the positive side, the proposed MMMG dataset and FLUX-Reason framework aim to advance multimodal reasoning and enhance educational content generation, with promising applications in knowledge visualization and AI-assisted learning. On the negative side, the paper addresses potential risks through careful dataset curation, safety filtering, and ethical release practices to mitigate unintended harm.

Guidelines:

- The answer NA means that there is no societal impact of the work performed.
- If the authors answer NA or No, they should explain why their work has no societal impact or why the paper does not address societal impact.
- Examples of negative societal impacts include potential malicious or unintended uses (e.g., disinformation, generating fake profiles, surveillance), fairness considerations (e.g., deployment of technologies that could make decisions that unfairly impact specific groups), privacy considerations, and security considerations.
- The conference expects that many papers will be foundational research and not tied to particular applications, let alone deployments. However, if there is a direct path to any negative applications, the authors should point it out. For example, it is legitimate to point out that an improvement in the quality of generative models could be used to generate deepfakes for disinformation. On the other hand, it is not needed to point out that a generic algorithm for optimizing neural networks could enable people to train models that generate Deepfakes faster.
- The authors should consider possible harms that could arise when the technology is being used as intended and functioning correctly, harms that could arise when the technology is being used as intended but gives incorrect results, and harms following from (intentional or unintentional) misuse of the technology.
- If there are negative societal impacts, the authors could also discuss possible mitigation strategies (e.g., gated release of models, providing defenses in addition to attacks, mechanisms for monitoring misuse, mechanisms to monitor how a system learns from feedback over time, improving the efficiency and accessibility of ML).

11. **Safeguards**

Question: Does the paper describe safeguards that have been put in place for responsible release of data or models that have a high risk for misuse (e.g., pretrained language models, image generators, or scraped datasets)?

Answer: [Yes]

Justification: The MMMG dataset is developed and released under strict ethical and safety guidelines. From an initial pool of 500K web-scraped images, we apply multi-stage human and automated filtering to remove unsafe, sensitive, or community-harming content. After rigorous quality control, only 3,452 educational and knowledge-grounded images are retained. The final dataset excludes personal, political, violent, or otherwise harmful material, and avoids content from high-risk domains.

All model-generated samples are produced using the GPT-4o-Image API, which includes built-in content moderation and safety filtering. A panel of domain experts (in STEM, history, and philosophy) further reviewed all prompts and outputs, removing (i) factually incorrect or ambiguous prompts, (ii) ethically inappropriate or culturally sensitive content, and (iii) redundant or low-quality entries.

To prevent misuse, the dataset release includes clear terms of use, restricting applications to research purposes and requiring independent verification for any educational or public deployment. Moreover, our evaluation framework explicitly measures knowledge fidelity—offering a means to identify and mitigate risks of misinformation in AI-generated educational imagery.

Guidelines:

- The answer NA means that the paper poses no such risks.
- Released models that have a high risk for misuse or dual-use should be released with necessary safeguards to allow for controlled use of the model, for example by requiring that users adhere to usage guidelines or restrictions to access the model or implementing safety filters.

- Datasets that have been scraped from the Internet could pose safety risks. The authors should describe how they avoided releasing unsafe images.
- We recognize that providing effective safeguards is challenging, and many papers do not require this, but we encourage authors to take this into account and make a best faith effort.

12. **Licenses for existing assets**

Question: Are the creators or original owners of assets (e.g., code, data, models), used in the paper, properly credited and are the license and terms of use explicitly mentioned and properly respected?

Answer: [Yes]

Justification: All external assets used in this work are properly credited, and their licenses and terms of use are explicitly acknowledged and strictly adhered to. The MMMG-Benchmark data were collected through Bing Image Search using SafeSearch (strict) mode, ensuring compliance with Microsoft's content policies. Only watermark-bearing preview thumbnails were retrieved, and they are used solely for internal graph construction under fair use principles—no copyrighted or full-resolution content is distributed or reproduced in the released dataset.

Additionally, all model-generated images are produced via the official OpenAI GPT-4o Image API, which enforces comprehensive safety checks and adheres to OpenAI's published terms of use. These generated images serve exclusively as controlled training data for analysis purposes and are not included in the public evaluation benchmark. Thus, the paper fully respects all license conditions and ensures transparent and compliant use of third-party assets.

Guidelines:

- The answer NA means that the paper does not use existing assets.
- The authors should cite the original paper that produced the code package or dataset.
- The authors should state which version of the asset is used and, if possible, include a URL.
- The name of the license (e.g., CC-BY 4.0) should be included for each asset.
- For scraped data from a particular source (e.g., website), the copyright and terms of service of that source should be provided.
- If assets are released, the license, copyright information, and terms of use in the package should be provided. For popular datasets, paperswithcode.com/datasets has curated licenses for some datasets. Their licensing guide can help determine the license of a dataset.
- For existing datasets that are re-packaged, both the original license and the license of the derived asset (if it has changed) should be provided.
- If this information is not available online, the authors are encouraged to reach out to the asset's creators.

13. **New assets**

Question: Are new assets introduced in the paper well documented and is the documentation provided alongside the assets?

Answer: [Yes]

Justification: All new assets introduced in the paper, including datasets and models, are accompanied by clear documentation and usage instructions.

Guidelines:

- The answer NA means that the paper does not release new assets.
- Researchers should communicate the details of the dataset/code/model as part of their submissions via structured templates. This includes details about training, license, limitations, etc.
- The paper should discuss whether and how consent was obtained from people whose asset is used.

- At submission time, remember to anonymize your assets (if applicable). You can either create an anonymized URL or include an anonymized zip file.

14. **Crowdsourcing and research with human subjects**

Question: For crowdsourcing experiments and research with human subjects, does the paper include the full text of instructions given to participants and screenshots, if applicable, as well as details about compensation (if any)?

Answer: [Yes]

Justification: The appendix includes the full questionnaire format used in the human evaluation study, along with detailed task instructions. All participants were paid fairly for their time and effort, and provided informed consent prior to participation. The study setup, compensation details, and participant guidance are fully documented to ensure transparency and reproducibility.

Guidelines:

- The answer NA means that the paper does not involve crowdsourcing nor research with human subjects.
- Including this information in the supplemental material is fine, but if the main contribution of the paper involves human subjects, then as much detail as possible should be included in the main paper.
- According to the NeurIPS Code of Ethics, workers involved in data collection, curation, or other labor should be paid at least the minimum wage in the country of the data collector.

15. **Institutional review board (IRB) approvals or equivalent for research with human subjects**

Question: Does the paper describe potential risks incurred by study participants, whether such risks were disclosed to the subjects, and whether Institutional Review Board (IRB) approvals (or an equivalent approval/review based on the requirements of your country or institution) were obtained?

Answer: [NA]

Justification: The MMMG project does not involve human-subjects research as defined by standard research-ethics rules because the work did not (a) collect data through interaction or intervention with living individuals, nor (b) collect identifiable private information about individuals. MMMG is a benchmark of knowledge images composed from two sources: (i) public web imagery used as ground-truth / reference and (ii) images generated by models (GPT-4o/Image and other T2I systems) for evaluation and training purposes. The only human role in the dataset construction was expert validation / annotation (expert reviewers validated image–prompt pairs and knowledge graphs), and the released benchmark and evaluation toolkit do not include personal data or full-resolution copyrighted images.

Guidelines:

- The answer NA means that the paper does not involve crowdsourcing nor research with human subjects.
- Depending on the country in which research is conducted, IRB approval (or equivalent) may be required for any human subjects research. If you obtained IRB approval, you should clearly state this in the paper.
- We recognize that the procedures for this may vary significantly between institutions and locations, and we expect authors to adhere to the NeurIPS Code of Ethics and the guidelines for their institution.
- For initial submissions, do not include any information that would break anonymity (if applicable), such as the institution conducting the review.

16. **Declaration of LLM usage**

Question: Does the paper describe the usage of LLMs if it is an important, original, or non-standard component of the core methods in this research? Note that if the LLM is used only for writing, editing, or formatting purposes and does not impact the core methodology, scientific rigorousness, or originality of the research, declaration is not required.

Answer: [Yes]

Justification: Large Language Models (LLMs) constitute an essential methodological component of this work. Specifically, the OpenAI-o3 model is employed to evaluate reasoning quality and structured understanding within the MMMG benchmark. The LLM is used to (i) extract and interpret reasoning traces from model-generated descriptions, (ii) verify the logical consistency and factual correctness of structured outputs, and (iii) assist in automatic subgraph alignment and knowledge-fidelity assessment. To ensure scientific rigor, we conducted extensive analyses including expert–LLM alignment, cross-LLM robustness, and failure case statistics. The LLM usage is confined strictly to evaluation and benchmarking purposes, under compliant API-based access and within the terms of service of each provider. No human data, sensitive information, or downstream deployment decisions depend solely on LLM-generated outputs.

Guidelines:

- The answer NA means that the core method development in this research does not involve LLMs as any important, original, or non-standard components.
- Please refer to our LLM policy (https://neurips.cc/Conferences/2025/LLM) for what should or should not be described.

