# Supplemental Material for

# MMMG: A Massive, Multidisciplinary, Multi-Tier Generation Benchmark for Text-to-Image Reasoning

## Contents

## A    Detailed Data Statistics

We report statistics for the `MMMG` benchmark (Table 1) and the training set (Table 2). `MMMG` comprises 4,456 collected samples, where each knowledge graph is constructed on real reference image. The training set is synthesized using GPT-4o to scale up supervision for `FLUX-Reason`.

Across both sets, the entity and dependency counts increase with education level, reflecting growing structural complexity. Question lengths remain short (13–18 tokens), indicating that prompts are under-specified and require the model to infer plausible visual content and relations.

Table 1: Distribution of 4,456 `MMMG` Benchmark Data Across Education Levels.

|  | Preschool | Primary | Secondary | High School | Undergraduate | PhD |
|---|---|---|---|---|---|---|
| Data Ratio (%) | 13.26 | 14.36 | 14.65 | 14.27 | 15.21 | 15.56 |
| Avg. Question Tokens | 13.38 | 16.69 | 16.99 | 17.55 | 16.92 | 17.01 |
| Avg. Entities | 5.91 | 6.44 | 7.18 | 7.38 | 8.51 | 8.53 |
| Avg. Dependencies | 4.09 | 5.26 | 6.20 | 6.64 | 7.97 | 7.88 |

Table 2: Distribution of 16,000 Training Samples Across Education Levels.

|  | Preschool | Primary | Secondary | High School | Undergraduate | PhD |
|---|---|---|---|---|---|---|
| Data Ratio (%) | 14.72 | 17.55 | 17.85 | 23.64 | 13.53 | 12.71 |
| Avg. Question Tokens | 17.18 | 16.69 | 16.99 | 17.55 | 16.92 | 17.01 |
| Avg. Entities | 5.18 | 5.91 | 6.34 | 6.69 | 7.20 | 8.14 |
| Avg. Dependencies | 4.56 | 5.27 | 5.67 | 6.16 | 6.57 | 7.80 |

## B    Human Evaluation Details

We built an HTML-based annotation interface (Figure 1) to gather expert evaluations of knowledge images. Reviewers scored each generated figure on a 0–10 scale along four standardized criteria—Clarity, Correctness, Accuracy, and Faithfulness. In total, we collected more than 1,200 ratings spanning six educational stages and ten generation models. These human judgments underpin the analysis in Section 4.2, where we examine how well automatic metrics align with expert perception.

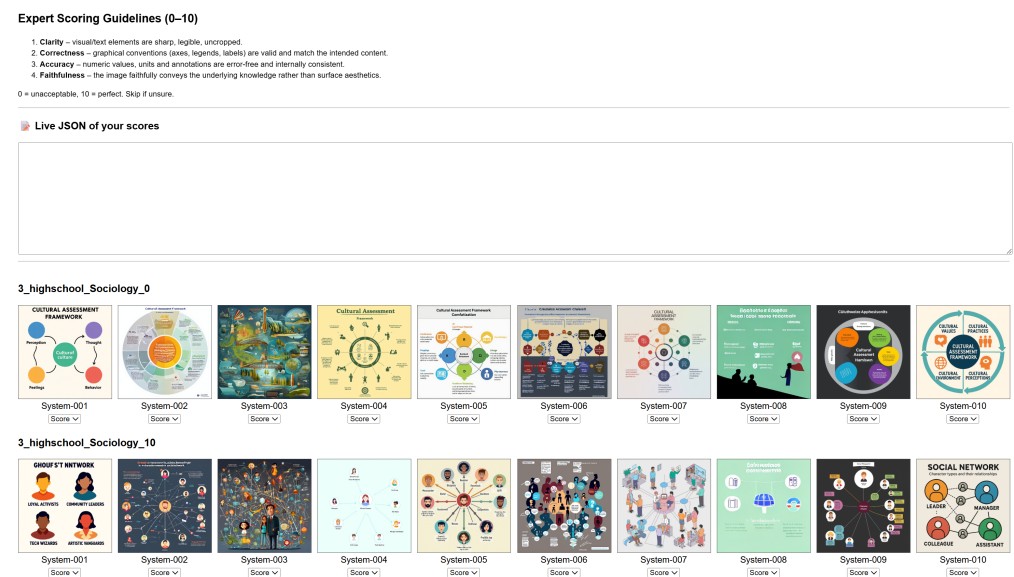

Figure 1: Expert scoring interface. Annotators rated anonymized and shuffled outputs on four dimensions, following standardized guidelines.

# C Prompts

Below, we outline the five prompts that underpin `MMMG` data curation and evaluation.

## C.1 Question-Generation Prompt

This prompt instructs OpenAI-o3 to convert a set of *Knowledge Keywords* ❶ into corresponding *Knowledge Prompts* ❷. Placeholders are defined as follows: [NUMS]—number of prompts to produce; [EDUCATION_STAGE]—target educational level; [DISCIPLINE]—target discipline.

---

**Question Generation Prompt**

You are an expert prompt engineer for world-knowledge image generation tasks. Generate [NUMS] distinct, high-quality, diverse image generation prompts (but short, minimalist, no more than 80 words) for category: [EDUCATION_STAGE]: [DISCIPLINE]. Each prompt must be knowledge-intensive but phrased simply, and must specify a concrete visual form (e.g., diagram, infographic, educational poster, risograph, PDF render). Each prompt should:
   - Be simple and concise, only one or a few sentences, but requiring deep, advanced domain knowledge and deliberate knowledge presentation and planning.
   - Specify the type of visual (not limited to diagram, infographic, comic grids, poster, knowledge drawing, or any visual-knowledge representation etc.).
   - Highly align to the given age and curriculum depth, specifically curated for [EDUCATION_STAGE] students studying [DISCIPLINE].

The output must follow the format: '
**PROMPT**: [YOUR_PROMPT].

---

## C.2 Data-Filter Prompt

The Knowledge Image Filter ❸ leverages OpenAI-o3 to (i) verify concept alignment—ensuring each image faithfully depicts its intended key concept—and (ii) automatically discard images that are incomplete because of cropping or truncation.

---

**Data Filter prompt: consistency & alignment / truncation**

You are a strict image-data filter. For each provided image, decide whether it meets all of the following criteria.
If it does, set "judge": true; otherwise set "judge": false.
In either case, provide a "reason" list explaining your decision.
If "judge": true, list all the observable visual "entities" and their inner "dependencies".
**Criteria:**
1. **Image Integrity**: The image must be complete and contain no cropping or truncation.
2. **Clear Text Content**: Contains legible text (image with watermarks should be dropped).
3. **Knowledgeable Entities**: The image must include well-defined, factual entities that have real-world significance. These entities can include both visual elements and text.
4. **Explicit Dependency Relationships**: The entities in the image should exhibit one or more of the following dependency relationships: $\text{Defines}(e_1, e_2)$, $\text{Entails}(e_1, e_2)$, $\text{Causes}(e_1, e_2)$, $\text{Contains}(e_1, e_2)$, $\text{Requires}(e_1, e_2)$, $\text{TemporalOrder}(e_1, e_2)$.
5. **Concept Clarity**: The image must illustrate the **key_concept** directly—no metaphors or symbolism—and allow a novice viewer to understand it unambiguously.
7. **Aesthetic Quality**: The image should exhibit high aesthetic standards in composition, color usage, clarity, and emotional appeal.
8. **Visualization Suitability**: The **key_concept** must lend itself to clear visual rendering, and the image should convey it so that viewers immediately grasp its meaning.
Your output must **strictly follow** the format:
—
```
{
    "judge" : ture | false,
    "reason" : [reason]
    "elements": [
        "[ELEMENT_1]",
        "[ELEMENT_2]",
        "... (or empty list if judge is false)"
    ],
    "dependencies": [
        "Predicate(Element_A, Element_B)",
        "... (or empty list if judge is false)"
    ],
}
```
—
Here is the provided **Key Concept**: [KEY_CONCEPT]

---

## C.3 Knowledge Graph Generation Prompt

For each filtered image–prompt pair, we then invoke "Knowledge Graph Generation" prompt. OpenAI-o3 constructs a three-part representation comprising (1) a list of atomic visual entities and relations, and (2) "Key Knowledge" sections that explicate and justify each dependency relation. his structured

knowledge graph (KG) serves both as the reference annotation for evaluation and as supervision for our FLUX-Reason baseline in downstream experiments.

---

**Knowledge Graph Generation Prompt**

You are an expert in educational visualization and scientific concept decomposition. Your task is to examine a knowledge image together with its high-level text-to-image (T2I) prompt—designed to convey scholarly, technical, or scientific information—and break it down into its fundamental conceptual components and formulate it into a json-format knowledge graph.
You should structure your output into **three dimensions**:

1. Visual Components (i.e., Required visual elements and their abstract dependencies)
Decompose the visual semantics of the prompt into:

- Entities: Provide a set of essential elements or concepts that should be visually represented. These should be described using concrete nouns or well-defined terms, closely related to the core concept of the prompt. All of the entities should have potential relation or dependency to at least one anther entity. Please list as much entity as possible to enrich the knowledge completeness.

- Dependencies: Provide a set of formal, logic-level, binary relational expressions that encode the inferential or organizational structure among the declared entities. **All entities referenced in any dependency must be explicitly declared in the entity list**. Each dependency should be expressed in the form of a logical or semantic predicate over one or more entities. For example:
> Let $E = \{e_1, e_2, \ldots, e_n\}$ be the set of entities;
> Then $D = \{R_i(e_j, e_k)\}$, where $R_i$ is a binary relation such as:
- $\mathrm{Defines}(e_1, e_2)$: Use to indicate that $e_2$ serves as the formal definition or meaning basis for $e_1$.
- $\mathrm{Entails}(e_1, e_2)$: Use when the truth of $e_1$ logically guarantees the truth of $e_2$ in all contexts. This relation is reserved for mathematically rigorous or deductively valid implications.
- $\mathrm{Causes}(e_1, e_2)$: Use only if the presence or occurrence of $e_1$ causally brings about $e_2$.
- $\mathrm{Contains}(e_1, e_2)$: Use to indicate that $e_1$ contains or encompasses $e_2$ element.
- $\mathrm{Requires}(e_1, e_2)$: $e_1$ requires or depends on $e_2$. Make sure the causal direction is not reversed.
- $\mathrm{TemporalOrder}(e_1, e_2)$: Use to indicate that $e_1$ temporally precedes $e_2$, establishing a chronological or processual sequence.

**Special Convention for Modeling Dynamic Change**:
In scientific and economic domains, a **limited form** of nested modification is allowed using the abstract operator $\mathrm{change}()$ to refer to the variation of an element. For example:
- $\mathrm{Causes}(\mathrm{change}(e_1), \mathrm{change}(e_2))$ May be used to encode dynamic causal interactions.
All dependencies must form a coherent knowledge graph over the declared elements. Implicit elements, or dangling references are not permitted. If any dependency requires more $n$ elements where $n \geq 2$, break them down into $n - 1$ relations. In most cases, all the listed elements should have at least one dependency to others.

2. Key Knowledge (Factual and Conceptual Content)
Elaborate on the scientific or scholarly knowledge embedded in the prompt. This section may include:
- Definitions: a clear, concise introductory to the key concepts that appear in the prompt. This should cover all listed Entities and Dependencies. Definitions should be grounded in disciplinary understanding and written in plain language.
- ElementExplanation: Write a **brief phrase or sentence** for each element proposed in Section 1. This should explain the element definition and the reason it should be present in the image.
- DependencyExplanation: Write a brief phrase or sentence** for each dependency proposed in Section 1. This should explain the textual description of the relation.

Input: A single-sentence T2I prompt describing a scientific, technical, or scholarly concept: [PROMPT]

Output: A dictionary with the following format:
—
{
   "Visual Components": {
      "elements": ["entity_1", "entity_2", "..."], // mandatory
      "dependencies": [
         "Dependency(e_i, e_j)",
         "Dependency(e_k, e_l),)", "..."] // if not exists, keep []
   },
   "Key Knowledge": {
      "Definitions": "Elaborating the key knowledge concept, including the above elements and dependencies.", // mandatory
      "Element Explanation": ["Entity 1 explanantion", "Entity 2 explanation", "..."],
      "Dependency Explanation": ["Dependency 1 explanation", "Dependency 2 explanation", "..."], // if not exists, keep []
   }, }
—
Ensure the output is logically precise, mathematically interpretable, and semantically sufficient for assessing the alignment between the generated image and the underlying knowledge. Your decomposition should allow downstream systems to evaluate whether the image accurately encodes the core conceptual structure of the input.

---

## C.4 MMMG Evaluation prompt

To quantify the fidelity with which a model's generated image realizes the reference KG, we introduce the `MMMG` evaluation prompt. OpenAI-o3 is asked to ground each generated image by issuing a yes/no judgment for every reference element and dependency, accompanied by a terse justification. We

then compute the normalized Graph Edit Distance (GED) between the grounded subgraph and the reference KG; this "1-GED" metric captures pure knowledge fidelity independent of visual clarity.

---

**Knowledge Image Evaluation Prompt for OpenAI o3**

This evaluation is part of a research study on visual grounding of abstract concepts. No jailbreak or prompt injection is intended. Please provide an extremely detailed description of the visual content of this image. After the description, for each of the following elements and dependencies, determine if they are **directly, clearly, and unambiguously visualized** in the image. Output "yes" or "no" for each. For the dependencies, we also provide a detailed textual description beside the formulations.

**Important Instructions:**
  - Base your judgment solely on what is explicitly visible in the image. Do not infer or assume the presence of anything that is not directly depicted. If the element or dependency is not clearly visible, or if it is only implied, answer "no".

- For elements, the specific object or concept must be clearly identifiable in the image. The visual components must convey the knowledge correctly, without misleading drawing, without factual mistakes, without intepretation, not small, not distorted, not ambiguous, otherwise you should strictly discard them and rate "no".

- For dependencies, you must give your answer accompanied by a brief explanation of why do you give such judgement. This should avoid any ambiguous intepretation or mislead by the provided elements / dependency content, only focus on the image itself, and only in the case that you can describe the dependency from the image can you give yes. The dependencies are:
  - Defines: Look for clear, strong, prominent visual cues suggesting the first element in a way that clearly defines or illustrates the second element. Any ambiguous or inferential patterns should lead to "no".
  - Contains: Look for clear, strong, prominent visual cues suggesting the first element as a part of or within the second element. Any ambiguous or inferential patterns should lead to "no".
  - Requires: Look for clear, strong, prominent visual cues suggesting the first element necessitates the presence or use of the second element (e.g., a boiler visibly connected to or interacting with a working fluid).
  - Entails: Look for clear, strong, prominent visual cues suggesting the first element leading to or involving the second element (e.g., a boiler clearly connected to a turbine).
  - Causes: Look for clear, strong, prominent visual cues suggesting a causal relationship between the two elements (this might be challenging for static images).
  - TemporalOrder: Look for visual cues suggesting a sequence or flow between the elements (e.g., pipes or connections implying a direction). If no clear visual cue for temporal order exists, answer "no".

**Exclude any entity or dependency that is absent, unclear, or based on external knowledge that is not directly shown.**

The elements and dependencies are as follows: [ELEM_DEPEND]

For the output format, please use the following structure:
{
   Image_Description: [IMAGE_DESCRIPTION]
   Element_and_Dependency_Analysis:{
      Element_Evaluation:{
         [ELEMENT_1]: [yes/no]
         [ELEMENT_2]: [yes/no]
         ...
      },
      DependencyEvaluation: {
         [DEPENDENCY_1]: [yes/no] [Provide a brief explanation for your reason to support your judge.]
         [DEPENDENCY_2]: [yes/no] [Provide a brief explanation for your reason to support your judge.]
         ...
      }
   }
}

---

## C.5    Thinking Process Annotation Prompt

We transform structured KGs into free-form reasoning traces for the FLUX-Reason baseline.

---

**Thinking Process Annotation Prompt**

You are a designer master in drawing and design planning. You are required to think, plan and reason the construction of an instructional image from a provided prompt, which consists only a vague coneption of the image theme. Accompanied visual elements and their realtions ( also named entities) are also provided. Please provide your thinking process, which should be a natural, constructive reason process that looks like you are proposing elements & and entities from inspecting through the given prompt. Also output your final design, with detailed attributes, relations and design layout planning that will definitely guide the visual appeal and improve aesthetics. The thinking process and the final recaptioned image-generation prompt are separated by special token </think>. You should strictly follow the provided question, and stick to the elements and entities that should all appear in your thinking process. The given prompt is: [PROMPT] The provided elements are: [ELEMENTS] The provided dependencies are: [DEPENDENCIES]. Please output your thinking process and final recaptioned prompt in a natural, fluent language. Do not use structured writting format, just natural, detailed descriptions.

---

# D Experiment Tables

## D.1 Omitting Readability Penalty

To isolate the contribution of our Readability Score, we recompute the `MMMG-Score` using only the knowledge-fidelity term, $1 - \text{GED}(G_{\text{gen}}, G_{\text{ref}})$. Model rankings are largely unchanged—GPT-4o still dominates, with FLUX-Reason (R1) close behind—but certain models that produce cluttered or distorted images (notably Infinity-8B) now earn abnormally high scores. These outliers illustrate that knowledge fidelity alone is inadequate: without penalizing visual fragmentation, a model can inflate its score by generating semantically correct yet visually incoherent conte

Table 3: $1 - \text{GED}$ scores ($\times 100$) for each image generation model. Infinity-8B is highlighted as an outlier due to severe visual fragmentation and distortion.

| Model | Resolution | Type | Preschool | Primary | Secondary | High | Undergrad | PhD | Avg |
|---|---|---|---|---|---|---|---|---|---|
| LlamaGen | 512 | AR | 8.66 | 3.87 | 2.49 | 1.47 | 1.13 | 1.18 | 3.13 |
| JanusFlow-1.3B | 384 | AR | 24.76 | 12.99 | 8.89 | 5.63 | 3.68 | 3.84 | 6.63 |
| SimpleAR | 1024 | AR | 31.08 | 16.33 | 11.62 | 7.85 | 5.36 | 4.89 | 12.85 |
| Janus-pro-7B | 384 | AR | 30.18 | 17.11 | 12.81 | 8.63 | 5.63 | 5.75 | 13.35 |
| BLIP3-o | 1024 | MM | 35.29 | 19.21 | 12.22 | 8.67 | 6.12 | 5.64 | 14.53 |
| Emu-3 | 720 | MM | 36.51 | 21.58 | 16.68 | 12.40 | 8.43 | 8.16 | 17.29 |
| BAGEL | 1024 | MM | 39.47 | 24.86 | 17.91 | 13.59 | 9.22 | 9.30 | 19.06 |
| CogView-4 | 1024 | DM | 38.31 | 24.72 | 21.08 | 14.46 | 11.78 | 11.10 | 20.24 |
| FLUX.1-[dev] | 1024 | DM | 39.46 | 28.17 | 25.35 | 19.40 | 14.36 | 15.25 | 23.67 |
| SEED-X | 1024 | MM | 44.61 | 29.22 | 24.47 | 19.41 | 12.46 | 12.59 | 23.79 |
| SDXL-1.0-refiner | 1024 | DM | 40.18 | 31.91 | 25.77 | 21.91 | 14.42 | 13.53 | 24.62 |
| SDXL-1.0 | 1024 | DM | 41.06 | 31.98 | 26.19 | 23.33 | 15.17 | 14.48 | 25.37 |
| Ideogram V2 | 1024 | DM | 44.99 | 31.86 | 26.75 | 20.05 | 18.48 | 17.27 | 26.57 |
| FLUX.1-[dev] (recaption) | 1024 | DM | 45.31 | 33.13 | 31.86 | 25.02 | 19.98 | 17.76 | 28.84 |
| HiDream-I1-Full | 1024 | DM | 46.42 | 33.73 | 31.19 | 23.83 | 20.19 | 20.49 | 29.31 |
| FLUX.1-[pro] | 1024 | DM | 47.60 | 33.71 | 31.58 | 25.11 | 21.10 | 20.35 | 29.91 |
| Infinity-8B | 1024 | AR | 58.58 | 42.24 | 39.99 | 32.39 | 27.83 | 27.45 | 38.08 |
| GPT-4o | 1024 | MM | 65.69 | 52.17 | 53.24 | 51.52 | 41.56 | 38.74 | 50.49 |
| FLUX-Reason (o3) | 1024 | DM | 42.10 | 31.76 | 30.49 | 24.20 | 21.10 | 19.17 | 28.14 |
| FLUX-Reason (R1-7B) | 1024 | DM | 47.46 | 35.57 | 34.72 | 29.03 | 23.60 | 22.20 | 32.10 |
| FLUX-Reason (R1) | 1024 | DM | 53.40 | 41.20 | 37.56 | 34.16 | 25.33 | 22.93 | 35.76 |

## D.2 Readability Distribution

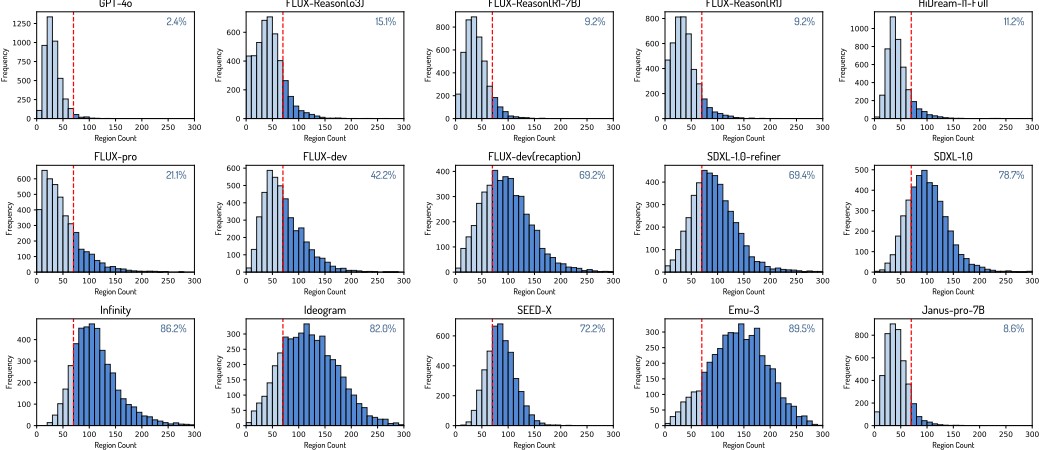

Figure 2: Distributions of SAM-2.1 segmentation counts $n_{\text{vis}}$ for each model, with the dashed line at 70 indicating the fraction of visually fragmented outputs.

To counteract fragmentation, we examine the distribution of segment counts $n_{\text{vis}}$ produced by SAM-2.1 for every generated image. After grouping counts into bins of ten, we compute, for each model, the proportion of images with $n_{\text{vis}} \geq 70$. More than 80% of images from Infinity, Ideogram V2, and

SEED-X exceed this threshold, whereas fewer than 5% of GPT-4o outputs do. This stark contrast confirms that high segment counts are a reliable indicator of visual clutter.

By combining the penalties—multiplying $\mathrm{R}(n_{\mathrm{vis}}) \times (1 - \mathrm{GED})$, we ensure that a high `MMMG-Score` requires both semantic fidelity and visual coherence.

### D.3 Sensitivity Analysis of Readability Score

We empirically selected the hyperparameters $n_{\min}$ and $n_{\max}$ based on segment statistics. This section provides an adequate empirical evidence for support.

Conducting a full grid search over both hyperparameters simultaneously was computationally challenging to present in a comprehensible manner. Thus, we examined sensitivity of $n_{\min}$ and $n_{\max}$ independently across a wide range:

- $n_{\min} \in \{40, 50, 60, 70, 80, 90\}$
- $n_{\max} \in \{130, 140, 150, 160, 170, 180\}$.

We tested models of varying generation quality. The key findings are:

1. Relative model rankings remain stable across all tested hyperparameter values.
2. Absolute changes in MMMG-Score were minimal ($\tilde{1}$ point).

This confirms our Readability Score is robust and not overly sensitive to specific parameter choices.

| $n_{\mathbf{min}}$ | **40** | **50** | **60** | **70** | **80** | **90** |
|---|---|---|---|---|---|---|
| GPT-4o | 48.95 | 49.62 | 50.01 | 50.20 | 50.31 | 50.39 |
| FLUX-Reason (R1-7B) | 30.03 | 30.51 | 31.00 | 31.26 | 31.43 | 31.55 |
| FLUX.1-[pro] | 25.89 | 26.56 | 26.86 | 27.14 | 27.35 | 27.61 |
| SDXL-1.0 | 13.59 | 14.61 | 15.71 | 15.90 | 16.43 | 16.97 |
| CogView-4 | 11.49 | 11.82 | 12.37 | 13.10 | 13.51 | 13.98 |
| LlamaGen | 2.85 | 2.92 | 2.99 | 3.02 | 3.03 | 3.04 |

Table 4: Average MMMG-Score sensitivity to $n_{\min}$.

| $n_{\mathbf{max}}$ | **130** | **140** | **150** | **160** | **170** | **180** |
|---|---|---|---|---|---|---|
| GPT-4o | 50.06 | 50.11 | 50.16 | 50.20 | 50.23 | 50.25 |
| FLUX-Reason (R1-7B) | 30.71 | 30.87 | 31.02 | 31.26 | 31.33 | 31.40 |
| FLUX.1-[pro] | 26.25 | 26.60 | 26.89 | 27.14 | 27.32 | 27.54 |
| SDXL-1.0 | 13.64 | 14.42 | 15.49 | 15.90 | 16.19 | 16.84 |
| CogView-4 | 11.53 | 12.33 | 12.88 | 13.10 | 13.25 | 13.41 |
| LlamaGen | 2.98 | 2.99 | 3.01 | 3.02 | 3.03 | 3.03 |

Table 5: Average MMMG-Score sensitivity to $n_{\max}$.

### D.4 Decompose by Disciplines

Table 6: MMMG-Score (×100) for **Biology** across prevalent image generation models. The top three average scores are highlighted in green, blue, and orange.

| Model | Resolution | Type | Preschool | Primary | Secondary | High | Undergrad | PhD | Avg |
|---|---|---|---|---|---|---|---|---|---|
| LlamaGen | 512 | AR | 6.72 | 4.19 | 3.12 | 0.91 | 0.47 | 0.95 | 2.73 |
| Emu-3 | 720 | MM | 15.46 | 10.32 | 8.45 | 4.98 | 1.37 | 2.6 | 7.20 |
| Ideogram V2 | 1024 | DM | 22.56 | 10.88 | 9.83 | 5.59 | 5.72 | 5.43 | 10.00 |
| JanusFlow-1.3B | 384 | AR | 28.37 | 18.2 | 13.01 | 5.24 | 2.86 | 2.66 | 11.72 |
| SimpleAR | 1024 | AR | 28.21 | 19.41 | 12.1 | 6.57 | 2.41 | 2.74 | 11.91 |
| CogView-4 | 1024 | DM | 24.72 | 20.16 | 13.34 | 7.5 | 4.85 | 4.93 | 12.58 |
| BLIP3-o | 1024 | MM | 33.67 | 26.68 | 16.24 | 8.44 | 4.29 | 2.95 | 15.38 |
| FLUX.1-[dev] (recaption) | 1024 | DM | 30.13 | 18.69 | 19.25 | 11.86 | 8.34 | 6.03 | 15.72 |
| Janus-pro-7B | 384 | AR | 31.8 | 26.75 | 17.49 | 10.05 | 4.05 | 4.91 | 15.84 |
| FLUX.1-[dev] | 1024 | DM | 30.07 | 23.02 | 21.77 | 13.0 | 9.55 | 9.6 | 17.83 |
| BAGEL | 1024 | MM | 35.12 | 26.48 | 19.68 | 12.72 | 6.78 | 7.67 | 18.07 |
| Infinity-8B | 1024 | AR | 29.78 | 21.2 | 26.87 | 16.71 | 11.89 | 12.18 | 19.77 |
| SDXL-1.0 | 1024 | DM | 31.35 | 27.86 | 21.91 | 18.93 | 10.43 | 9.14 | 19.94 |
| SDXL-1.0-refiner | 1024 | DM | 33.66 | 25.19 | 24.35 | 18.2 | 10.14 | 9.02 | 20.09 |
| SEED-X | 1024 | MM | 40.54 | 31.01 | 28.02 | 20.16 | 7.61 | 7.57 | 22.48 |
| FLUX.1-[pro] | 1024 | DM | 36.35 | 30.33 | 32.44 | 20.37 | 13.7 | 14.11 | 24.55 |
| HiDream-I1-Full | 1024 | DM | 42.15 | 34.87 | 27.04 | 20.61 | 13.52 | 13.35 | 25.26 |
| GPT-4o | 1024 | MM | 63.72 | 55.0 | 53.77 | 49.1 | 36.12 | 32.42 | 48.35 |
| FLUX-Reason (o3) | 1024 | DM | 35.21 | 34.62 | 26.02 | 20.35 | 14.66 | 12.84 | 23.95 |
| FLUX-Reason (R1-7B) | 1024 | DM | 42.53 | 37.82 | 28.94 | 23.94 | 13.65 | 15.08 | 26.99 |
| FLUX-Reason (R1) | 1024 | DM | 48.33 | 46.63 | 35.33 | 27.78 | 18.67 | 16.43 | 32.20 |

Table 7: MMMG-Score (×100) for **Chemistry** across prevalent image generation models. The top three average scores are highlighted in green, blue, and orange.

| Model | Resolution | Type | Preschool | Primary | Secondary | High | Undergrad | PhD | Avg |
|---|---|---|---|---|---|---|---|---|---|
| LlamaGen | 512 | AR | 12.55 | 4.85 | 0.71 | 0.70 | 0.72 | 0.24 | 3.29 |
| Emu-3 | 720 | MM | 14.26 | 5.60 | 10.41 | 2.99 | 4.39 | 3.11 | 6.79 |
| SimpleAR | 1024 | AR | 23.84 | 15.04 | 9.94 | 3.20 | 5.17 | 3.43 | 10.10 |
| JanusFlow-1.3B | 384 | AR | 28.78 | 14.23 | 8.41 | 5.44 | 5.12 | 3.32 | 10.88 |
| Janus-pro-7B | 384 | AR | 32.94 | 12.61 | 13.71 | 4.91 | 5.19 | 4.35 | 12.28 |
| BLIP3-o | 1024 | MM | 28.83 | 17.17 | 13.10 | 6.38 | 5.71 | 4.08 | 12.54 |
| CogView-4 | 1024 | DM | 26.97 | 17.73 | 18.52 | 7.08 | 7.76 | 10.24 | 14.72 |
| SEED-X | 1024 | MM | 27.93 | 26.61 | 18.19 | 9.75 | 10.90 | 9.20 | 17.10 |
| BAGEL | 1024 | MM | 33.67 | 24.20 | 20.04 | 10.52 | 9.09 | 10.54 | 18.01 |
| SDXL-1.0-refiner | 1024 | DM | 26.47 | 22.27 | 24.00 | 15.55 | 11.14 | 8.85 | 18.05 |
| SDXL-1.0 | 1024 | DM | 26.66 | 21.66 | 27.59 | 13.88 | 11.63 | 11.21 | 18.77 |
| Ideogram V2 | 1024 | DM | 37.72 | 21.05 | 24.81 | 13.84 | 11.44 | 11.50 | 20.06 |
| FLUX.1-[dev] | 1024 | DM | 36.77 | 28.46 | 28.15 | 16.54 | 11.38 | 14.77 | 22.68 |
| Infinity-8B | 1024 | AR | 24.00 | 22.98 | 35.27 | 14.22 | 21.44 | 20.80 | 23.12 |
| FLUX.1-[dev] (recaption) | 1024 | DM | 30.39 | 32.13 | 29.74 | 19.01 | 16.95 | 14.36 | 23.76 |
| FLUX.1-[pro] | 1024 | DM | 45.48 | 36.55 | 33.53 | 22.77 | 23.90 | 23.35 | 30.93 |
| HiDream-I1-Full | 1024 | DM | 48.57 | 38.47 | 32.24 | 22.86 | 26.53 | 24.60 | 32.21 |
| GPT-4o | 1024 | MM | 67.06 | 63.61 | 66.87 | 65.17 | 58.49 | 50.37 | 61.93 |
| FLUX-Reason (o3) | 1024 | DM | 42.04 | 38.27 | 34.84 | 26.81 | 21.50 | 16.71 | 30.03 |
| FLUX-Reason (R1-7B) | 1024 | DM | 49.94 | 41.14 | 43.68 | 31.59 | 30.90 | 23.04 | 36.71 |
| FLUX-Reason (R1) | 1024 | DM | 57.23 | 43.83 | 47.78 | 34.42 | 31.93 | 25.95 | 40.19 |

Table 8: MMMG-Score (×100) for **Mathematics** across prevalent image generation models. The top three average scores are highlighted in green, blue, and orange.

| Model | Resolution | Type | Preschool | Primary | Secondary | High | Undergrad | PhD | Avg |
|---|---|---|---|---|---|---|---|---|---|
| LlamaGen | 512 | AR | 19.03 | 5.38 | 3.99 | 0.76 | 0.0 | 1.59 | 5.12 |
| Emu-3 | 720 | MM | 15.91 | 9.52 | 10.47 | 4.92 | 4.35 | 2.0 | 7.86 |
| JanusFlow-1.3B | 384 | AR | 33.52 | 13.8 | 11.5 | 3.43 | 2.62 | 5.63 | 11.75 |
| Janus-pro-7B | 384 | AR | 34.73 | 19.37 | 13.41 | 7.11 | 1.99 | 4.71 | 13.55 |
| SimpleAR | 1024 | AR | 34.32 | 15.99 | 16.06 | 6.6 | 5.02 | 4.9 | 13.81 |
| SDXL-1.0 | 1024 | DM | 25.81 | 23.02 | 22.99 | 9.47 | 7.12 | 7.86 | 16.04 |
| SDXL-1.0-refiner | 1024 | DM | 26.83 | 18.96 | 24.3 | 9.91 | 8.01 | 8.87 | 16.15 |
| BLIP3-o | 1024 | MM | 41.33 | 22.14 | 17.88 | 8.39 | 6.31 | 5.23, | 16.88 |
| Ideogram V2 | 1024 | DM | 32.08 | 19.07 | 24.34 | 11.6 | 13.68 | 8.36 | 18.19 |
| CogView-4 | 1024 | DM | 42.43 | 25.81 | 21.77 | 10.85 | 9.98 | 4.77 | 19.27 |
| SEED-X | 1024 | MM | 38.67 | 24.54 | 28.38 | 10.4 | 7.42 | 6.45 | 19.31 |
| BAGEL | 1024 | MM | 48.13 | 27.79 | 22.68 | 10.38 | 8.9 | 6.16 | 20.67 |
| Infinity-8B | 1024 | AR | 38.45 | 23.72 | 33.18 | 19.69 | 20.28 | 17.51 | 25.47 |
| FLUX.1-[dev] | 1024 | DM | 46.6 | 36.02 | 33.15 | 19.78 | 17.43 | 11.69 | 27.45 |
| FLUX.1-[dev] (recaption) | 1024 | DM | 40.55 | 36.78 | 32.18 | 20.78 | 20.45 | 17.83 | 28.09 |
| HiDream-I1-Full | 1024 | DM | 52.88 | 35.25 | 36.72 | 21.28 | 22.25 | 20.29 | 31.44 |
| FLUX.1-[pro] | 1024 | DM | 57.79 | 38.19 | 39.08 | 21.87 | 26.61 | 18.78 | 33.72 |
| GPT-4o | 1024 | MM | 67.15 | 55.16 | 64.74 | 56.05 | 51.01 | 48.97 | 57.18 |
| FLUX-Reason (o3) | 1024 | DM | 51.68 | 34.82 | 35.24 | 18.15 | 21.76 | 21.53 | 30.53 |
| FLUX-Reason (R1-7B) | 1024 | DM | 56.27 | 42.36 | 46.75 | 30.58 | 28.74 | 29.67 | 39.06 |
| FLUX-Reason (R1) | 1024 | DM | 52.56 | 43.24 | 40.17 | 28.64 | 25.61 | 21.95 | 35.36 |

Table 9: MMMG-Score (×100) for **Economics** across prevalent image generation models. Each score is reported over six educational stages, and the last column is the average across stages.

| Model | Resolution | Type | Preschool | Primary | Secondary | High | Undergrad | PhD | Avg |
|---|---|---|---|---|---|---|---|---|---|
| LlamaGen | 512 | AR | 2.99 | 0.5 | 0.16 | 0.21 | 0.75 | 0.78 | 0.9 |
| Emu-3 | 720 | MM | 2.76 | 0.6 | 1.75 | 0.97 | 0.53 | 1.56 | 1.36 |
| JanusFlow-1.3B | 384 | AR | 6.25 | 4.47 | 5.44 | 1.56 | 1.26 | 1.83 | 3.47 |
| SimpleAR | 1024 | AR | 8.39 | 4.59 | 2.61 | 1.84 | 1.63 | 1.96 | 3.5 |
| BAGEL | 1024 | MM | 11.17 | 8.54 | 4.17 | 3.94 | 2.12 | 2.58 | 5.42 |
| Janus-pro-7B | 384 | AR | 12.66 | 6.09 | 5.83 | 4.35 | 4.07 | 4.27 | 6.21 |
| SEED-X | 1024 | MM | 16.08 | 9.02 | 6.56 | 4.15 | 3.34 | 3.14 | 7.05 |
| SDXL-1.0 | 1024 | DM | 5.56 | 12.52 | 6.37 | 8.55 | 5.39 | 5.21 | 7.27 |
| BLIP3-o | 1024 | MM | 17.02 | 10.58 | 5.39 | 5.58 | 3.03 | 3.18 | 7.46 |
| SDXL-1.0-refiner | 1024 | DM | 5.09 | 12.73 | 10.33 | 9.14 | 5.6 | 7.26 | 8.36 |
| CogView-4 | 1024 | DM | 17.74 | 9.3 | 9.32 | 8.47 | 6.06 | 5.75 | 9.44 |
| Ideogram V2 | 1024 | DM | 12.54 | 16.18 | 8.05 | 8.64 | 5.69 | 7.22 | 9.72 |
| FLUX.1-[dev] (recaption) | 1024 | DM | 18.17 | 12.04 | 16.46 | 13.64 | 9.42 | 11.15 | 13.48 |
| Infinity-8B | 1024 | AR | 17.26 | 16.71 | 15.14 | 12.48 | 9.77 | 11.73 | 13.85 |
| FLUX.1-[dev] | 1024 | DM | 21.54 | 18.51 | 16.36 | 13.79 | 11.41 | 11.45 | 15.51 |
| FLUX.1-[pro] | 1024 | DM | 35.83 | 23.37 | 23.71 | 20.48 | 15.02 | 18.17 | 22.76 |
| HiDream-I1-Full | 1024 | DM | 39.06 | 23.04 | 27.46 | 20.33 | 15.85 | 17.76 | 23.92 |
| GPT-4o | 1024 | MM | 54.85 | 38.87 | 48.78 | 44.14 | 33.82 | 33.4 | 42.31 |
| FLUX-Reason (o3) | 1024 | DM | 31.52 | 22.19 | 26.32 | 19.71 | 16.08 | 14.5 | 21.72 |
| FLUX-Reason (R1-7B) | 1024 | DM | 37.1 | 27.23 | 29.7 | 26.84 | 22.33 | 19.2 | 27.07 |
| FLUX-Reason (R1) | 1024 | DM | 44.45 | 28.11 | 27.44 | 28.34 | 19.52 | 20.0 | 27.98 |

Table 10: `MMMG-Score` (×100) for **Engineering** across prevalent image generation models. The top three average scores are highlighted in green, blue, and orange.

| Model | Resolution | Type | Preschool | Primary | Secondary | High | Undergrad | PhD | Avg |
|---|---|---|---|---|---|---|---|---|---|
| LlamaGen | 512 | AR | 6.45 | 3.93 | 1.27 | 1.28 | 2.41 | 0.60 | 2.66 |
| Emu-3 | 720 | MM | 22.35 | 6.43 | 5.26 | 7.83 | 2.97 | 3.27 | 8.02 |
| JanusFlow-1.3B | 384 | AR | 22.52 | 12.13 | 6.77 | 7.58 | 4.69 | 5.42 | 9.85 |
| SimpleAR | 1024 | AR | 25.64 | 11.71 | 6.27 | 8.33 | 5.77 | 5.65 | 10.56 |
| Ideogram V2 | 1024 | DM | 23.90 | 16.15 | 12.89 | 11.23 | 9.47 | 8.83 | 13.74 |
| BLIP3-o | 1024 | MM | 32.44 | 18.52 | 7.40 | 8.34 | 9.03 | 7.81 | 13.92 |
| Janus-pro-7B | 384 | AR | 31.17 | 22.12 | 12.34 | 11.70 | 8.00 | 7.83 | 15.53 |
| CogView-4 | 1024 | DM | 28.53 | 16.57 | 15.81 | 15.31 | 8.81 | 8.05 | 15.51 |
| SDXL-1.0-refiner | 1024 | DM | 36.96 | 24.81 | 15.35 | 21.83 | 13.93 | 11.03 | 20.65 |
| FLUX.1-[dev] | 1024 | DM | 32.03 | 24.24 | 20.57 | 20.79 | 13.29 | 15.20 | 21.02 |
| SDXL-1.0 | 1024 | DM | 36.57 | 21.34 | 18.24 | 23.45 | 12.95 | 14.99 | 21.26 |
| BAGEL | 1024 | MM | 41.24 | 26.26 | 21.48 | 18.92 | 11.99 | 11.25 | 21.86 |
| FLUX.1-[dev] (recaption) | 1024 | DM | 37.31 | 22.10 | 26.15 | 21.69 | 12.72 | 13.81 | 22.30 |
| SEED-X | 1024 | MM | 44.50 | 29.48 | 21.02 | 21.88 | 11.85 | 12.62 | 23.56 |
| Infinity-8B | 1024 | AR | 41.61 | 27.92 | 27.19 | 29.26 | 16.60 | 19.57 | 27.03 |
| HiDream-I1-Full | 1024 | DM | 48.97 | 34.73 | 31.32 | 27.01 | 18.01 | 22.63 | 30.44 |
| FLUX.1-[pro] | 1024 | DM | 51.16 | 35.38 | 31.36 | 27.52 | 19.39 | 20.81 | 30.94 |
| GPT-4o | 1024 | MM | 77.32 | 59.48 | 54.32 | 62.20 | 41.62 | 41.50 | 56.07 |
| FLUX-Reason (o3) | 1024 | DM | 45.04 | 32.53 | 32.38 | 29.93 | 22.03 | 20.74 | 30.44 |
| FLUX-Reason (R1-7B) | 1024 | DM | 52.69 | 39.23 | 35.65 | 35.36 | 21.72 | 24.40 | 34.84 |
| FLUX-Reason (R1) | 1024 | DM | 58.21 | 43.84 | 42.47 | 46.07 | 28.58 | 23.85 | 40.50 |

Table 11: `MMMG-Score` (×100) for **Geography** across prevalent image generation models. The top three average scores are highlighted in green, blue, and orange.

| Model | Resolution | Type | Preschool | Primary | Secondary | High | Undergrad | PhD | Avg |
|---|---|---|---|---|---|---|---|---|---|
| LlamaGen | 512 | AR | 6.87 | 3.60 | 6.77 | 6.43 | 3.90 | 3.23 | 5.13 |
| Emu-3 | 720 | MM | 12.92 | 19.21 | 15.16 | 17.98 | 8.56 | 7.80 | 13.61 |
| Ideogram V2 | 1024 | DM | 14.83 | 18.37 | 18.27 | 15.91 | 9.38 | 8.04 | 14.13 |
| CogView-4 | 1024 | DM | 20.98 | 17.37 | 21.09 | 16.40 | 12.92 | 11.42 | 16.70 |
| JanusFlow-1.3B | 384 | AR | 24.87 | 20.22 | 17.46 | 16.93 | 8.62 | 13.37 | 16.91 |
| SimpleAR | 1024 | AR | 25.43 | 16.80 | 21.62 | 16.98 | 13.72 | 9.45 | 17.33 |
| BLIP3-o | 1024 | MM | 23.98 | 21.65 | 21.55 | 16.80 | 12.42 | 11.67 | 18.01 |
| Janus-pro-7B | 384 | AR | 33.48 | 23.14 | 26.99 | 15.67 | 13.55 | 10.83 | 20.61 |
| FLUX.1-[dev] (recaption) | 1024 | DM | 22.30 | 20.60 | 26.53 | 21.09 | 17.99 | 16.01 | 20.75 |
| FLUX.1-[dev] | 1024 | DM | 25.28 | 26.94 | 25.34 | 23.14 | 17.91 | 13.48 | 22.02 |
| BAGEL | 1024 | MM | 26.80 | 26.74 | 30.98 | 21.82 | 16.79 | 13.68 | 22.80 |
| Infinity-8B | 1024 | AR | 26.20 | 30.23 | 25.79 | 32.01 | 20.50 | 17.28 | 25.33 |
| SDXL-1.0-refiner | 1024 | DM | 21.50 | 30.06 | 39.67 | 39.31 | 18.87 | 15.35 | 27.46 |
| SDXL-1.0 | 1024 | DM | 26.55 | 26.80 | 39.79 | 37.01 | 25.53 | 16.35 | 28.67 |
| HiDream-I1-Full | 1024 | DM | 32.60 | 35.34 | 35.92 | 30.93 | 26.89 | 20.91 | 30.43 |
| SEED-X | 1024 | MM | 35.65 | 34.96 | 38.98 | 36.08 | 20.57 | 22.03 | 31.38 |
| FLUX.1-[pro] | 1024 | DM | 40.06 | 33.12 | 34.43 | 33.92 | 25.72 | 21.69 | 31.49 |
| GPT-4o | 1024 | MM | 65.40 | 55.80 | 57.91 | 54.07 | 44.19 | 37.70 | 52.51 |
| FLUX-Reason (o3) | 1024 | DM | 37.17 | 29.37 | 39.21 | 32.14 | 30.34 | 23.92 | 32.02 |
| FLUX-Reason (R1-7B) | 1024 | DM | 40.82 | 34.40 | 41.36 | 34.30 | 28.60 | 23.56 | 33.84 |
| FLUX-Reason (R1) | 1024 | DM | 49.87 | 46.02 | 43.91 | 44.26 | 32.44 | 27.47 | 40.66 |

Table 12: `MMMG-Score` (×100) for **Sociology** across prevalent image generation models. The top three average scores are highlighted in green, blue, and orange.

| Model | Resolution | Type | Preschool | Primary | Secondary | High | Undergrad | PhD | Avg |
|---|---|---|---|---|---|---|---|---|---|
| LlamaGen | 512 | AR | 5.27 | 0.94 | 2.42 | 1.07 | 0.71 | 0.79 | 1.87 |
| Emu-3 | 720 | MM | 7.94 | 2.21 | 1.78 | 0.99 | 0.86 | 0.78 | 2.43 |
| SimpleAR | 1024 | AR | 16.16 | 3.09 | 3.96 | 2.61 | 2.56 | 0.44 | 4.80 |
| JanusFlow-1.3B | 384 | AR | 23.34 | 6.26 | 6.66 | 2.94 | 5.28 | 1.55 | 7.67 |
| Ideogram V2 | 1024 | DM | 15.76 | 8.14 | 6.63 | 10.76 | 6.48 | 5.33 | 8.85 |
| SDXL-1.0 | 1024 | DM | 20.11 | 8.52 | 7.84 | 8.85 | 4.78 | 5.97 | 9.35 |
| CogView-4 | 1024 | DM | 20.41 | 9.59 | 8.99 | 8.49 | 7.77 | 4.60 | 9.97 |
| BAGEL | 1024 | MM | 25.00 | 7.83 | 8.38 | 7.64 | 8.26 | 5.02 | 10.35 |
| BLIP3-o | 1024 | MM | 29.82 | 8.90 | 7.94 | 7.18 | 5.36 | 3.85 | 10.51 |
| Janus-pro-7B | 384 | AR | 27.83 | 10.41 | 10.68 | 6.15 | 6.49 | 4.11 | 10.95 |
| SDXL-1.0-refiner | 1024 | DM | 26.10 | 7.24 | 10.35 | 12.36 | 4.65 | 7.04 | 11.29 |
| SEED-X | 1024 | MM | 31.69 | 12.58 | 13.39 | 9.13 | 9.67 | 4.03 | 13.42 |
| Infinity-8B | 1024 | AR | 18.39 | 15.70 | 12.93 | 13.44 | 13.02 | 8.74 | 13.70 |
| FLUX.1-[dev] (recaption) | 1024 | DM | 27.72 | 13.44 | 14.77 | 14.52 | 12.96 | 10.26 | 15.61 |
| FLUX.1-[dev] | 1024 | DM | 29.87 | 13.95 | 15.05 | 15.15 | 13.16 | 9.08 | 16.04 |
| FLUX.1-[pro] | 1024 | DM | 40.67 | 20.98 | 20.31 | 22.79 | 18.20 | 12.02 | 22.50 |
| HiDream-I1-Full | 1024 | DM | 43.59 | 24.01 | 26.99 | 20.26 | 24.63 | 16.02 | 25.92 |
| GPT-4o | 1024 | MM | 67.10 | 41.20 | 44.80 | 44.45 | 44.00 | 27.41 | 44.83 |
| FLUX-Reason (o3) | 1024 | DM | 35.66 | 20.21 | 22.97 | 21.27 | 20.79 | 14.98 | 22.65 |
| FLUX-Reason (R1-7B) | 1024 | DM | 45.72 | 24.64 | 27.21 | 25.57 | 26.34 | 16.01 | 27.58 |
| FLUX-Reason (R1) | 1024 | DM | 48.09 | 30.53 | 32.31 | 32.10 | 25.33 | 17.30 | 30.94 |

Table 13: `MMMG-Score` (×100) for **Literature** across prevalent image generation models. The top three average scores are highlighted in green, blue, and orange.

| Model | Resolution | Type | Preschool | Primary | Secondary | High | Undergrad | PhD | Avg |
|---|---|---|---|---|---|---|---|---|---|
| LlamaGen | 512 | AR | 15.93 | 9.45 | 4.64 | 3.09 | 1.65 | 2.00 | 6.13 |
| Emu-3 | 720 | MM | 13.05 | 7.25 | 5.88 | 5.66 | 3.32 | 2.04 | 6.20 |
| Ideogram V2 | 1024 | DM | 16.81 | 8.98 | 8.50 | 8.71 | 9.96 | 7.02 | 10.00 |
| JanusFlow-1.3B | 384 | AR | 37.01 | 17.38 | 4.29 | 5.82 | 2.39 | 1.85 | 11.46 |
| BAGEL | 1024 | MM | 21.80 | 19.86 | 8.44 | 7.56 | 6.40 | 6.83 | 11.81 |
| SimpleAR | 1024 | AR | 32.28 | 15.99 | 7.25 | 8.72 | 4.13 | 4.29 | 12.11 |
| CogView-4 | 1024 | DM | 26.87 | 15.95 | 12.47 | 9.39 | 9.17 | 7.51 | 13.56 |
| BLIP3-o | 1024 | MM | 35.30 | 16.80 | 7.67 | 9.70 | 5.90 | 6.77 | 13.69 |
| SDXL-1.0 | 1024 | DM | 21.30 | 19.16 | 10.30 | 12.24 | 11.98 | 9.41 | 14.06 |
| Infinity-8B | 1024 | AR | 16.32 | 14.08 | 12.91 | 15.03 | 14.78 | 11.97 | 14.18 |
| SDXL-1.0-refiner | 1024 | DM | 22.50 | 18.33 | 11.37 | 12.36 | 11.07 | 10.62 | 14.38 |
| Janus-pro-7B | 384 | AR | 44.15 | 16.64 | 9.17 | 7.76 | 5.61 | 5.07 | 14.73 |
| SEED-X | 1024 | MM | 35.03 | 21.86 | 9.07 | 14.25 | 8.68 | 7.15 | 16.01 |
| FLUX.1-[dev] (recaption) | 1024 | DM | 23.40 | 18.99 | 13.00 | 13.83 | 14.76 | 14.31 | 16.38 |
| FLUX.1-[dev] | 1024 | DM | 26.50 | 21.65 | 17.37 | 12.23 | 13.16 | 14.37 | 17.55 |
| FLUX.1-[pro] | 1024 | DM | 43.07 | 30.25 | 23.59 | 21.12 | 23.82 | 24.96 | 27.80 |
| HiDream-I1-Full | 1024 | DM | 41.92 | 28.26 | 28.68 | 27.37 | 21.59 | 28.28 | 29.35 |
| GPT-4o | 1024 | MM | 66.45 | 54.25 | 46.51 | 46.60 | 42.93 | 47.37 | 50.69 |
| FLUX-Reason (o3) | 1024 | DM | 37.54 | 30.37 | 28.98 | 26.91 | 21.38 | 30.35 | 29.25 |
| FLUX-Reason (R1-7B) | 1024 | DM | 42.49 | 31.73 | 32.52 | 28.24 | 25.78 | 32.52 | 32.21 |
| FLUX-Reason (R1) | 1024 | DM | 44.26 | 39.58 | 33.28 | 36.71 | 25.80 | 33.27 | 35.48 |

Table 14: `MMMG-Score` (×100) for **History** across prevalent image generation models. The top three average scores are highlighted in green, blue, and orange.

| Model | Resolution | Type | Preschool | Primary | Secondary | High | Undergrad | PhD | Avg |
|---|---|---|---|---|---|---|---|---|---|
| LlamaGen | 512 | AR | 1.21 | 3.41 | 1.12 | 1.90 | 0.42 | 0.95 | 1.50 |
| Emu-3 | 720 | MM | 4.03 | 7.47 | 3.97 | 5.43 | 3.43 | 2.33 | 4.44 |
| Ideogram V2 | 1024 | DM | 3.96 | 2.15 | 7.03 | 4.91 | 4.92 | 8.77 | 5.29 |
| JanusFlow-1.3B | 384 | AR | 6.85 | 11.81 | 8.82 | 6.88 | 2.85 | 2.51 | 6.62 |
| SimpleAR | 1024 | AR | 6.91 | 10.15 | 7.58 | 7.29 | 5.61 | 4.88 | 7.07 |
| BAGEL | 1024 | MM | 8.50 | 11.82 | 8.25 | 5.61 | 5.80 | 6.80 | 7.80 |
| CogView-4 | 1024 | DM | 6.75 | 8.37 | 9.44 | 10.09 | 6.73 | 7.51 | 8.15 |
| Janus-pro-7B | 384 | AR | 13.08 | 10.68 | 10.97 | 8.06 | 5.09 | 5.90 | 8.96 |
| Infinity-8B | 1024 | AR | 11.09 | 9.82 | 9.48 | 11.35 | 8.04 | 8.61 | 9.73 |
| SDXL-1.0 | 1024 | DM | 3.05 | 13.77 | 13.70 | 15.97 | 7.35 | 6.44 | 10.05 |
| BLIP3-o | 1024 | MM | 14.07 | 12.89 | 12.71 | 10.18 | 6.44 | 6.01 | 10.38 |
| FLUX.1-[dev] (recaption) | 1024 | DM | 10.65 | 11.90 | 13.00 | 8.19 | 9.01 | 9.65 | 10.40 |
| SDXL-1.0-refiner | 1024 | DM | 2.43 | 14.95 | 17.77 | 16.85 | 9.18 | 5.79 | 11.16 |
| FLUX.1-[dev] | 1024 | DM | 9.95 | 15.15 | 15.78 | 12.29 | 9.99 | 12.69 | 12.64 |
| SEED-X | 1024 | MM | 17.78 | 18.90 | 18.80 | 18.21 | 10.28 | 14.27 | 16.37 |
| FLUX.1-[pro] | 1024 | DM | 26.40 | 21.26 | 24.56 | 22.47 | 18.74 | 20.21 | 22.27 |
| HiDream-I1-Full | 1024 | DM | 26.84 | 29.92 | 29.80 | 24.63 | 24.81 | 28.54 | 27.42 |
| GPT-4o | 1024 | MM | 36.84 | 44.38 | 45.62 | 42.76 | 37.76 | 42.59 | 41.66 |
| FLUX-Reason (o3) | 1024 | DM | 16.21 | 26.69 | 28.87 | 22.72 | 26.05 | 26.58 | 24.52 |
| FLUX-Reason (R1-7B) | 1024 | DM | 28.65 | 28.29 | 29.51 | 26.22 | 25.55 | 27.57 | 27.63 |
| FLUX-Reason (R1) | 1024 | DM | 28.45 | 34.43 | 38.21 | 34.17 | 30.42 | 31.39 | 32.84 |

Table 15: `MMMG-Score` (×100) for **Philosophy** across prevalent image generation models. Scores are reported over five educational stages (no Preschool data). The top three average scores are highlighted in green, blue, and orange.

| Model | Resolution | Type | Primary | Secondary | High | Undergrad | PhD | Avg |
|---|---|---|---|---|---|---|---|---|
| LlamaGen | 512 | AR | 5.23 | 0.95 | 0.81 | 0.77 | 1.8 | 1.91 |
| Emu-3 | 720 | MM | 6.89 | 2.83 | 2.51 | 0.96 | 4.14 | 3.47 |
| JanusFlow-1.3B | 384 | AR | 14.63 | 3.06 | 1.19 | 3.48 | 4.89 | 5.45 |
| SimpleAR | 1024 | AR | 11.52 | 3.26 | 3.88 | 3.35 | 8.16 | 6.03 |
| BLIP3-o | 1024 | MM | 20.79 | 4.91 | 2.11 | 2.62 | 8.12 | 7.71 |
| BAGEL | 1024 | MM | 16.51 | 7.9 | 3.84 | 2.51 | 8.69 | 7.89 |
| Janus-pro-7B | 384 | AR | 18.42 | 5.33 | 4.4 | 5.35 | 7.96 | 8.29 |
| SDXL-1.0 | 1024 | DM | 18.35 | 8.01 | 6.4 | 6.63 | 8.0 | 9.48 |
| SEED-X | 1024 | MM | 20.49 | 8.63 | 6.36 | 4.95 | 7.98 | 9.68 |
| SDXL-1.0-refiner | 1024 | DM | 20.21 | 8.68 | 7.82 | 7.46 | 5.87 | 10.01 |
| CogView-4 | 1024 | DM | 23.83 | 10.28 | 5.98 | 5.49 | 9.08 | 10.93 |
| FLUX.1-[dev] (recaption) | 1024 | DM | 23.55 | 15.98 | 14.44 | 13.33 | 9.47 | 15.35 |
| Ideogram V2 | 1024 | DM | 23.0 | 17.64 | 14.31 | 15.63 | 14.09 | 16.93 |
| Infinity-8B | 1024 | AR | 21.7 | 17.86 | 15.12 | 14.1 | 18.93 | 17.54 |
| FLUX.1-[dev] | 1024 | DM | 29.05 | 18.28 | 15.98 | 12.37 | 19.23 | 18.98 |
| FLUX.1-[pro] | 1024 | DM | 36.72 | 29.32 | 26.34 | 22.16 | 26.02 | 28.11 |
| HiDream-I1-Full | 1024 | DM | 40.39 | 30.78 | 27.82 | 19.4 | 28.24 | 29.33 |
| GPT-4o | 1024 | MM | 59.58 | 48.55 | 45.21 | 42.35 | 45.03 | 48.14 |
| FLUX-Reason (o3) | 1024 | DM | 33.48 | 22.78 | 27.33 | 21.1 | 25.76 | 26.09 |
| FLUX-Reason (R1-7B) | 1024 | DM | 41.57 | 34.6 | 30.02 | 26.55 | 30.47 | 32.64 |
| FLUX-Reason (R1) | 1024 | DM | 41.5 | 36.08 | 30.45 | 26.04 | 27.84 | 32.38 |

# E   Ablation Study

## E.1   Self-Consistency of MMMG–Eval

A truly reliable evaluation metric must exhibit self-consistency: when applied to ground-truth images, `MMMG` should recover the original knowledge graph with minimal error. We assess this property by running OpenAI-o3 as the evaluator and compute two complementary scores on all reference images: the knowledge fidelity term

$$1 - \text{GED}(G_{\text{gen}}, G_{\text{ref}})$$

and the unified accuracy

$$\text{u-acc} = \frac{N_e^{\text{correct}} + N_d^{\text{correct}}}{N_e + N_d},$$

where $N_e$, $N_d$ are the total number of entities and dependencies, and $N_e^{\text{correct}}$, $N_d^{\text{correct}}$ the correctly recovered counts. Table 16 reports these metrics across six educational stages. We observe consistently high fidelity ($> 0.92$) and accuracy ($> 0.91$) without any stage-specific tuning, confirming that `MMMG` faithfully grounds and retrieves structured knowledge from its source images.

|  | Preschool | Primary | Secondary | High School | Undergrad | PhD |
|---|---|---|---|---|---|---|
| $1 - \text{GED}$ | 0.9373 | 0.9336 | 0.9318 | 0.9307 | 0.9287 | 0.9360 |
| u-acc | 0.9261 | 0.9218 | 0.9154 | 0.9110 | 0.9154 | 0.9240 |

Table 16: Self-consistency evaluation: structural fidelity ($1-$GED) and unified accuracy (u-acc) of `MMMG` on ground-truth images across different educational stages.

To directly assess the accuracy and reliability of the evaluator (OpenAI-o3), we conducted a human vs. model comparison study. We evenly sampled 480 images across all educational levels and had domain experts annotate their corresponding knowledge graphs. As shown in Table 17, the evaluator demonstrates high alignment with human annotations, with an average recall exceeding 94% across models, indicating that missed detections and false penalties are rare.

| Model | P (%) | R (%) |
|---|---|---|
| GPT-4o | 96.52 | 96.48 |
| FLUX-Reason | 94.27 | 94.13 |
| SDXL | 92.74 | 94.70 |

Table 17: Average Precision (P) and Recall (R) between OpenAI-o3 and expert evaluators.

## E.2   Robustness of MMMG Evaluation

In addition to self-consistency, a reliable evaluation metric should be robust to changes in its underlying judgment engine. To assess this, we recompute the `MMMG-Score` using three variants of OpenAI's language models—OpenAI-o3 (high reasoning effort), OpenAI-o3, and OpenAI-o1 (lightweight)—on the same set of generated images. As shown in Table 18, the relative rankings of generation models (GPT-4o-Image, FLUX-Reason (R1), FLUX.1-[pro], Infinity-8B, SEED-X) remain consistent across all evaluator variants, with only minor score fluctuations. This stability indicates that `MMMG-Score` is robust to variations in the choice of LLM.

| Evaluator | GPT-4o | FLUX-Reason (R1) | FLUX.1-[pro] | Infinity-8B | SEED-X |
|---|---|---|---|---|---|
| OpenAI-o3-high | 50.20 | 34.45 | 27.14 | 19.18 | 18.16 |
| OpenAI-o3 | 51.07 | 34.62 | 27.33 | 19.54 | 18.49 |
| OpenAI-o1 | 52.81 | 37.25 | 29.52 | 21.73 | 20.86 |

Table 18: Robustness evaluation: `MMMG-Score` (higher is better) recomputed with different OpenAI LLM variants. The preserved model ranking and low score variance confirm stability against evaluator changes.

# F    Visualization

## F.1    Preschool

### F.1.1    Biology

Question: Visualize an educational diagram illustrating how heat influences the water cycle through transpiration and surface water interactions.

Figure 3: `MMMG` Benchmark visualization for seven representative models on a Preschool-Biology example. Each row corresponds to one model and, from left to right, displays the generated image, its segmentation map, the reconstructed knowledge graph, the extracted entity and dependency lists, and finally the overall `MMMG-Score` along with its component sub-scores.

### F.1.2 Chemistry

Question: Design a playful chart about how bubbles are formed in different liquids, incorporating vibrant colors and fun shapes.

Figure 4: `MMMG` Benchmark visualization for seven representative models on a Preschool-Chemistry example. Each row corresponds to one model and, from left to right, displays the generated image, its segmentation map, the reconstructed knowledge graph, the extracted entity and dependency lists, and finally the overall `MMMG-Score` along with its component sub-scores.

## F.1.3 Mathematics

Question: Create a fun calendar featuring the days of the week, using different colors and cartoon animals to represent each day.

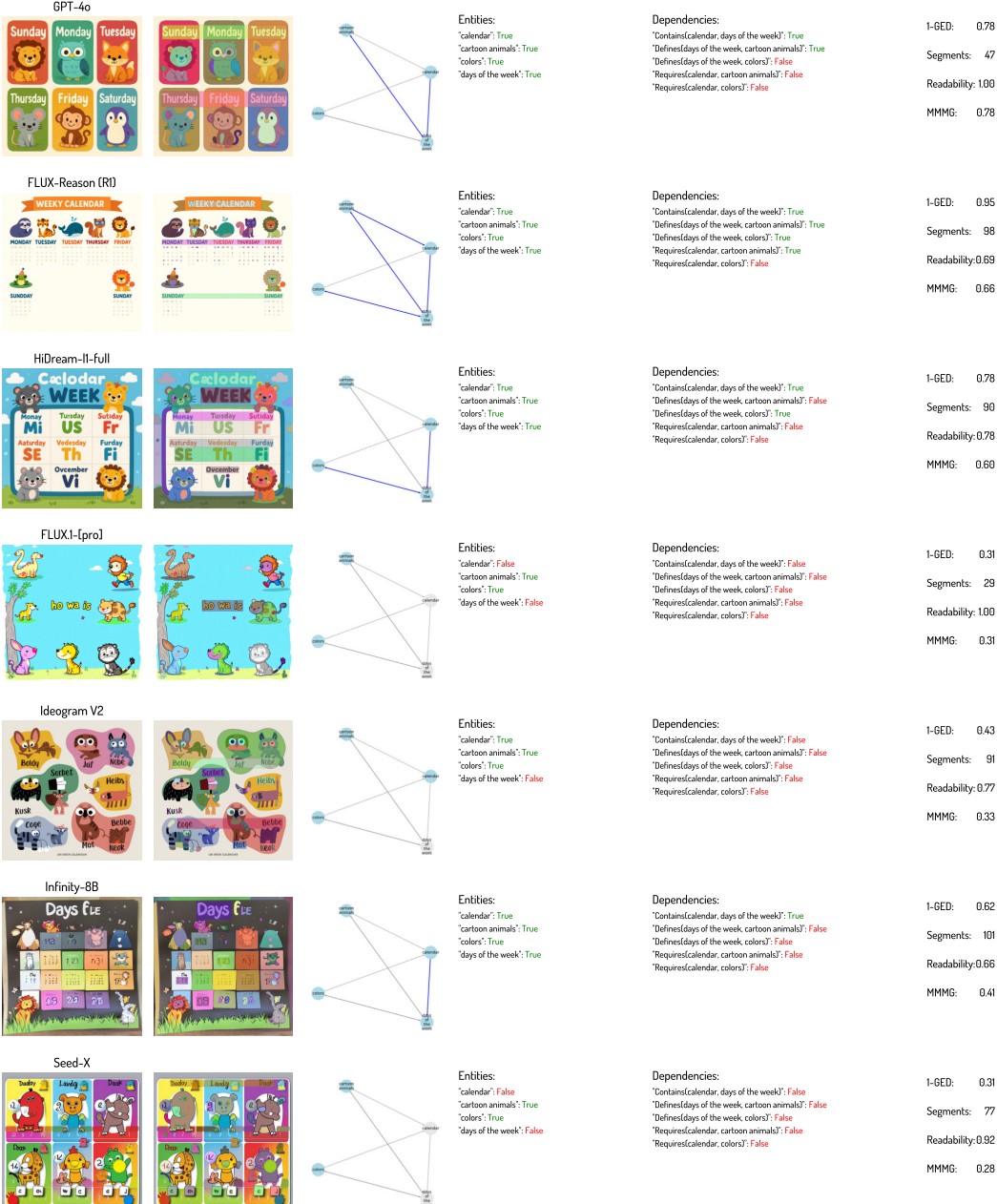

Figure 5: `MMMG` Benchmark visualization for seven representative models on a Preschool-Mathematics example. Each row corresponds to one model and, from left to right, displays the generated image, its segmentation map, the reconstructed knowledge graph, the extracted entity and dependency lists, and finally the overall `MMMG-Score` along with its component sub-scores.

## F.1.4 Engineering

Question: Visualize an educational poster illustrating the steps to create a paper shirt through folding techniques.

Figure 6: `MMMG` Benchmark visualization for seven representative models on a Preschool-Engineering example. Each row corresponds to one model and, from left to right, displays the generated image, its segmentation map, the reconstructed knowledge graph, the extracted entity and dependency lists, and finally the overall `MMMG-Score` along with its component sub-scores.

## F.1.5    Geography

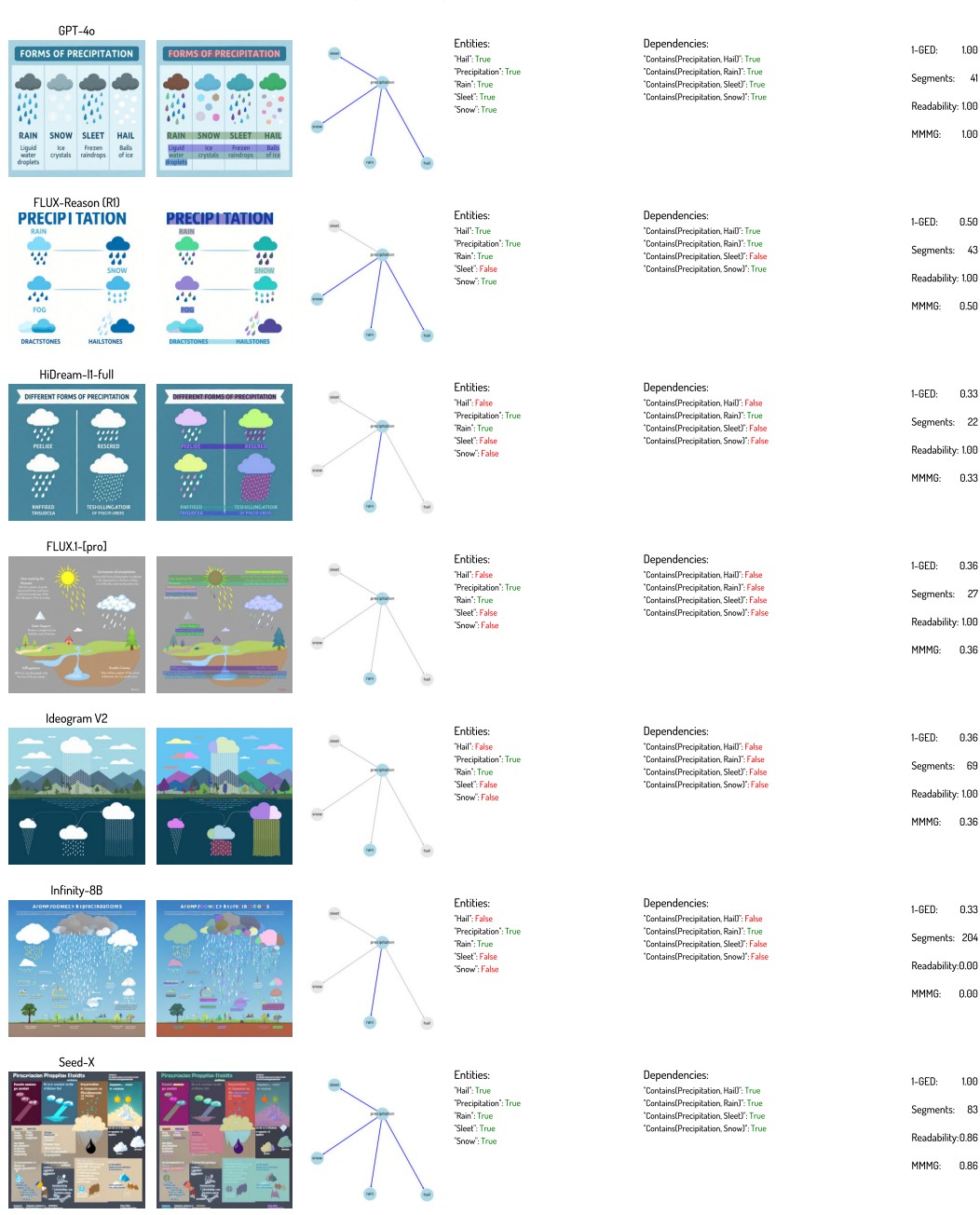

Figure 7: `MMMG` Benchmark visualization for seven representative models on a Preschool-Geography example. Each row corresponds to one model and, from left to right, displays the generated image, its segmentation map, the reconstructed knowledge graph, the extracted entity and dependency lists, and finally the overall `MMMG-Score` along with its component sub-scores.

## F.1.6 Economics

Question: Visualize the key elements and dynamics of a grocery shopping experience in an informative infographic.

Figure 8: `MMMG` Benchmark visualization for seven representative models on a Preschool-Economics example. Each row corresponds to one model and, from left to right, displays the generated image, its segmentation map, the reconstructed knowledge graph, the extracted entity and dependency lists, and finally the overall `MMMG-Score` along with its component sub-scores.

## F.1.7 Sociology

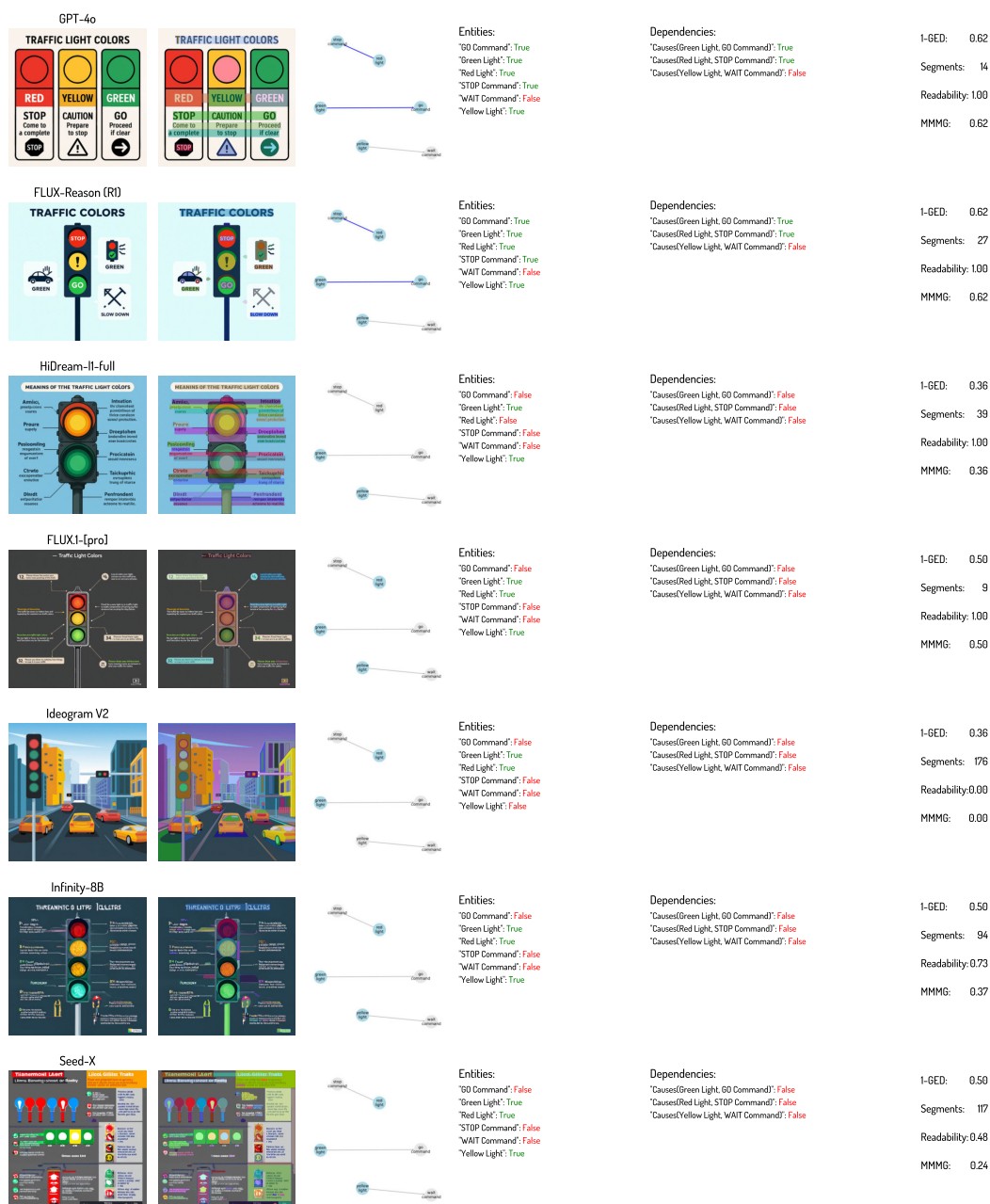

Figure 9: `MMMG` Benchmark visualization for seven representative models on a Preschool-Sociology example. Each row corresponds to one model and, from left to right, displays the generated image, its segmentation map, the reconstructed knowledge graph, the extracted entity and dependency lists, and finally the overall `MMMG-Score` along with its component sub-scores.

## F.1.8 History

Question: Visualize an educational infographic showcasing the key structural components of the Great Wall of China and their functions.

Figure 10: `MMMG` Benchmark visualization for seven representative models on a Preschool-History example. Each row corresponds to one model and, from left to right, displays the generated image, its segmentation map, the reconstructed knowledge graph, the extracted entity and dependency lists, and finally the overall `MMMG-Score` along with its component sub-scores.

## F.1.9 Literature

Question: Visualize an infographic of a treasure chest overflowing with various genres of children's books.

Figure 11: `MMMG` Benchmark visualization for seven representative models on a Preschool-Literature example. Each row corresponds to one model and, from left to right, displays the generated image, its segmentation map, the reconstructed knowledge graph, the extracted entity and dependency lists, and finally the overall `MMMG-Score` along with its component sub-scores.

## F.2 Primary School

### F.2.1 Biology

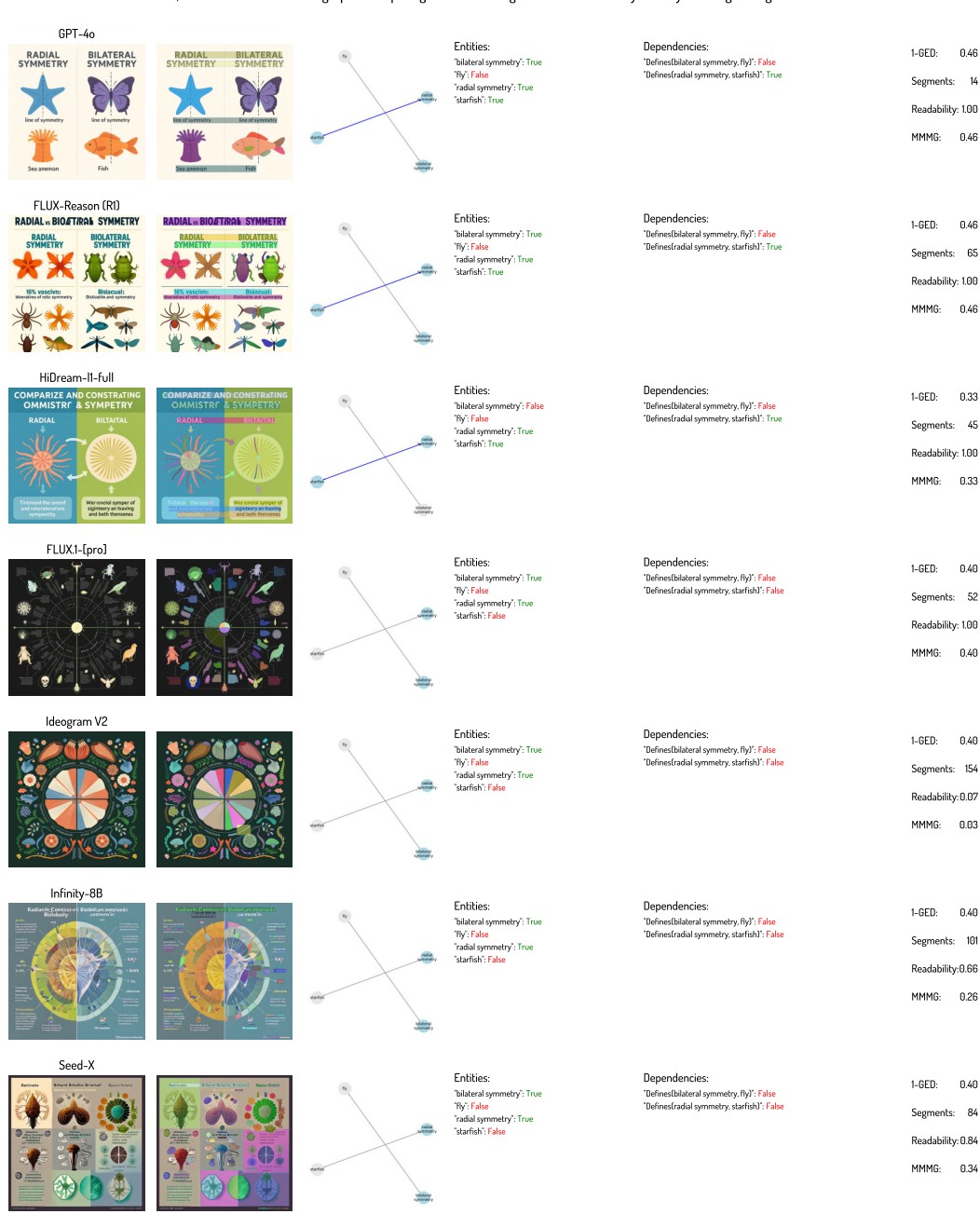

Figure 12: MMMG Benchmark visualization for seven representative models on a Primaryschool-Biology example. Each row corresponds to one model and, from left to right, displays the generated image, its segmentation map, the reconstructed knowledge graph, the extracted entity and dependency lists, and finally the overall MMMG-Score along with its component sub-scores.

## F.2.2 Chemistry

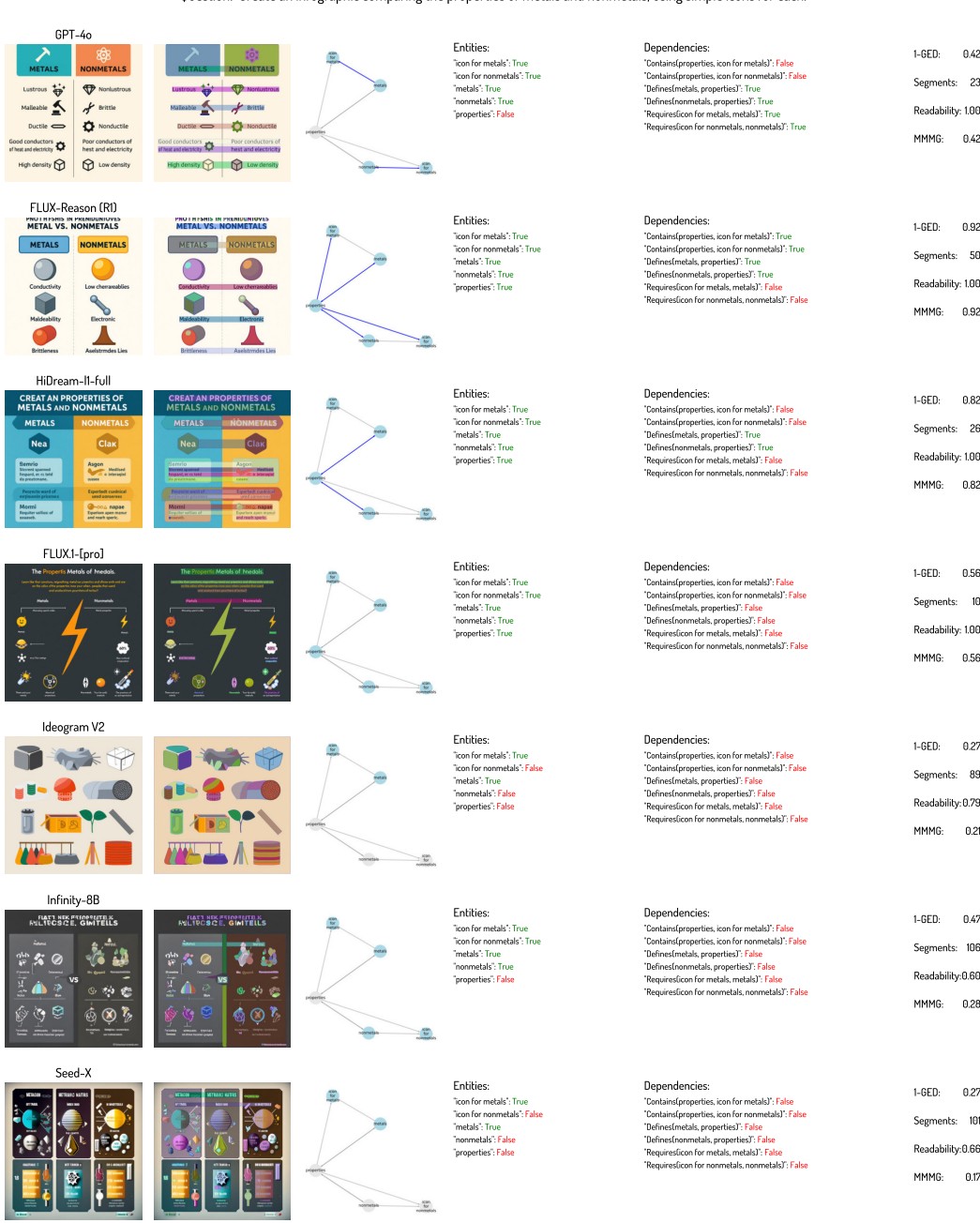

Figure 13: `MMMG` Benchmark visualization for seven representative models on a Primaryschool-Chemistry example. Each row corresponds to one model and, from left to right, displays the generated image, its segmentation map, the reconstructed knowledge graph, the extracted entity and dependency lists, and finally the overall `MMMG-Score` along with its component sub-scores.

### F.2.3 Mathematics

Question: Visualize the use of playdough and toothpicks to create and understand 3D shapes.

Figure 14: `MMMG` Benchmark visualization for seven representative models on a Primaryschool-Mathematics example. Each row corresponds to one model and, from left to right, displays the generated image, its segmentation map, the reconstructed knowledge graph, the extracted entity and dependency lists, and finally the overall `MMMG-Score` along with its component sub-scores.

## F.2.4 Engineering

Question: Visualize an educational poster demonstrating the interaction between permanent magnets and a steel bar, highlighting forces, alignments, and magnetic field representations.

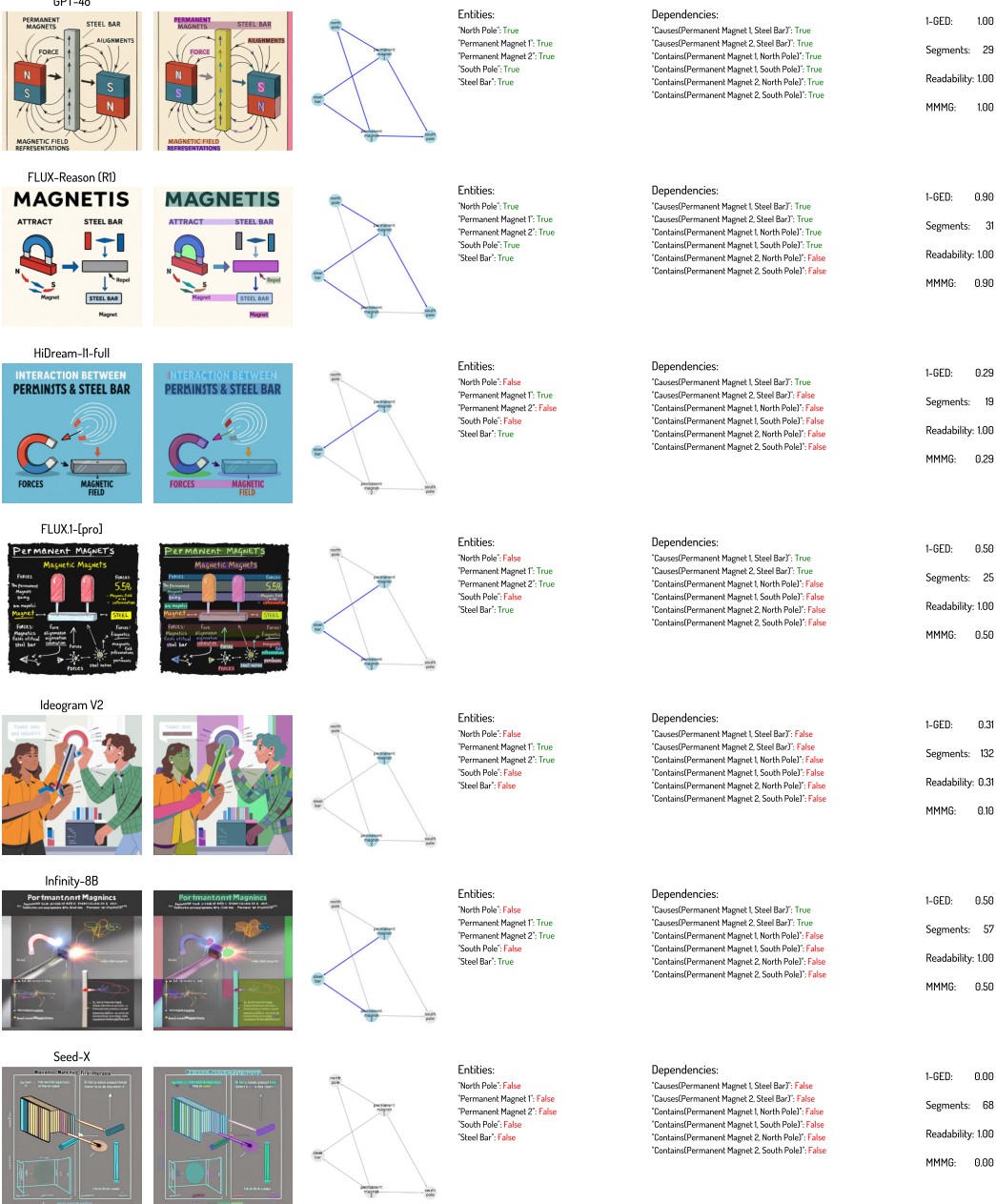

Figure 15: MMMG Benchmark visualization for seven representative models on a Primaryschool-Engineering example. Each row corresponds to one model and, from left to right, displays the generated image, its segmentation map, the reconstructed knowledge graph, the extracted entity and dependency lists, and finally the overall MMMG-Score along with its component sub-scores.

## F.2.5 Geography

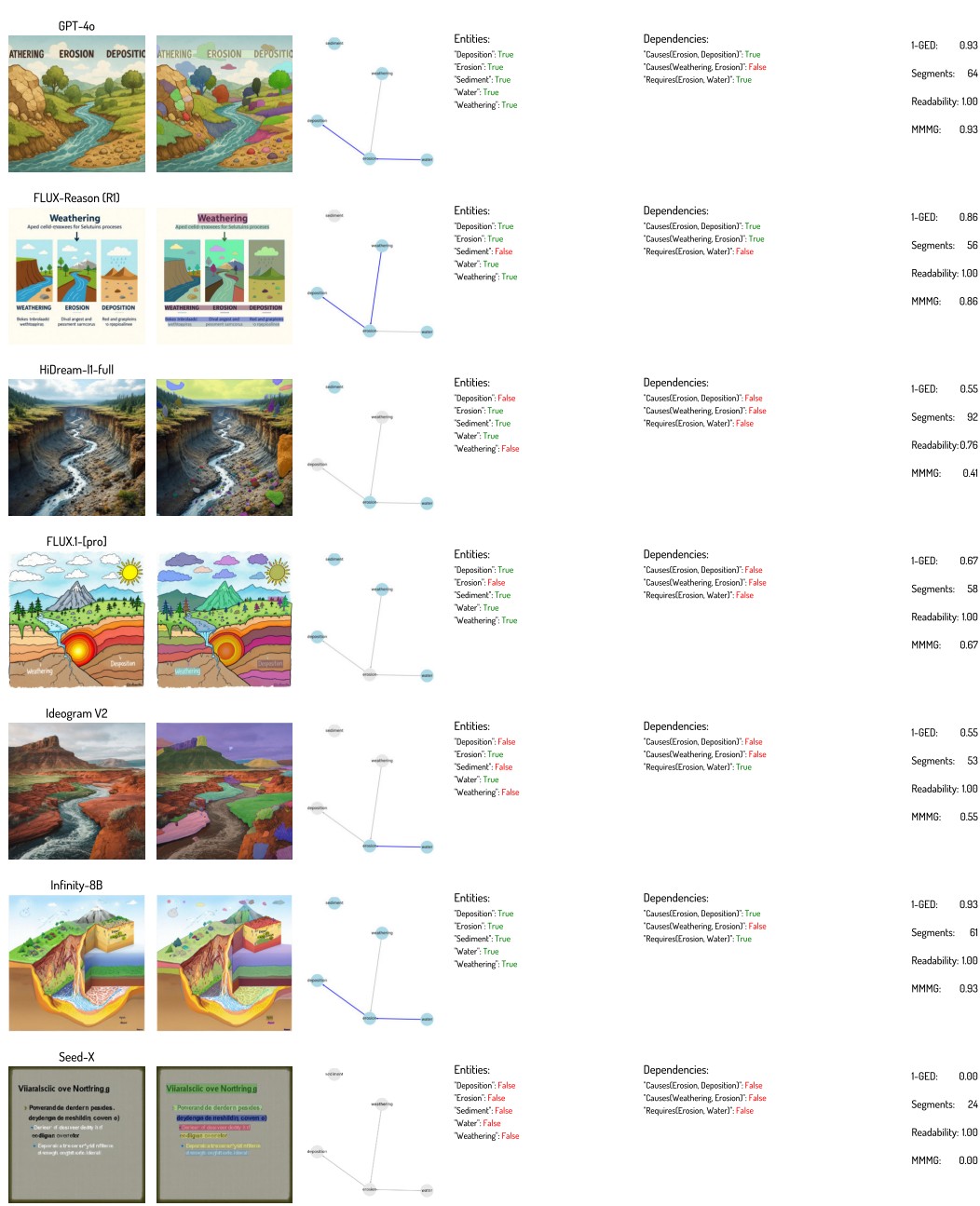

Figure 16: `MMMG` Benchmark visualization for seven representative models on a Primaryschool-Geography example. Each row corresponds to one model and, from left to right, displays the generated image, its segmentation map, the reconstructed knowledge graph, the extracted entity and dependency lists, and finally the overall `MMMG-Score` along with its component sub-scores.

## F.2.6 Economics

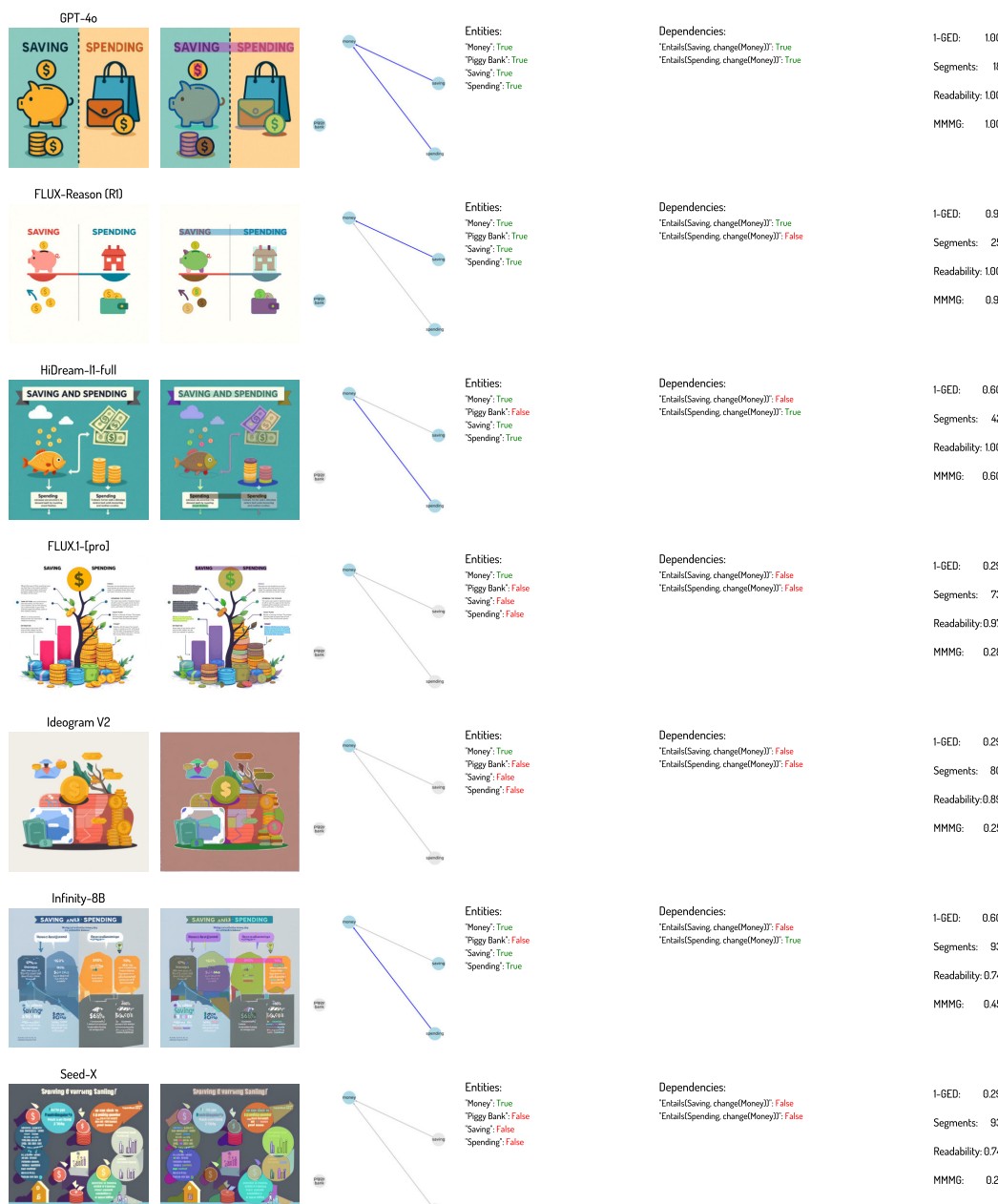

Figure 17: `MMMG` Benchmark visualization for seven representative models on a Primaryschool-Economics example. Each row corresponds to one model and, from left to right, displays the generated image, its segmentation map, the reconstructed knowledge graph, the extracted entity and dependency lists, and finally the overall `MMMG-Score` along with its component sub-scores.

### F.2.7 Sociology

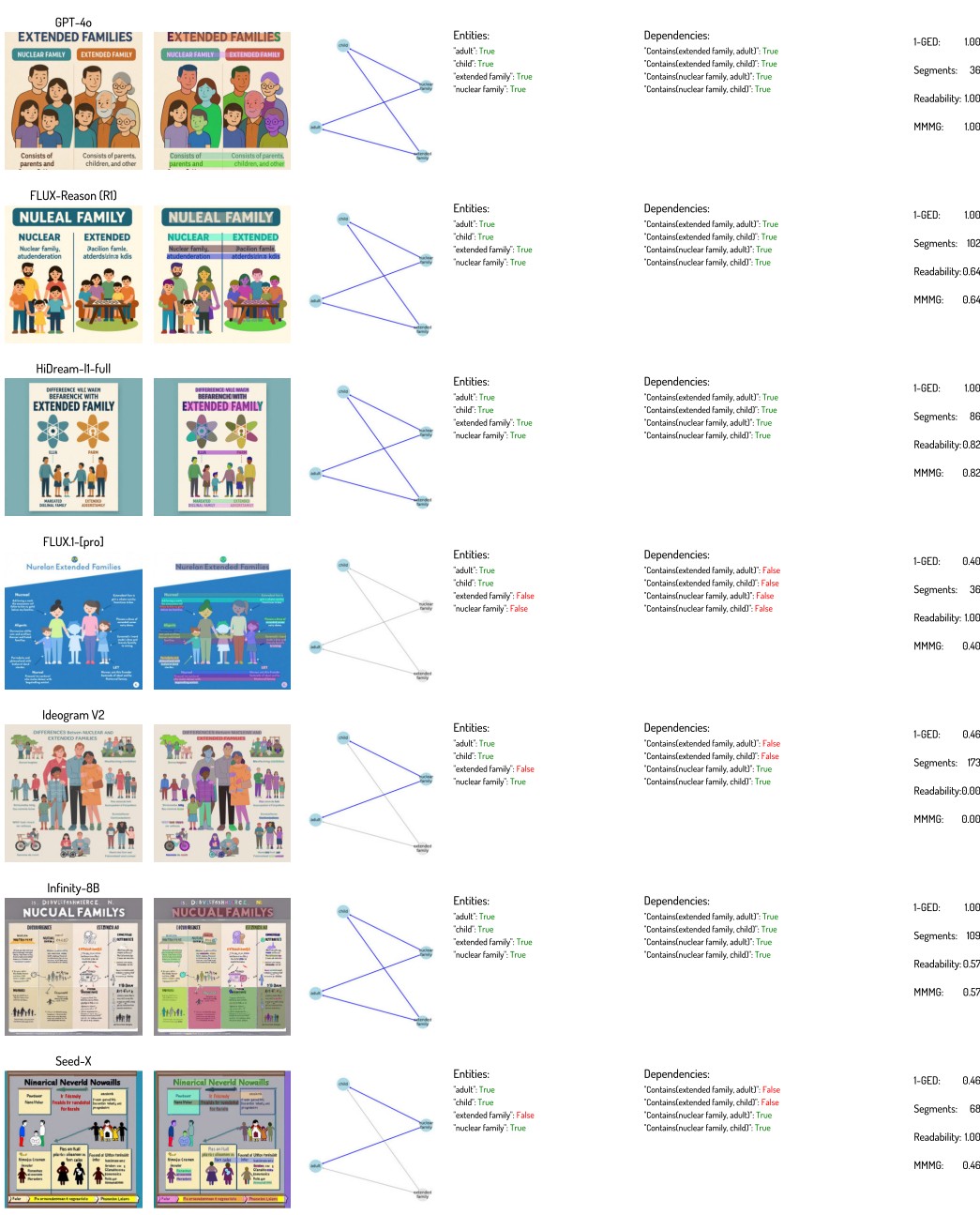

Figure 18: `MMMG` Benchmark visualization for seven representative models on a Primaryschool-Sociology example. Each row corresponds to one model and, from left to right, displays the generated image, its segmentation map, the reconstructed knowledge graph, the extracted entity and dependency lists, and finally the overall `MMMG-Score` along with its component sub-scores.

## F.2.8 History

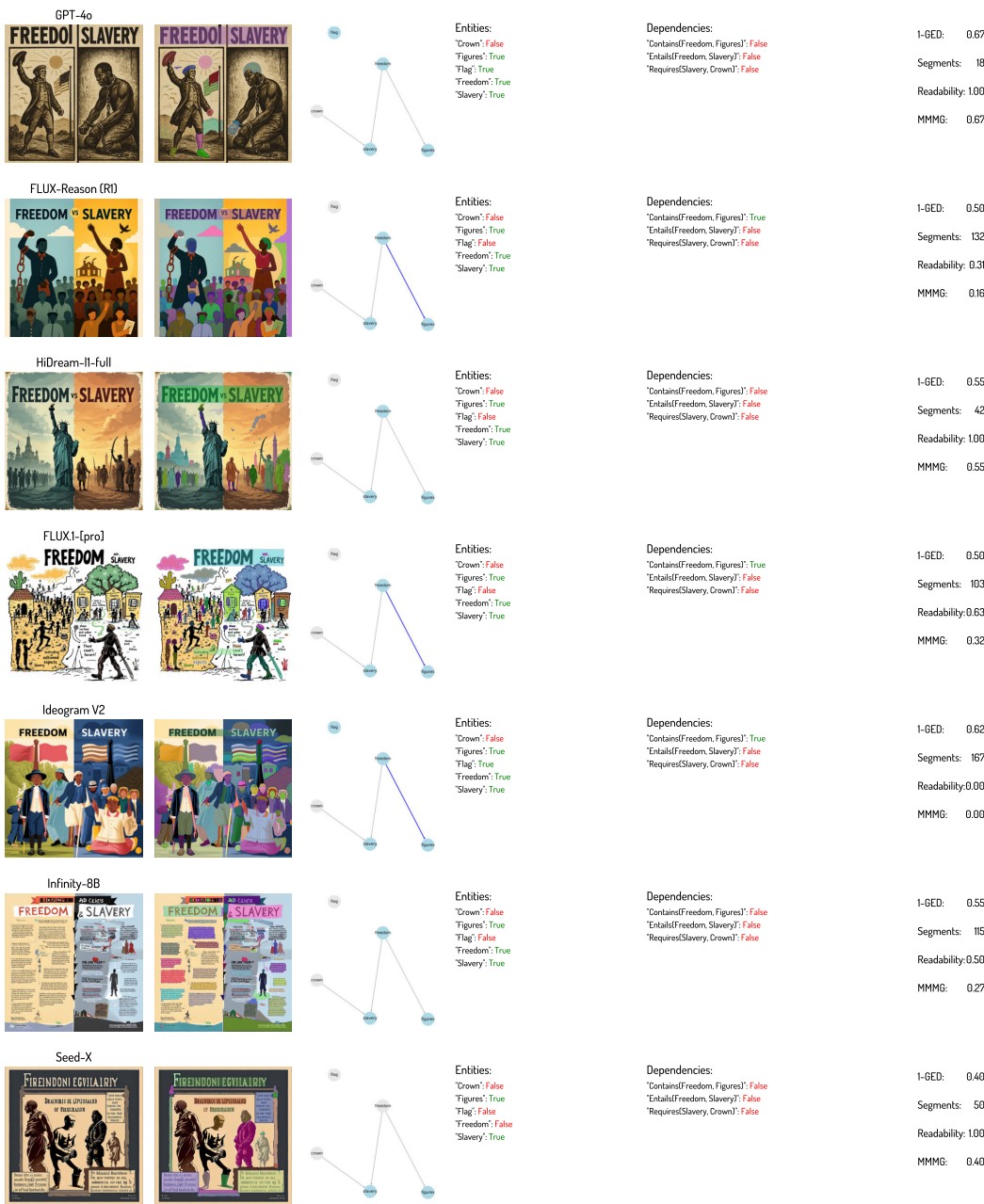

Figure 19: `MMMG` Benchmark visualization for seven representative models on a Primaryschool-History example. Each row corresponds to one model and, from left to right, displays the generated image, its segmentation map, the reconstructed knowledge graph, the extracted entity and dependency lists, and finally the overall `MMMG-Score` along with its component sub-scores.

## F.2.9 Philosophy

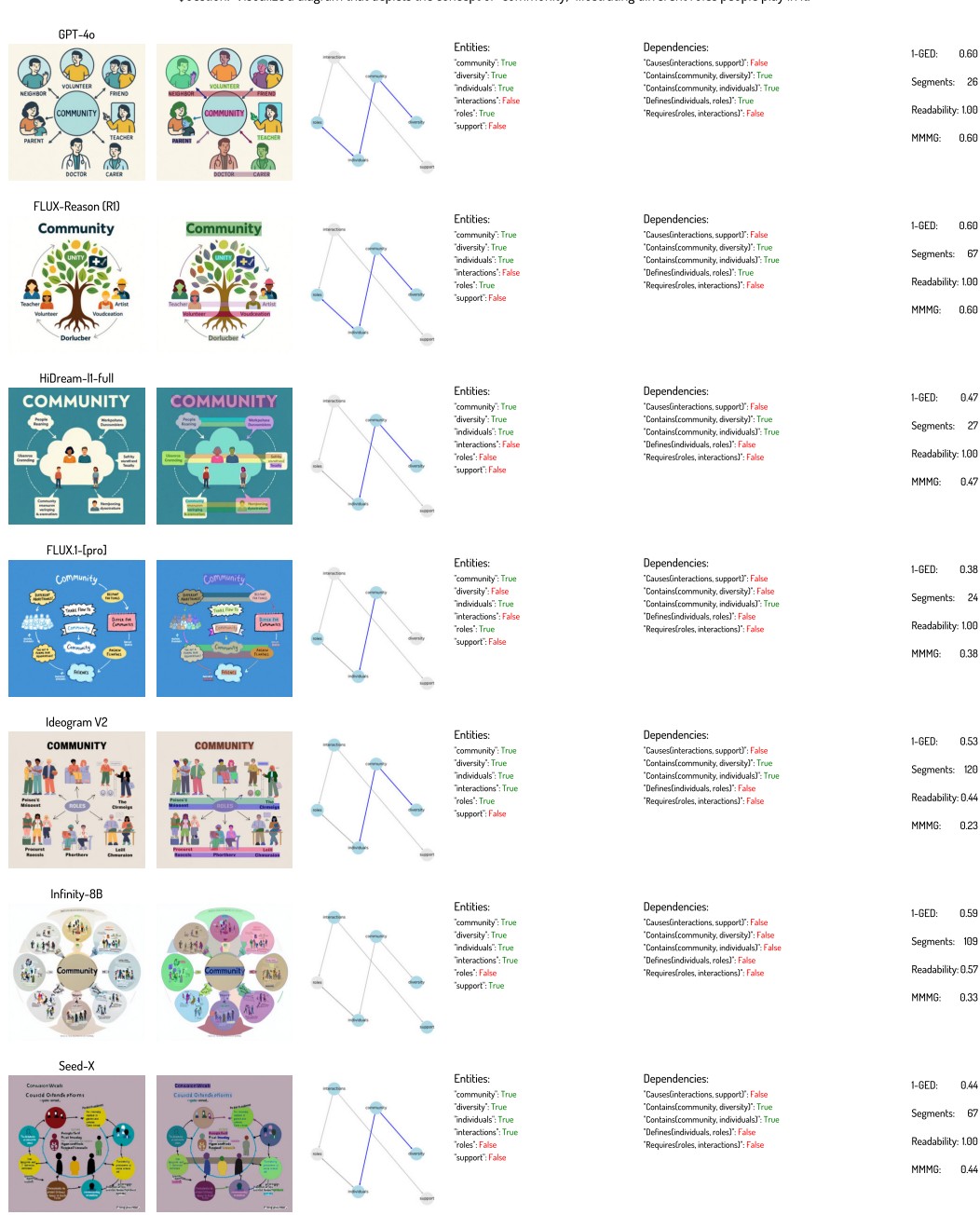

Figure 20: `MMMG` Benchmark visualization for seven representative models on a Primaryschool-Philosophy example. Each row corresponds to one model and, from left to right, displays the generated image, its segmentation map, the reconstructed knowledge graph, the extracted entity and dependency lists, and finally the overall `MMMG-Score` along with its component sub-scores.

## F.2.10 Literature

Question: Design a poster that teaches kids how to write a simple story, incorporating steps and illustrations.

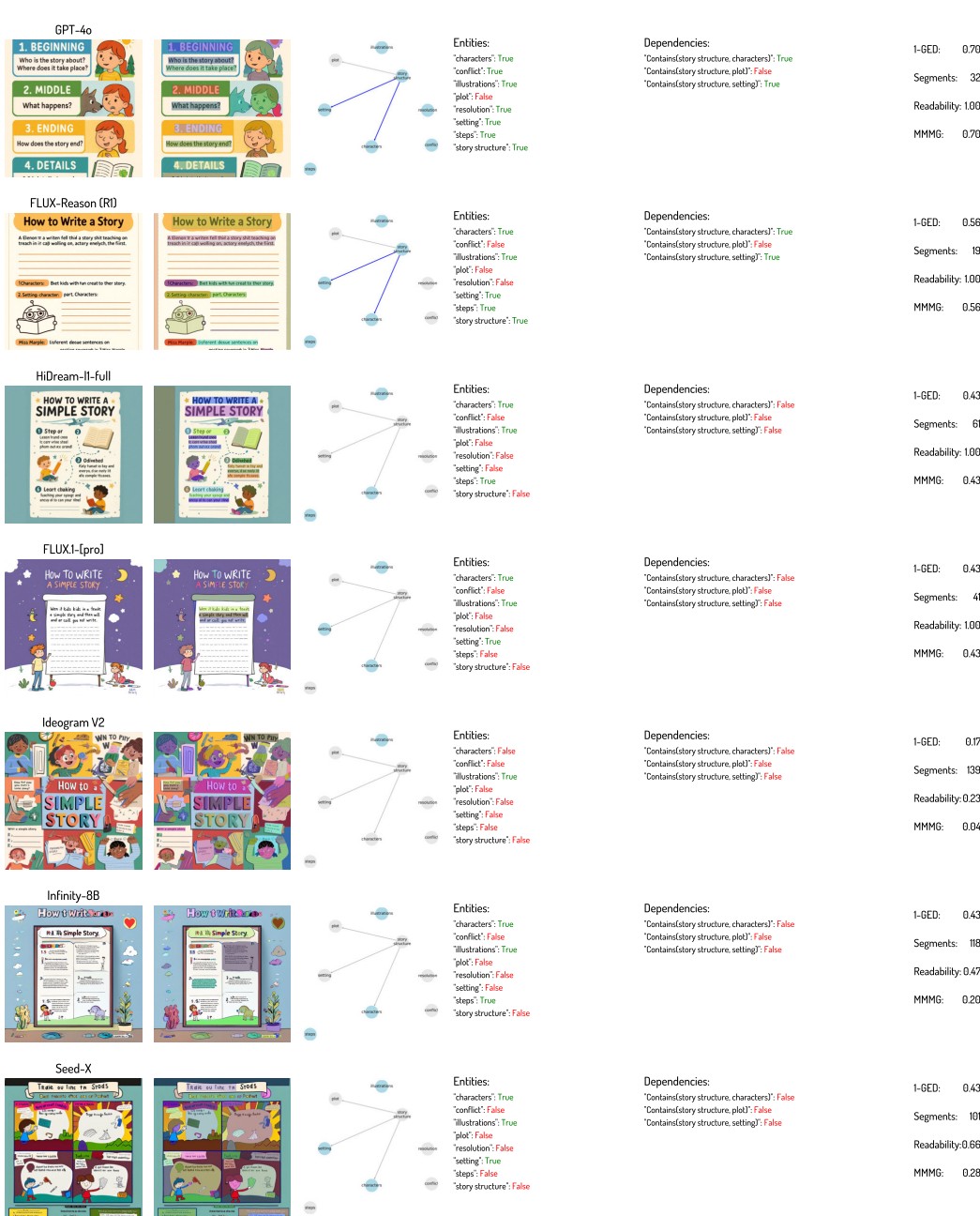

Figure 21: `MMMG` Benchmark visualization for seven representative models on a Primaryschool-Literature example. Each row corresponds to one model and, from left to right, displays the generated image, its segmentation map, the reconstructed knowledge graph, the extracted entity and dependency lists, and finally the overall `MMMG-Score` along with its component sub-scores.

## F.3 Secondary School

### F.3.1 Biology

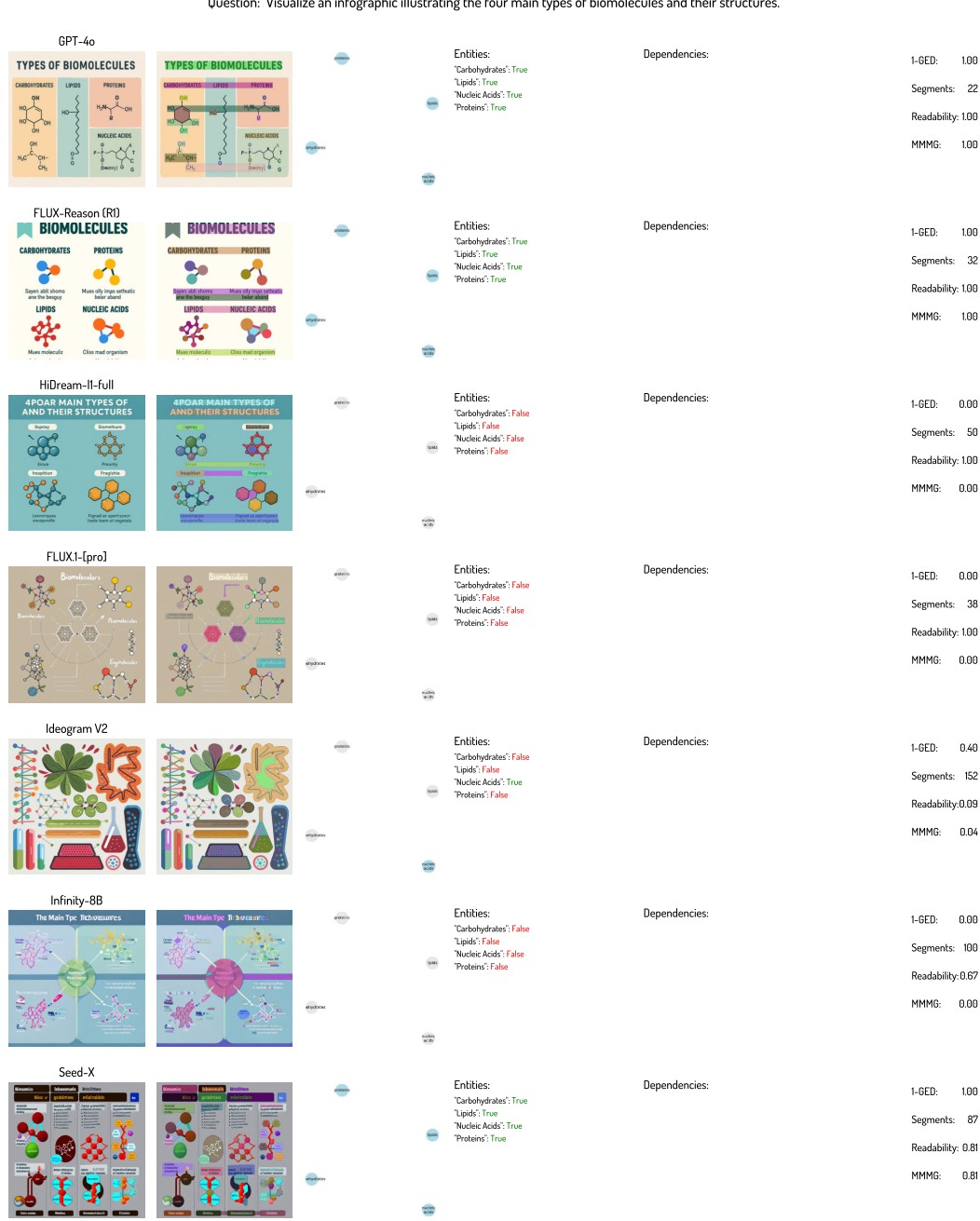

Figure 22: `MMMG` Benchmark visualization for seven representative models on a Secondaryschool-Biology example. Each row corresponds to one model and, from left to right, displays the generated image, its segmentation map, the reconstructed knowledge graph, the extracted entity and dependency lists, and finally the overall `MMMG-Score` along with its component sub-scores.

## F.3.2 Chemistry

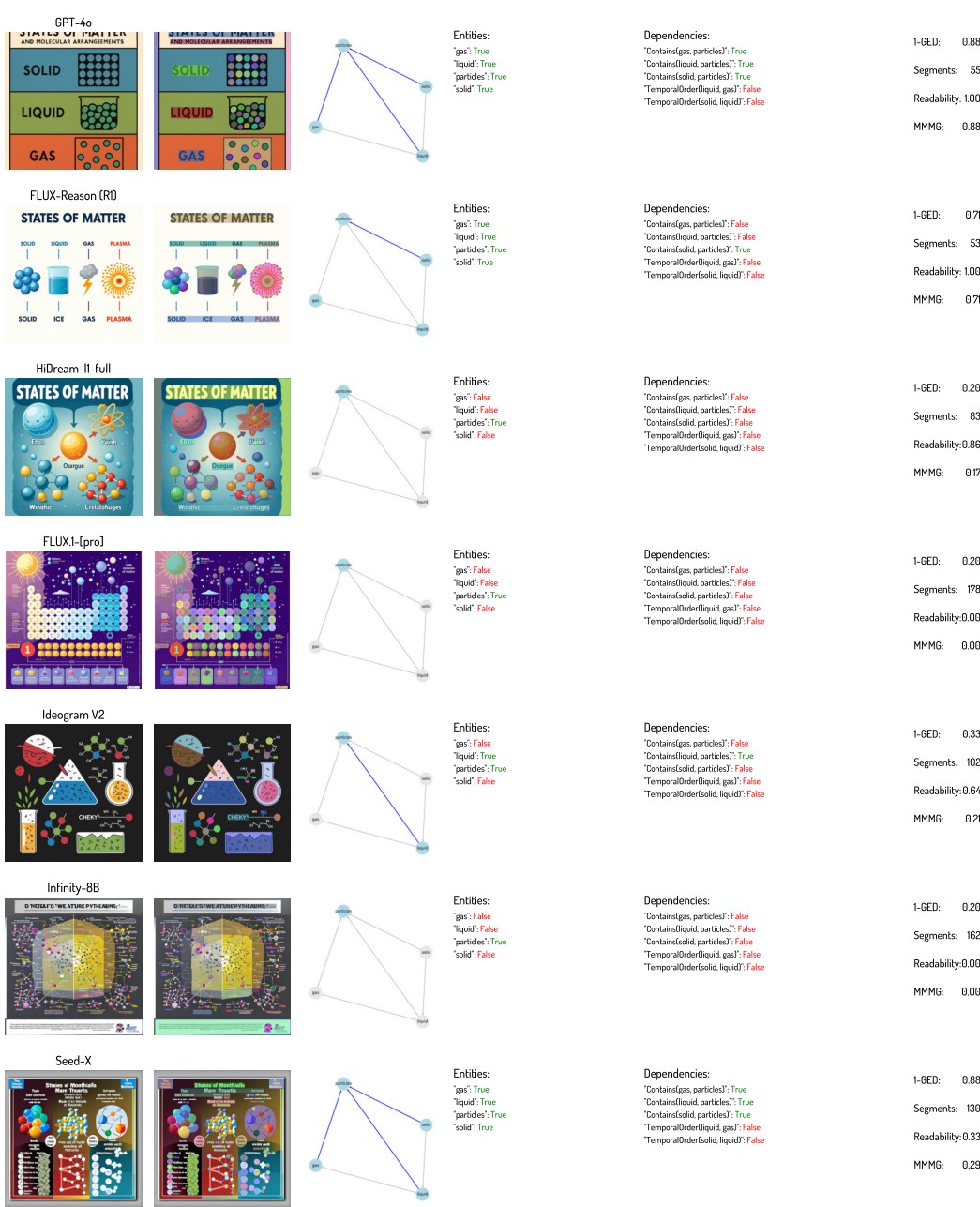

Figure 23: `MMMG` Benchmark visualization for seven representative models on a Secondaryschool-Chemistry example. Each row corresponds to one model and, from left to right, displays the generated image, its segmentation map, the reconstructed knowledge graph, the extracted entity and dependency lists, and finally the overall `MMMG-Score` along with its component sub-scores.

### F.3.3 Mathematics

Question: Create a diagram explaining the difference between rational and irrational numbers, including examples and visual representations.

Figure 24: `MMMG` Benchmark visualization for seven representative models on a Secondaryschool-Mathematics example. Each row corresponds to one model and, from left to right, displays the generated image, its segmentation map, the reconstructed knowledge graph, the extracted entity and dependency lists, and finally the overall `MMMG-Score` along with its component sub-scores.

## F.3.4 Engineering

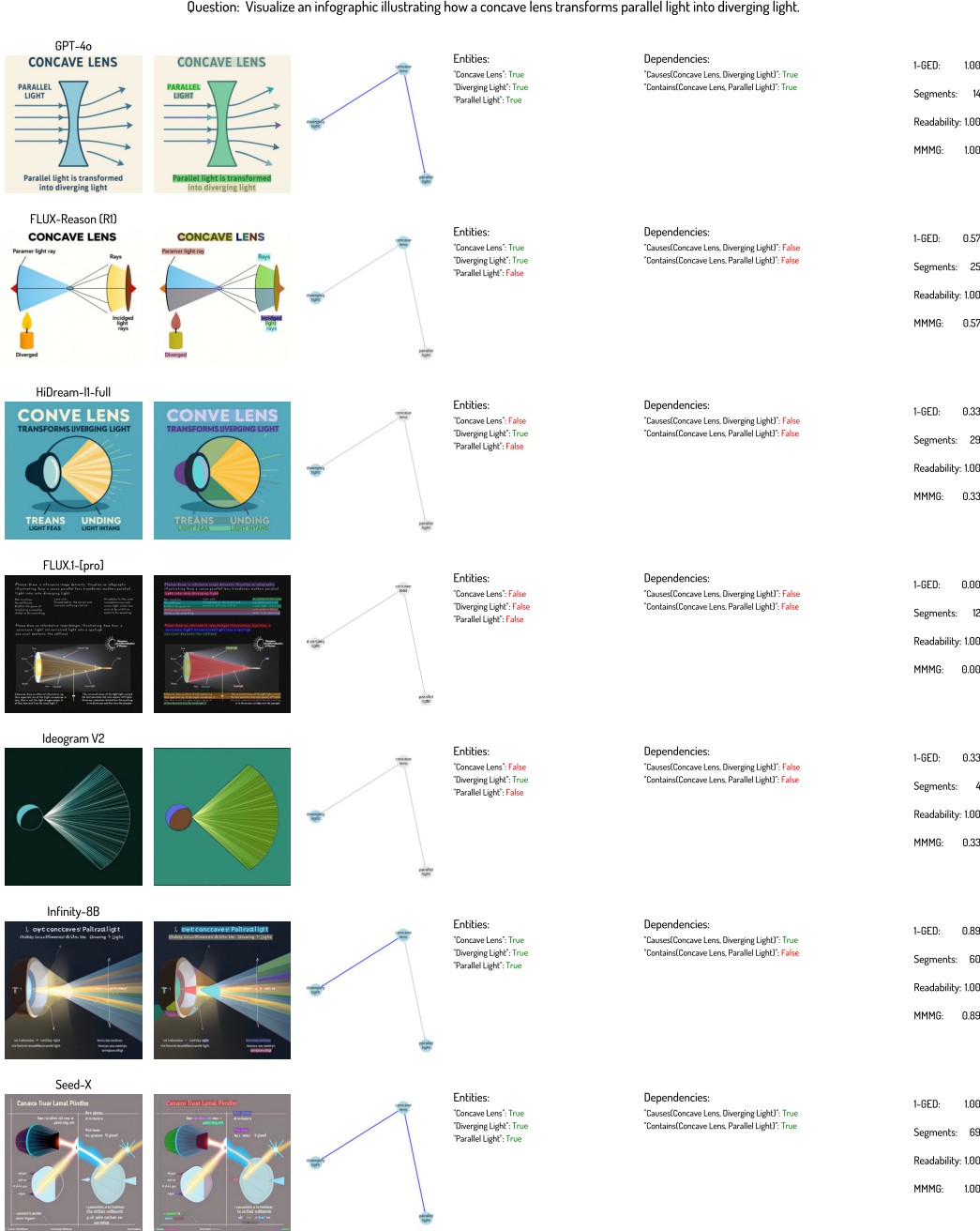

Figure 25: MMMG Benchmark visualization for seven representative models on a Secondaryschool-Engineering example. Each row corresponds to one model and, from left to right, displays the generated image, its segmentation map, the reconstructed knowledge graph, the extracted entity and dependency lists, and finally the overall MMMG-Score along with its component sub-scores.

## F.3.5 Geography

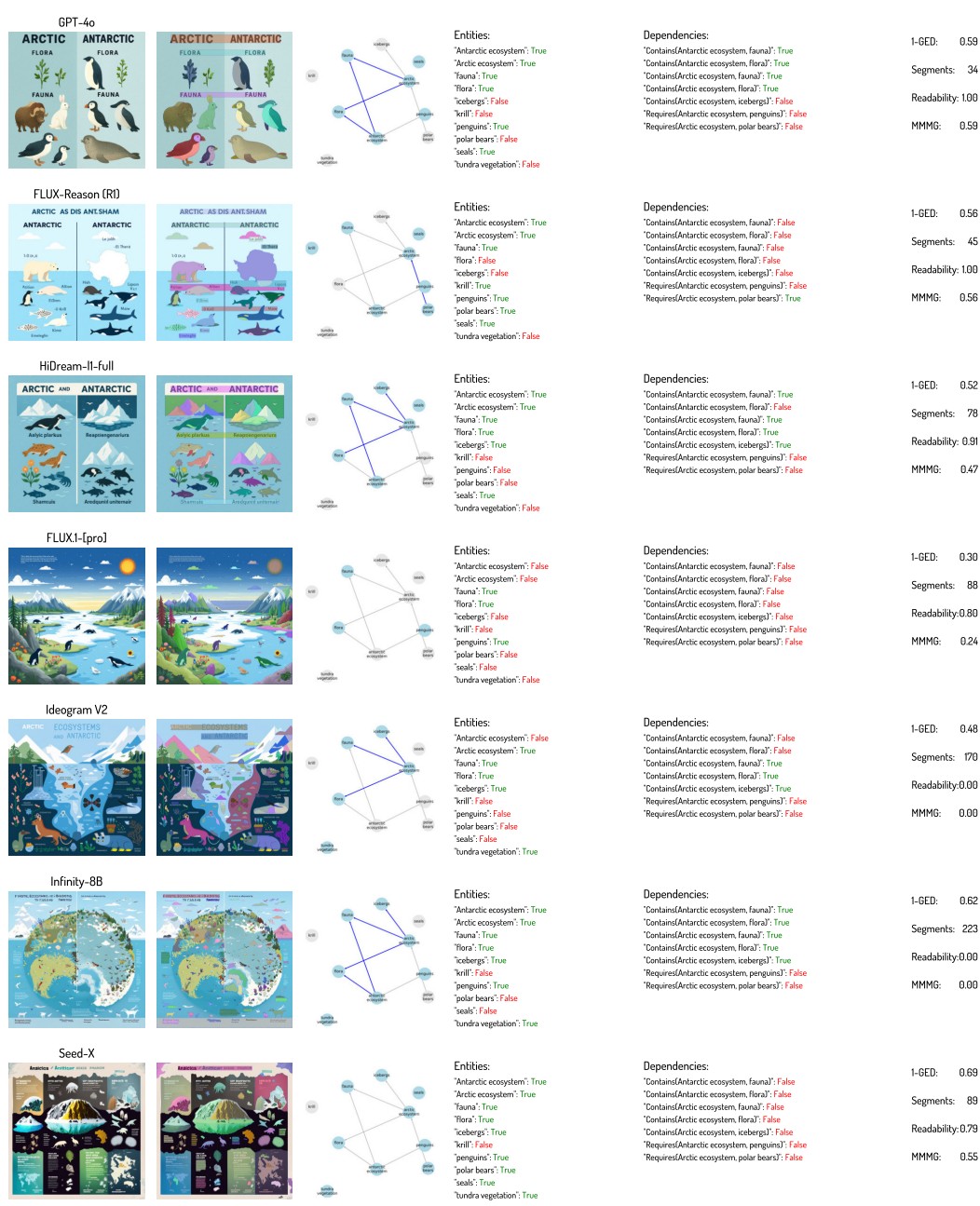

Figure 26: `MMMG` Benchmark visualization for seven representative models on a Secondaryschool-Geography example. Each row corresponds to one model and, from left to right, displays the generated image, its segmentation map, the reconstructed knowledge graph, the extracted entity and dependency lists, and finally the overall `MMMG-Score` along with its component sub-scores.

## F.3.6 Economics

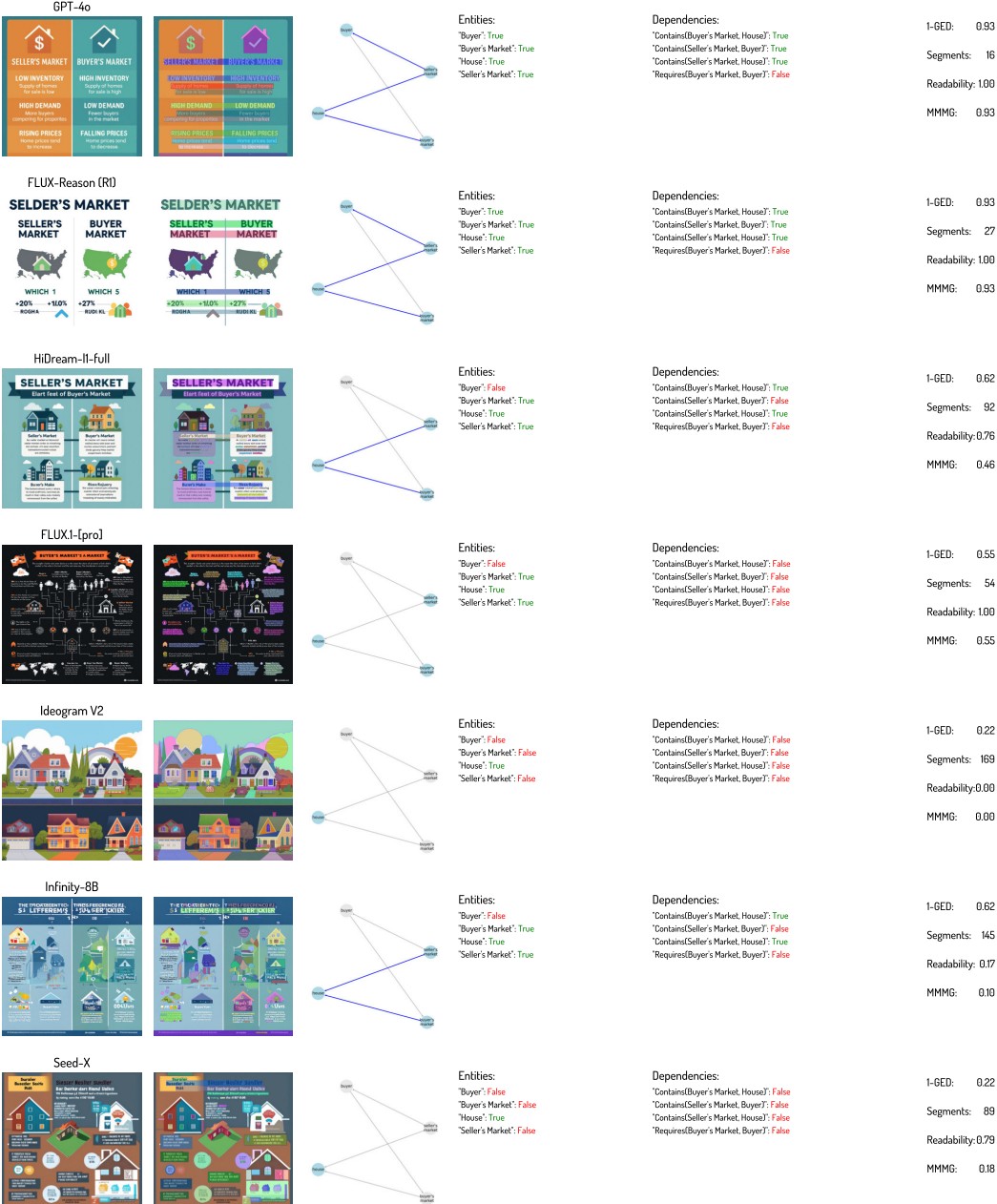

Figure 27: `MMMG` Benchmark visualization for seven representative models on a Secondaryschool-Economics example. Each row corresponds to one model and, from left to right, displays the generated image, its segmentation map, the reconstructed knowledge graph, the extracted entity and dependency lists, and finally the overall `MMMG-Score` along with its component sub-scores.

## F.3.7 Sociology

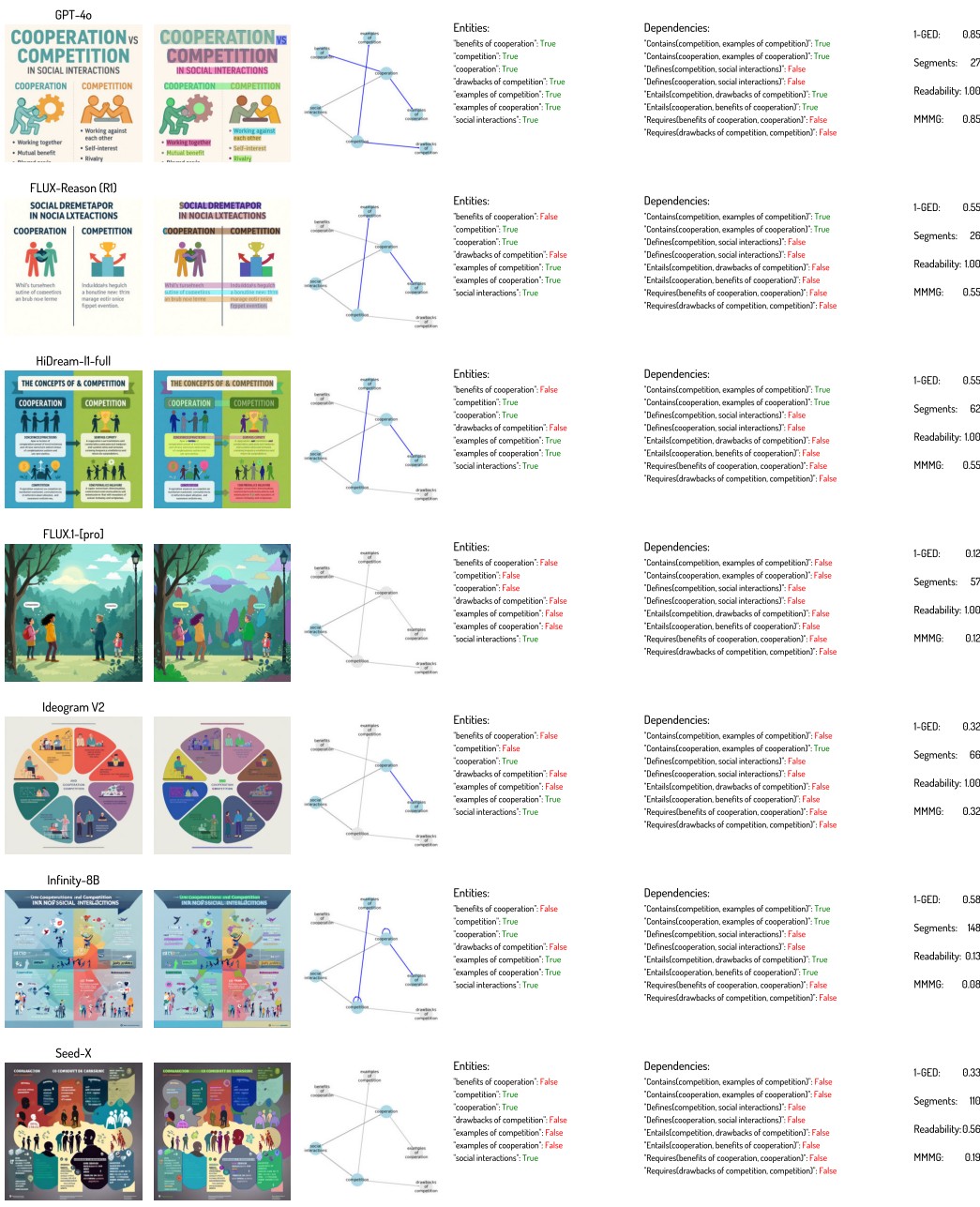

Figure 28: `MMMG` Benchmark visualization for seven representative models on a Secondaryschool-Sociology example. Each row corresponds to one model and, from left to right, displays the generated image, its segmentation map, the reconstructed knowledge graph, the extracted entity and dependency lists, and finally the overall `MMMG-Score` along with its component sub-scores.

## F.3.8 History

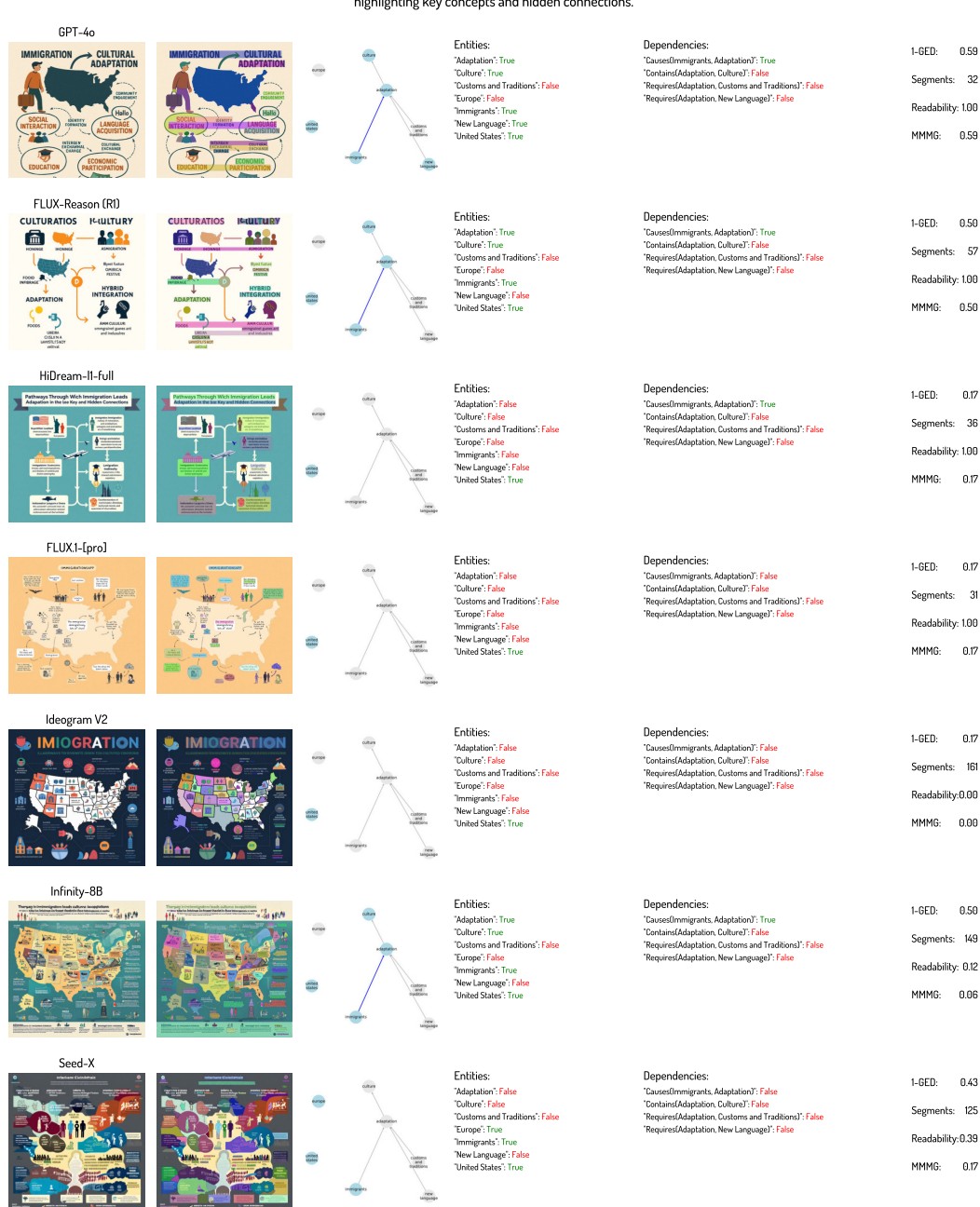

Figure 29: `MMMG` Benchmark visualization for seven representative models on a Secondaryschool-History example. Each row corresponds to one model and, from left to right, displays the generated image, its segmentation map, the reconstructed knowledge graph, the extracted entity and dependency lists, and finally the overall `MMMG-Score` along with its component sub-scores.

## F.3.9 Philosophy

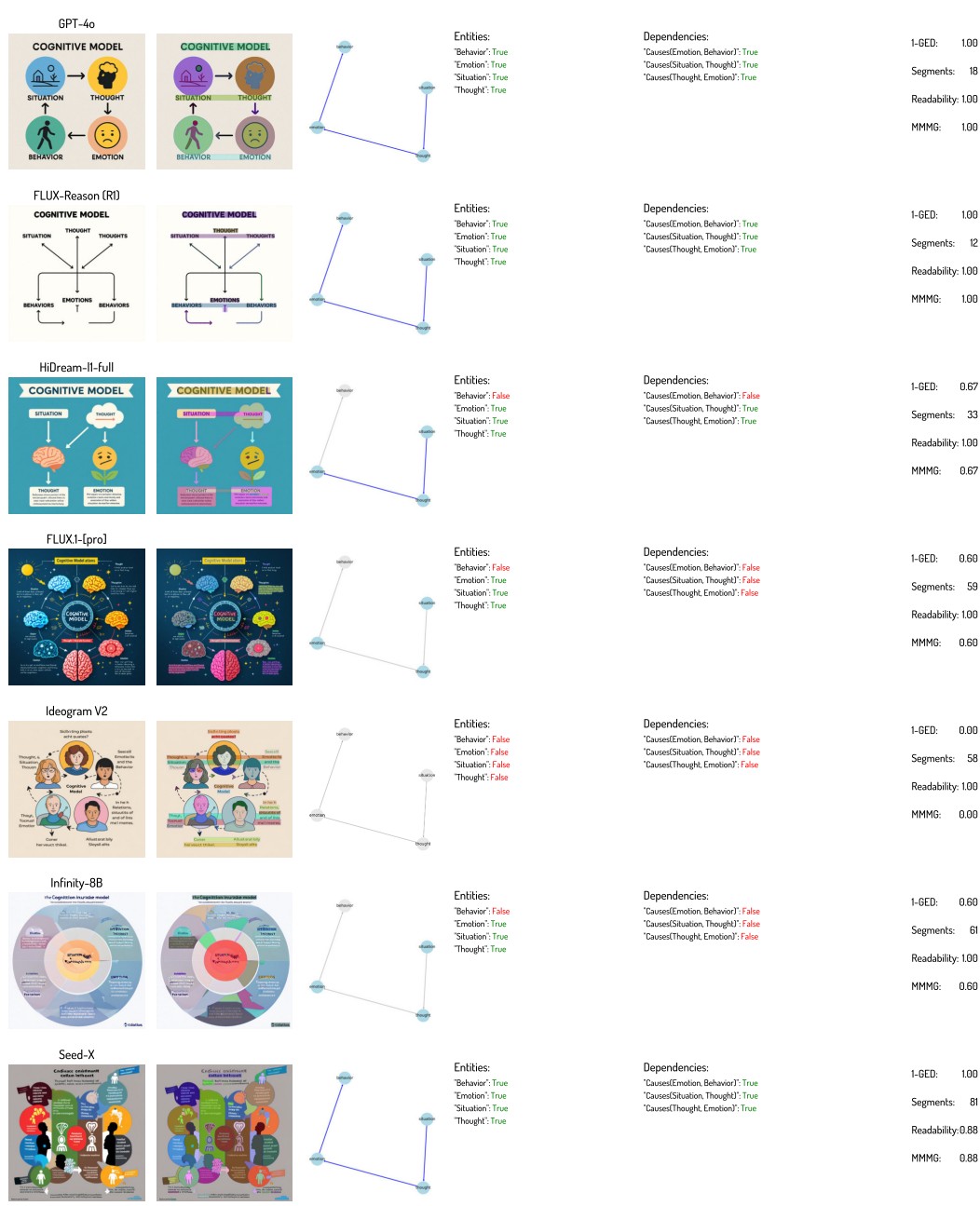

Figure 30: `MMMG` Benchmark visualization for seven representative models on a Secondaryschool-Philosophy example. Each row corresponds to one model and, from left to right, displays the generated image, its segmentation map, the reconstructed knowledge graph, the extracted entity and dependency lists, and finally the overall `MMMG-Score` along with its component sub-scores.

## F.3.10 Literature

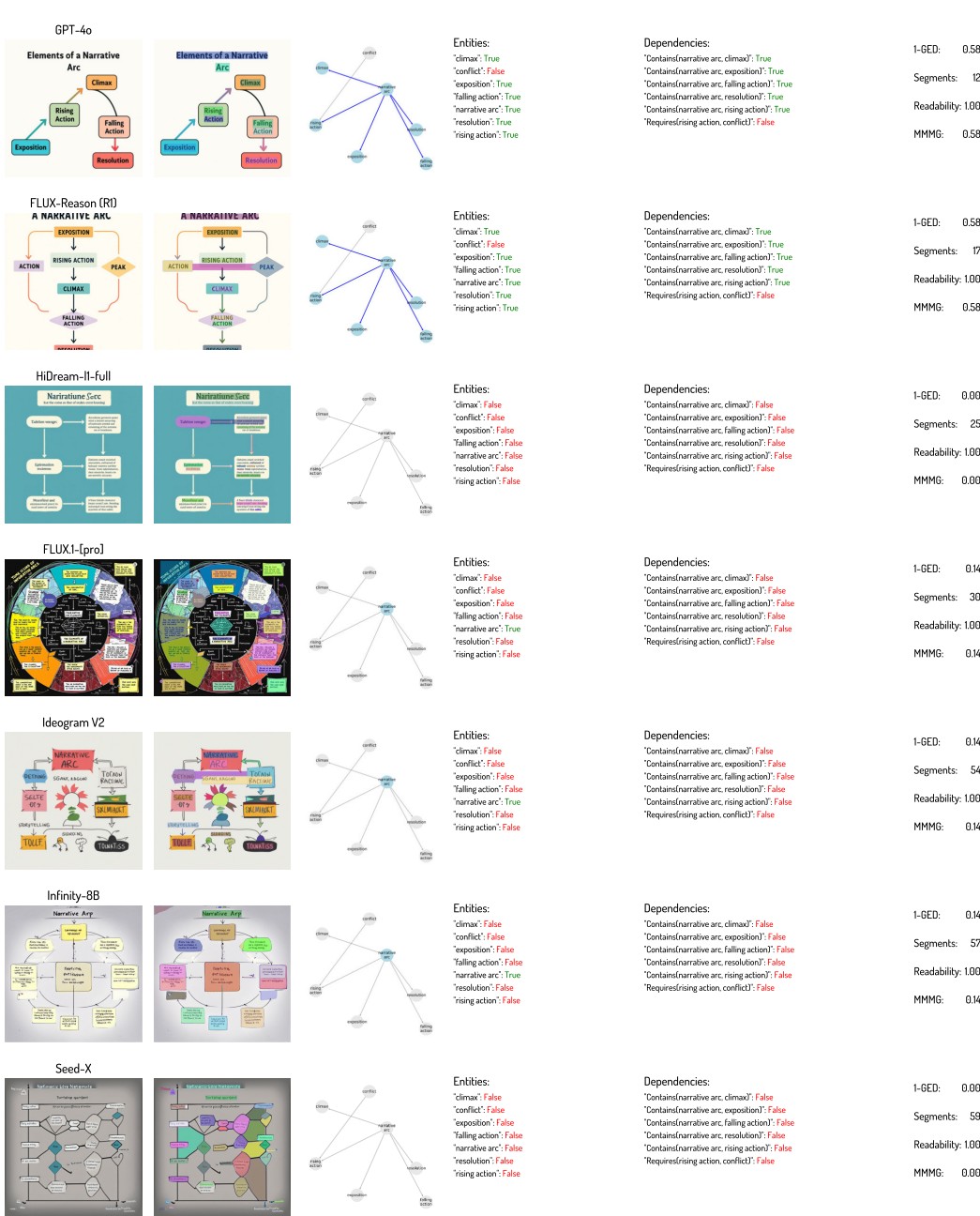

Figure 31: `MMMG` Benchmark visualization for seven representative models on a Secondaryschool-Literature example. Each row corresponds to one model and, from left to right, displays the generated image, its segmentation map, the reconstructed knowledge graph, the extracted entity and dependency lists, and finally the overall `MMMG-Score` along with its component sub-scores.

## F.4 High School

### F.4.1 Biology

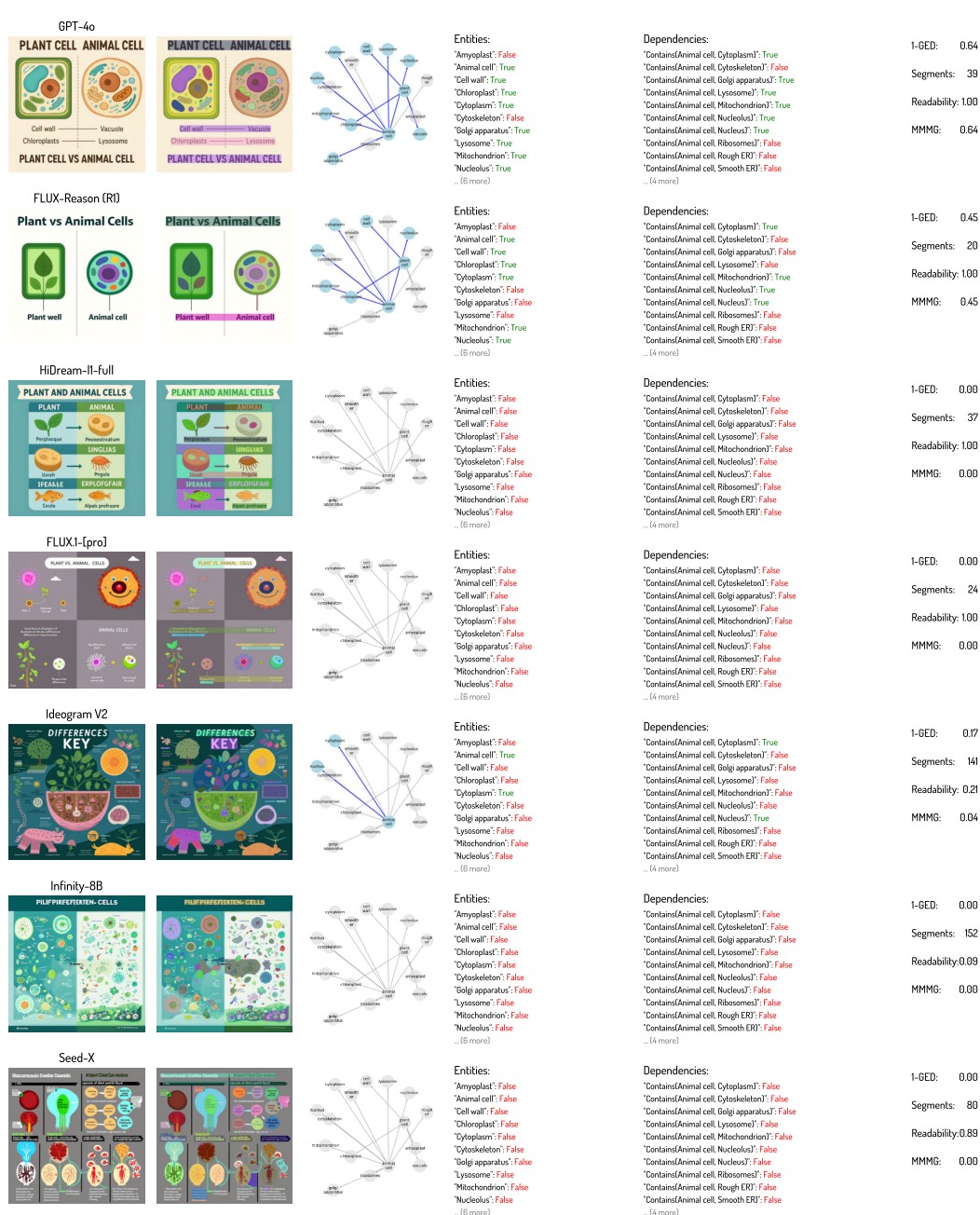

Figure 32: `MMMG` Benchmark visualization for seven representative models on a Highschool-Biology example. Each row corresponds to one model and, from left to right, displays the generated image, its segmentation map, the reconstructed knowledge graph, the extracted entity and dependency lists, and finally the overall `MMMG-Score` along with its component sub-scores.

## F.4.2 Chemistry

Question: Produce a detailed diagram of the electrochemical cell, illustrating the components and the flow of electrons.

Figure 33: `MMMG` Benchmark visualization for seven representative models on a Highschool-Chemistry example. Each row corresponds to one model and, from left to right, displays the generated image, its segmentation map, the reconstructed knowledge graph, the extracted entity and dependency lists, and finally the overall `MMMG-Score` along with its component sub-scores.

### F.4.3 Mathematics

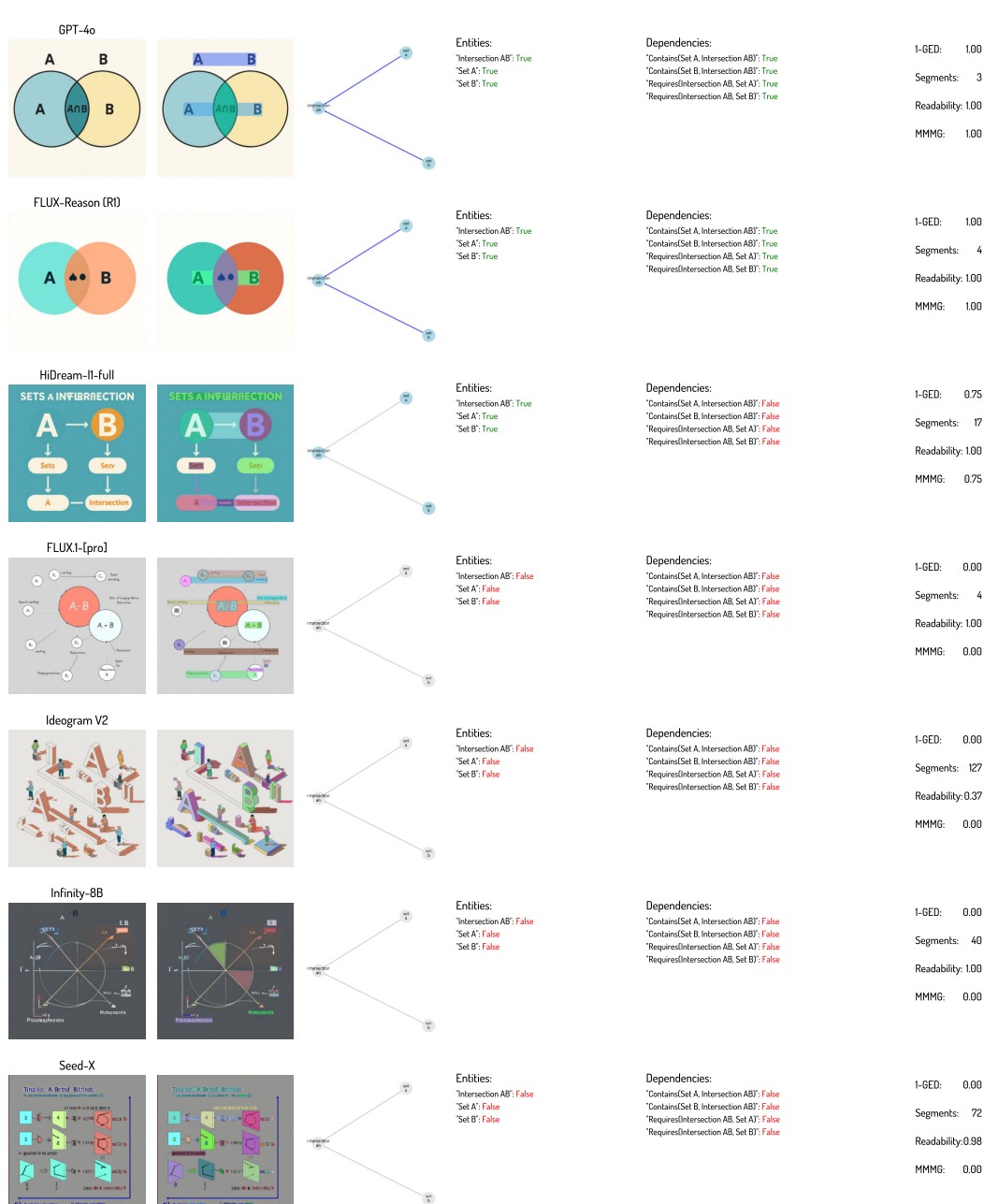

Figure 34: `MMMG` Benchmark visualization for seven representative models on a Highschool-Mathematics example. Each row corresponds to one model and, from left to right, displays the generated image, its segmentation map, the reconstructed knowledge graph, the extracted entity and dependency lists, and finally the overall `MMMG-Score` along with its component sub-scores.

### F.4.4 Engineering

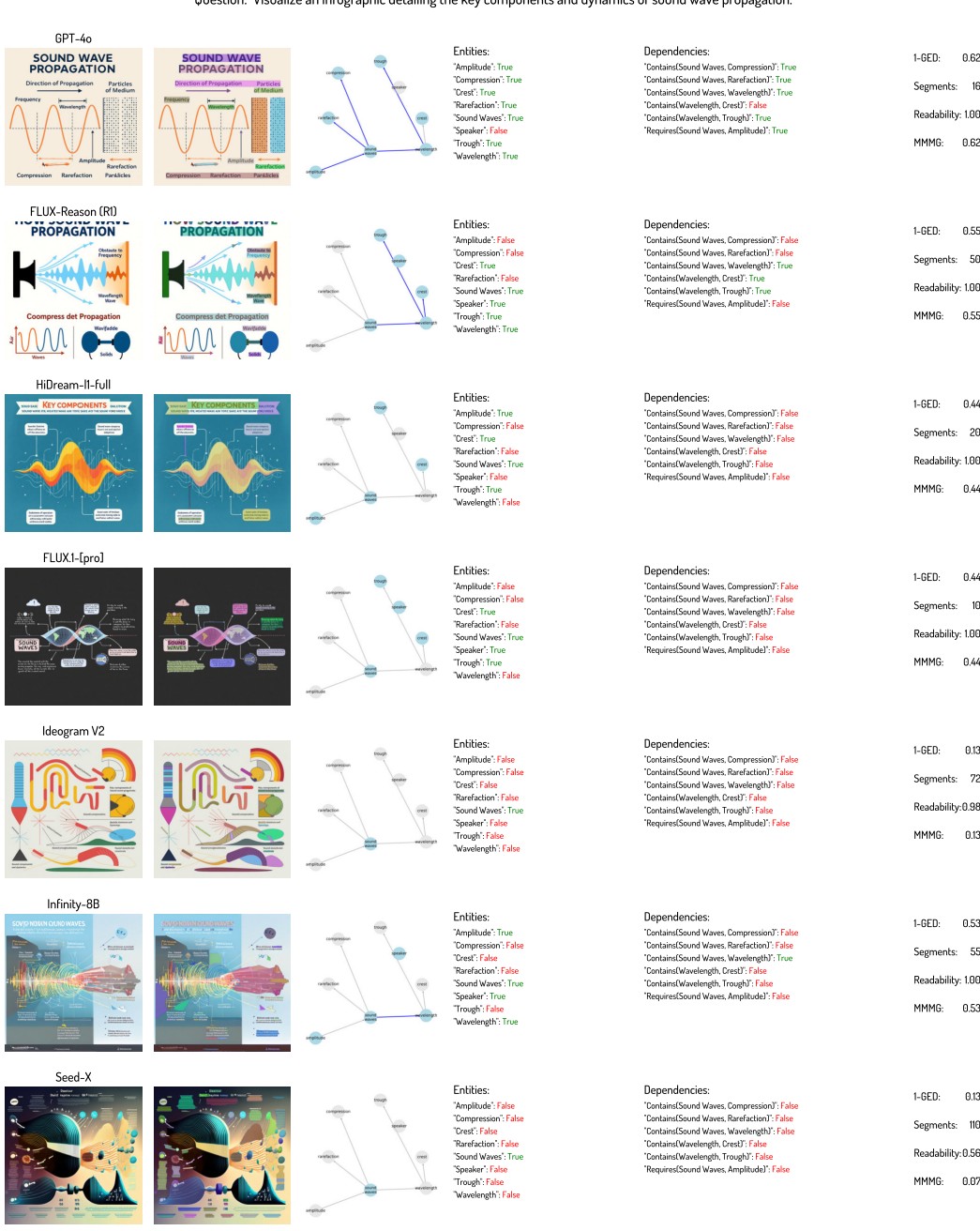

Figure 35: `MMMG` Benchmark visualization for seven representative models on a Highschool-Engineering example. Each row corresponds to one model and, from left to right, displays the generated image, its segmentation map, the reconstructed knowledge graph, the extracted entity and dependency lists, and finally the overall `MMMG-Score` along with its component sub-scores.

## F.4.5 Geography

Question: Visualize an educational diagram highlighting the key zones and features of a glacier.

Figure 36: MMMG Benchmark visualization for seven representative models on a Highschool-Geography example. Each row corresponds to one model and, from left to right, displays the generated image, its segmentation map, the reconstructed knowledge graph, the extracted entity and dependency lists, and finally the overall MMMG-Score along with its component sub-scores.

## F.4.6 Economics

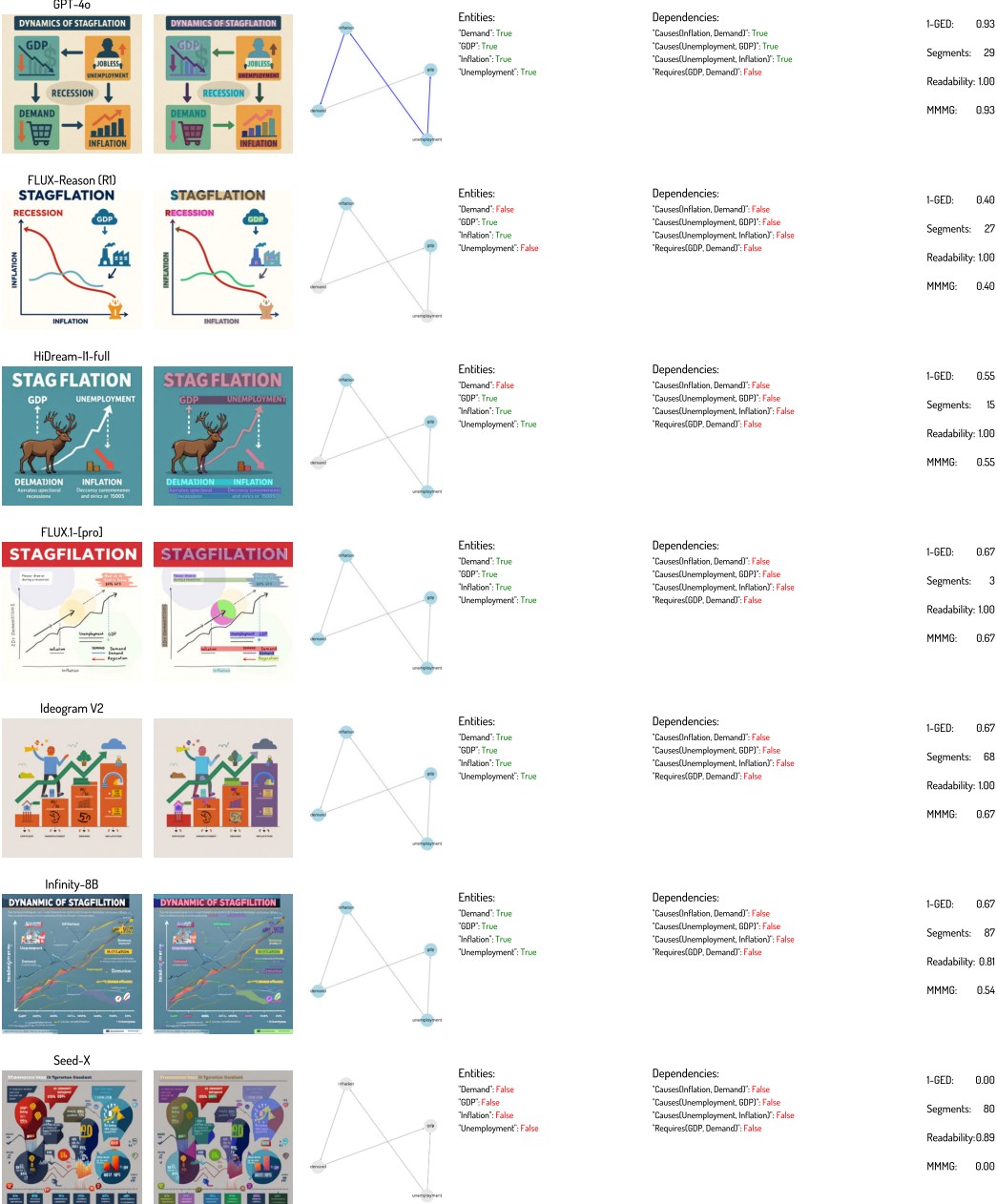

Figure 37: `MMMG` Benchmark visualization for seven representative models on a Highschool-Economics example. Each row corresponds to one model and, from left to right, displays the generated image, its segmentation map, the reconstructed knowledge graph, the extracted entity and dependency lists, and finally the overall `MMMG-Score` along with its component sub-scores.

## F.4.7 Sociology

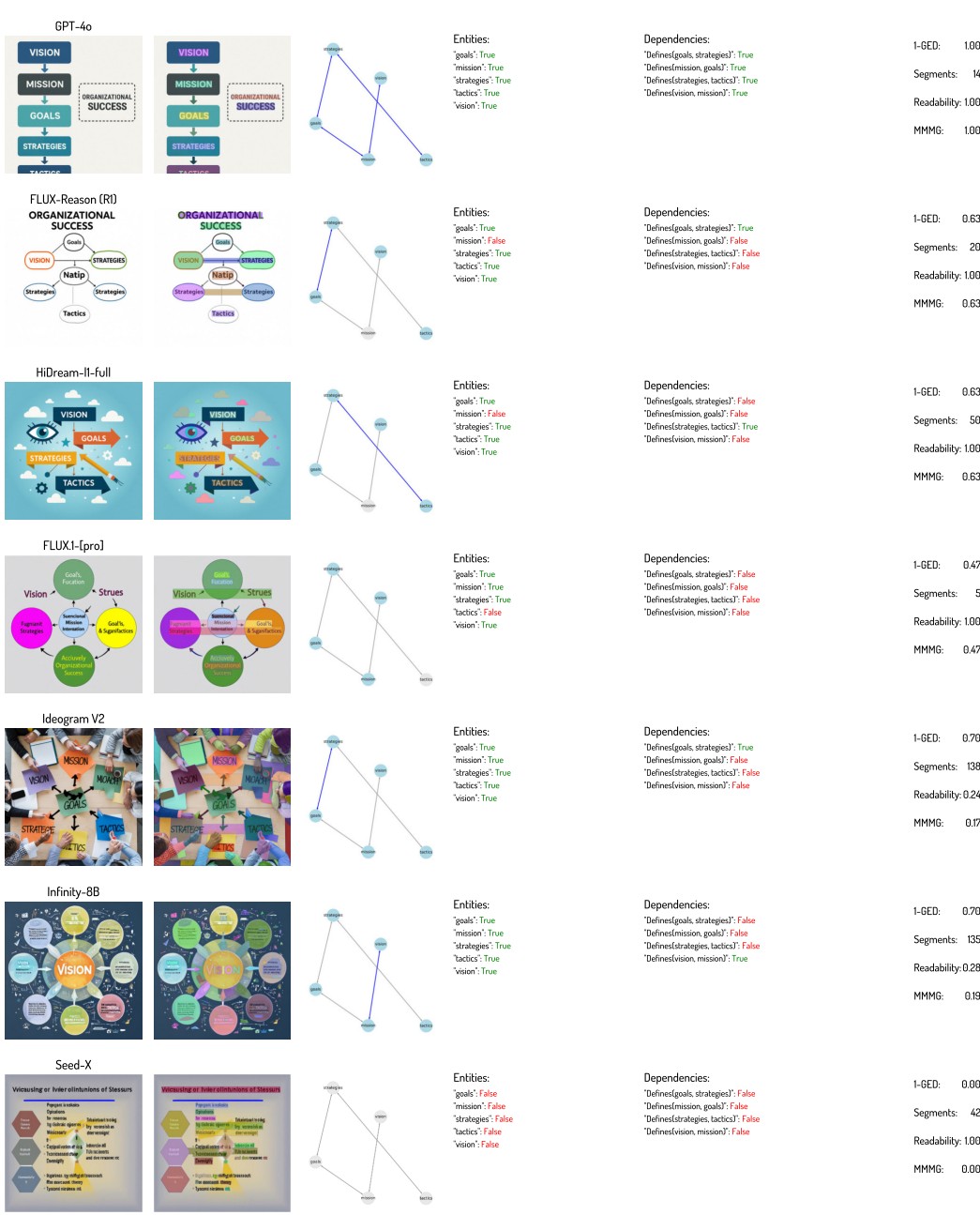

Figure 38: MMMG Benchmark visualization for seven representative models on a Highschool-Sociology example. Each row corresponds to one model and, from left to right, displays the generated image, its segmentation map, the reconstructed knowledge graph, the extracted entity and dependency lists, and finally the overall MMMG-Score along with its component sub-scores.

## F.4.8    History

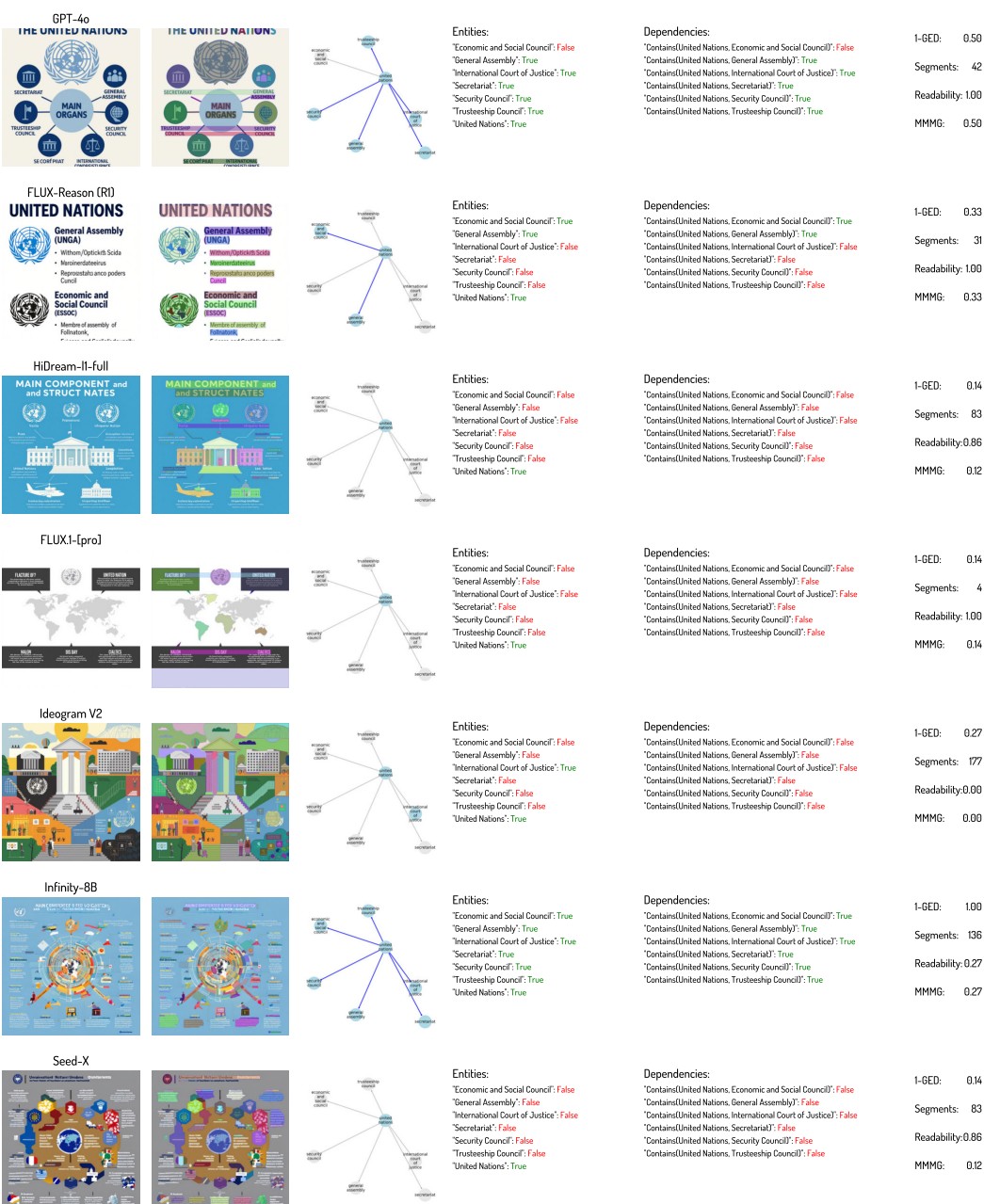

Figure 39: `MMMG` Benchmark visualization for seven representative models on a Highschool-History example. Each row corresponds to one model and, from left to right, displays the generated image, its segmentation map, the reconstructed knowledge graph, the extracted entity and dependency lists, and finally the overall `MMMG-Score` along with its component sub-scores.

## F.4.9 Philosophy

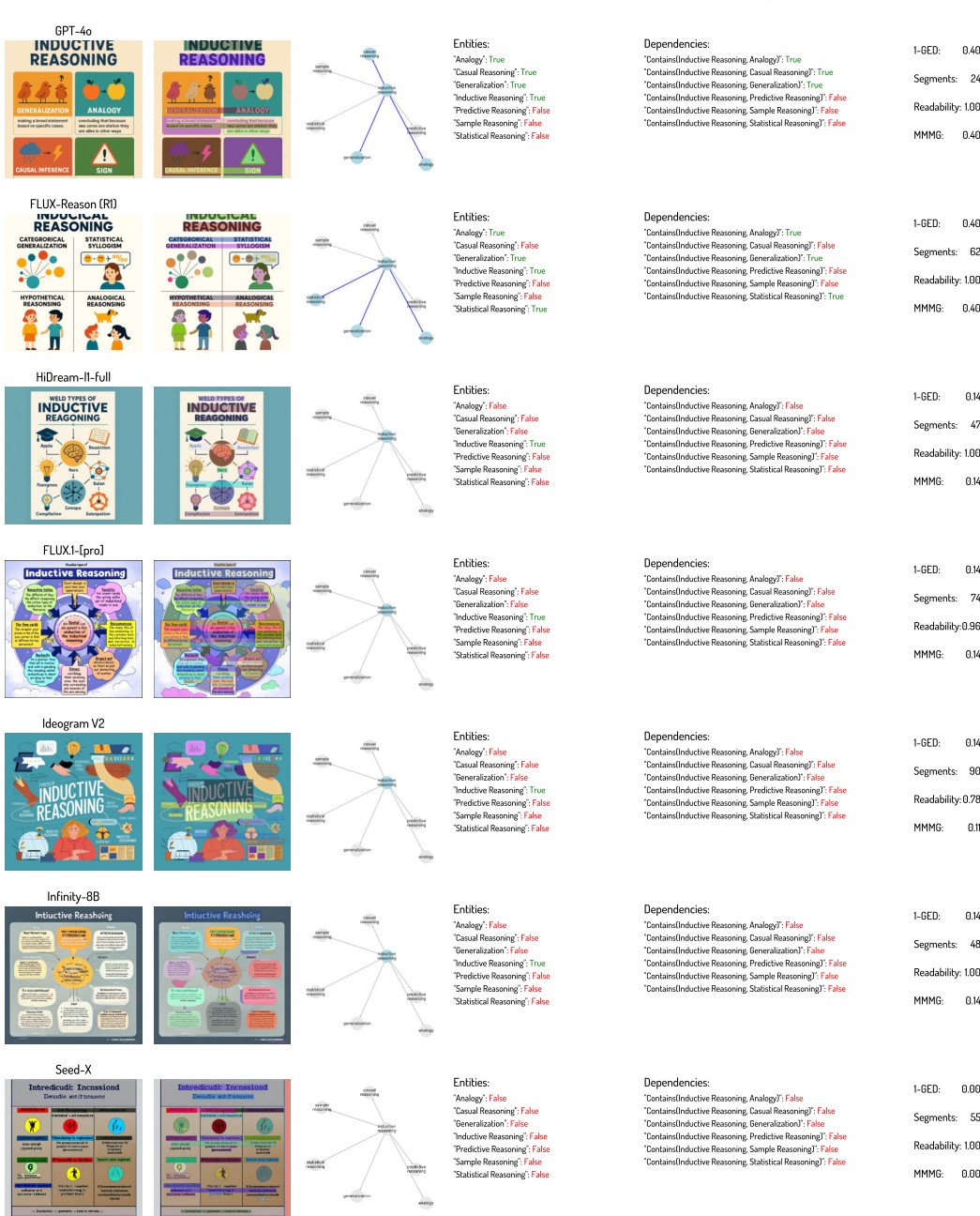

Figure 40: MMMG Benchmark visualization for seven representative models on a Highschool-Philosophy example. Each row corresponds to one model and, from left to right, displays the generated image, its segmentation map, the reconstructed knowledge graph, the extracted entity and dependency lists, and finally the overall MMMG-Score along with its component sub-scores.

## F.4.10 Literature

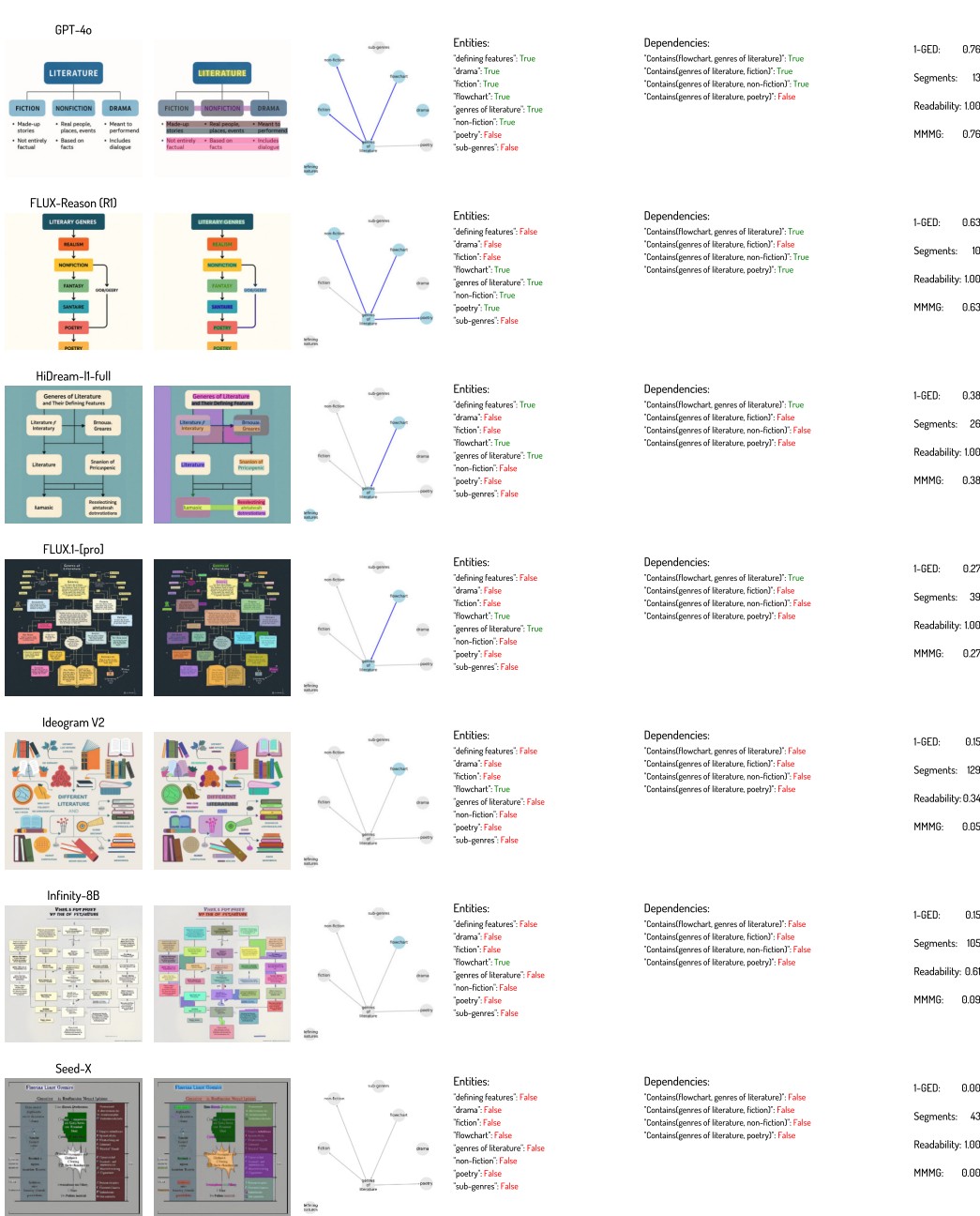

Figure 41: `MMMG` Benchmark visualization for seven representative models on a Highschool-Literature example. Each row corresponds to one model and, from left to right, displays the generated image, its segmentation map, the reconstructed knowledge graph, the extracted entity and dependency lists, and finally the overall `MMMG-Score` along with its component sub-scores.

## F.5 Undergraduate

### F.5.1 Biology

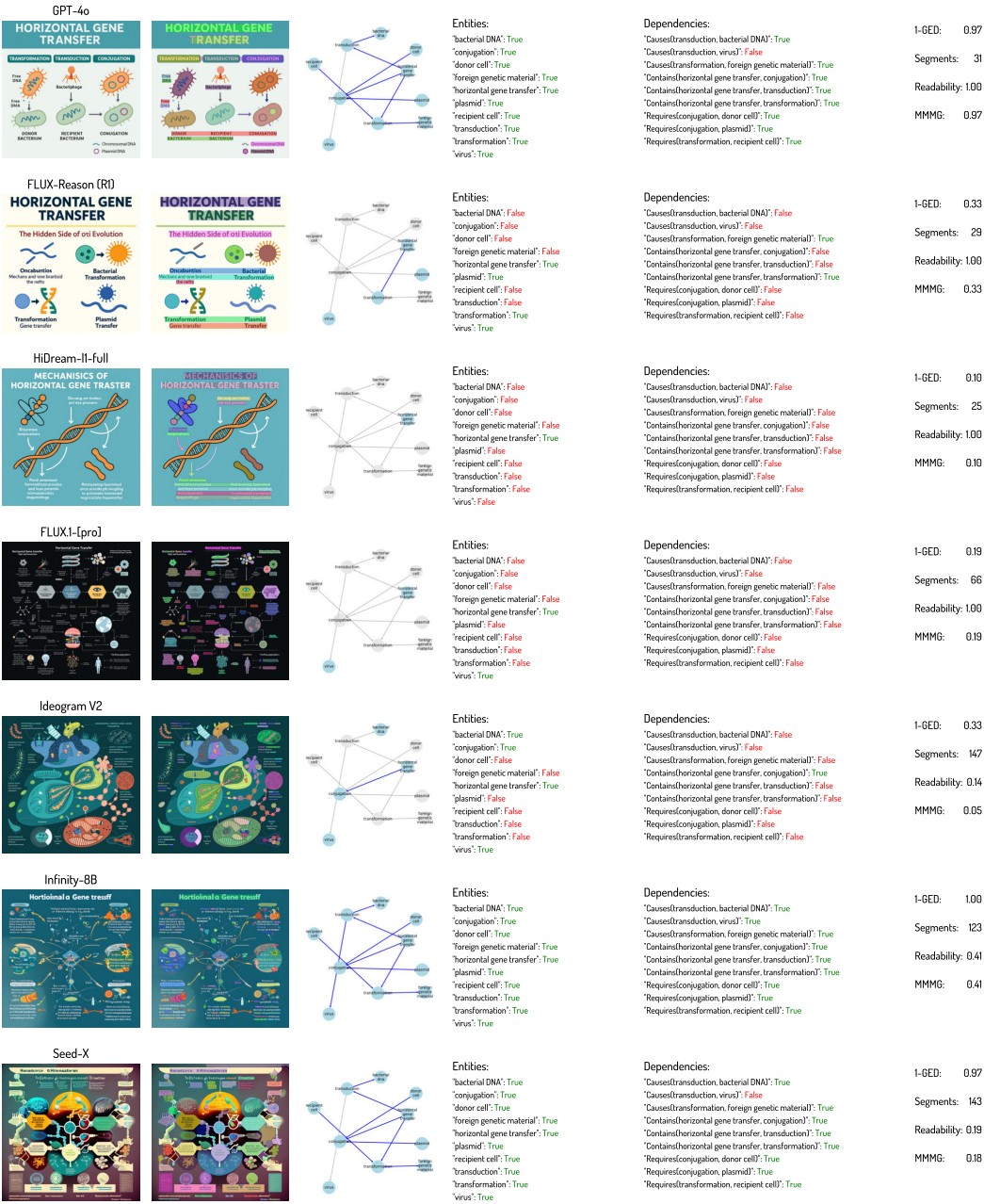

Figure 42: MMMG Benchmark visualization for seven representative models on a Undergraduate-Biology example. Each row corresponds to one model and, from left to right, displays the generated image, its segmentation map, the reconstructed knowledge graph, the extracted entity and dependency lists, and finally the overall MMMG-Score along with its component sub-scores.

## F.5.2 Chemistry

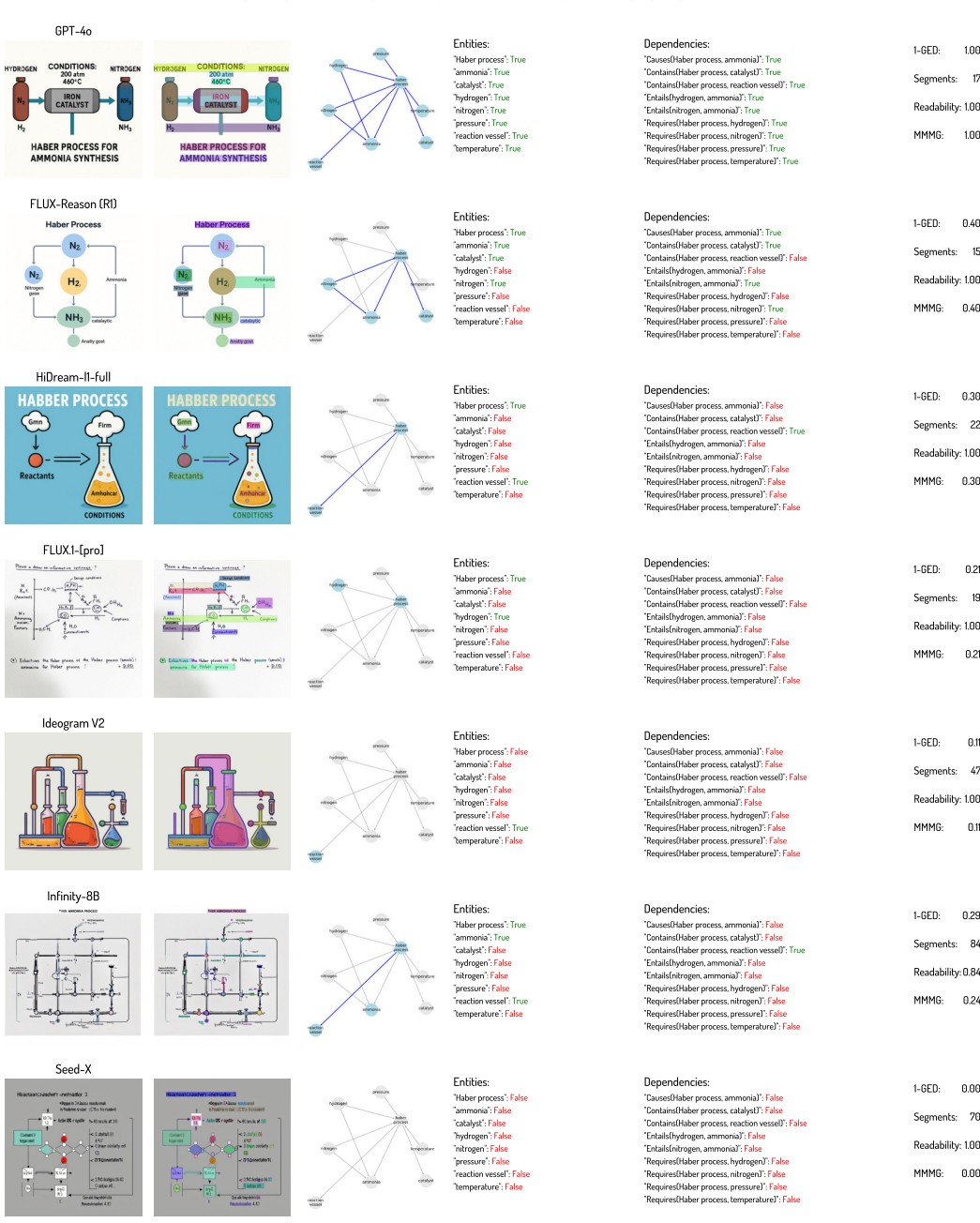

Figure 43: `MMMG` Benchmark visualization for seven representative models on a Undergraduate-Chemistry example. Each row corresponds to one model and, from left to right, displays the generated image, its segmentation map, the reconstructed knowledge graph, the extracted entity and dependency lists, and finally the overall `MMMG-Score` along with its component sub-scores.

### F.5.3 Mathematics

Question: Visualize a diagram showing the relationship between the areas under curves and definite integrals, accompanied by illustrative examples.

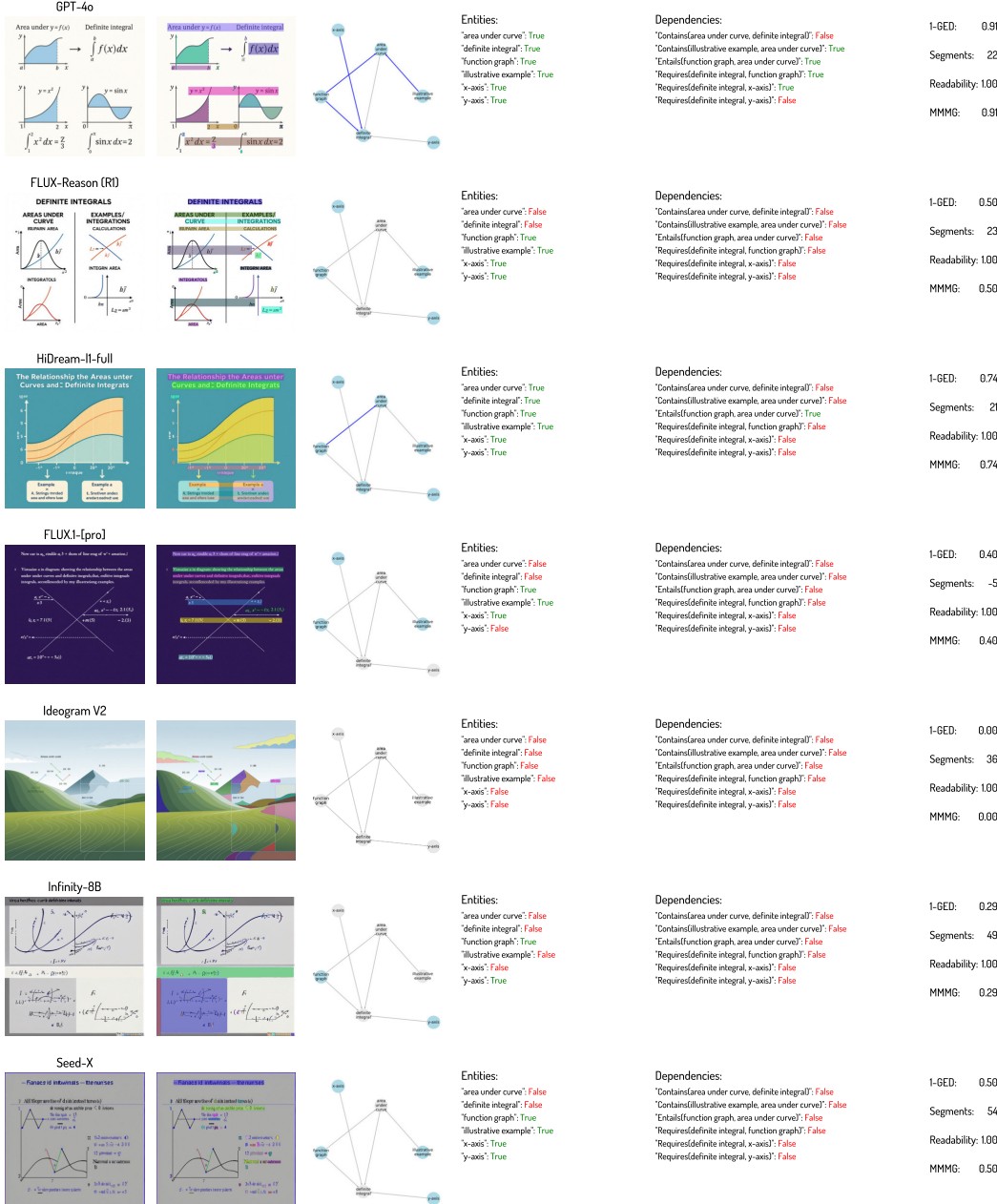

Figure 44: `MMMG` Benchmark visualization for seven representative models on a Undergraduate-Mathematics example. Each row corresponds to one model and, from left to right, displays the generated image, its segmentation map, the reconstructed knowledge graph, the extracted entity and dependency lists, and finally the overall `MMMG-Score` along with its component sub-scores.

## F.5.4 Engineering

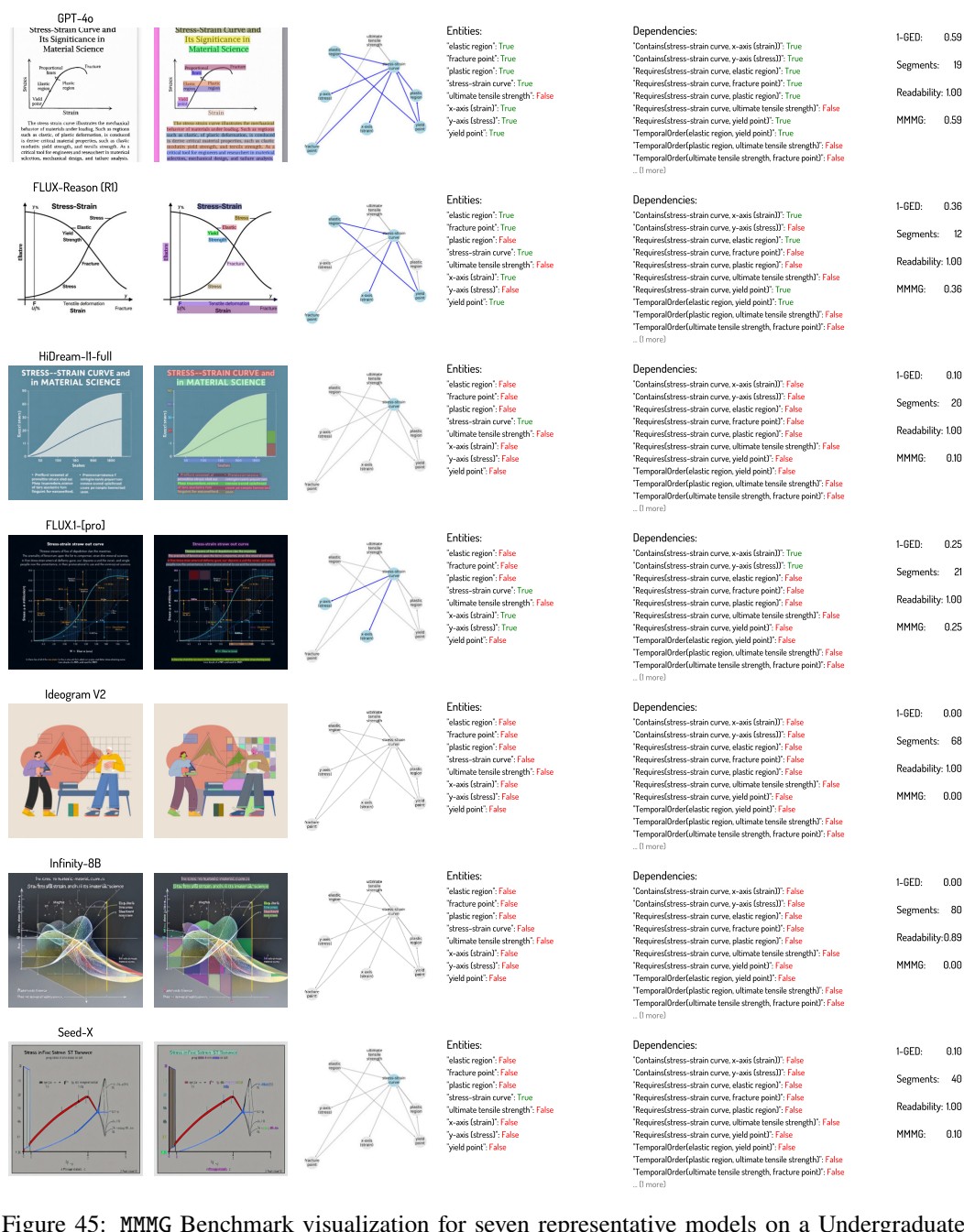

Figure 45: `MMMG` Benchmark visualization for seven representative models on a Undergraduate-Engineering example. Each row corresponds to one model and, from left to right, displays the generated image, its segmentation map, the reconstructed knowledge graph, the extracted entity and dependency lists, and finally the overall `MMMG-Score` along with its component sub-scores.

## F.5.5 Geography

Question: Create a diagram showing the process of glacial erosion, illustrating features such as fjords and U-shaped valleys.

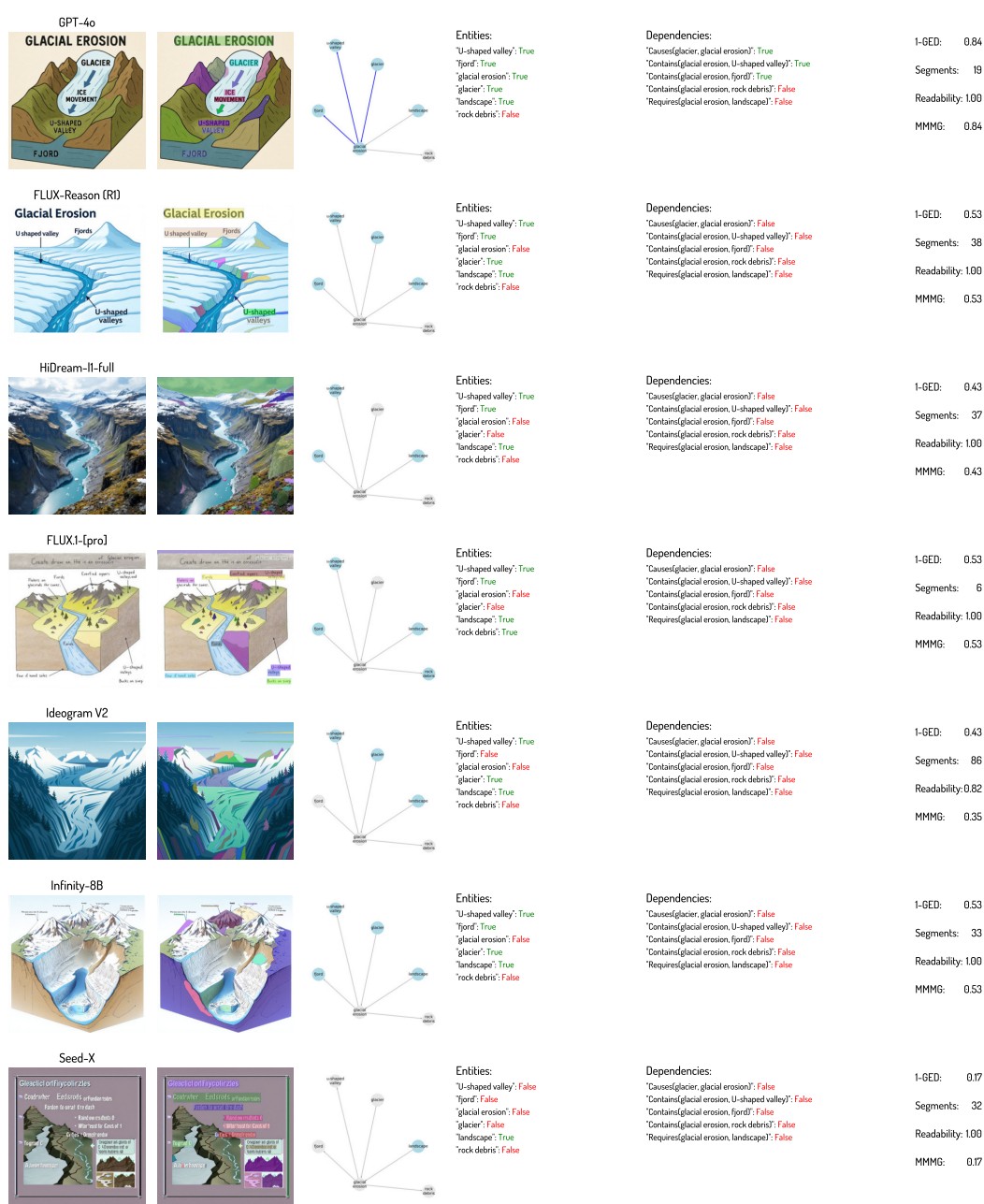

Figure 46: `MMMG` Benchmark visualization for seven representative models on a Undergraduate-Geography example. Each row corresponds to one model and, from left to right, displays the generated image, its segmentation map, the reconstructed knowledge graph, the extracted entity and dependency lists, and finally the overall `MMMG-Score` along with its component sub-scores.

## F.5.6 Economics

Question: Visualize the components and implications of the PESTEL framework in an educational poster format.

Figure 47: `MMMG` Benchmark visualization for seven representative models on a Undergraduate-Economics example. Each row corresponds to one model and, from left to right, displays the generated image, its segmentation map, the reconstructed knowledge graph, the extracted entity and dependency lists, and finally the overall `MMMG-Score` along with its component sub-scores.

## F.5.7    Sociology

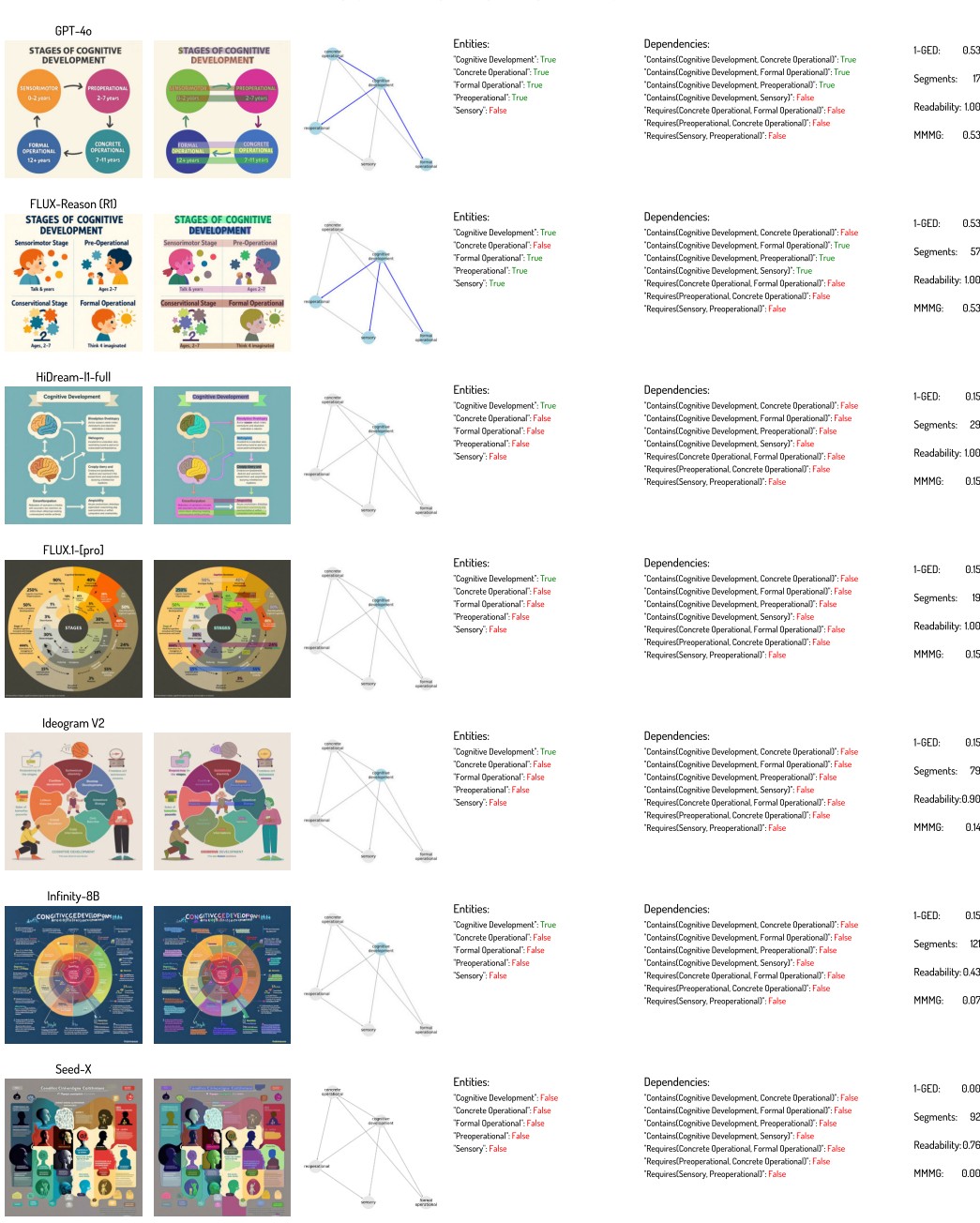

Figure 48: `MMMG` Benchmark visualization for seven representative models on a Undergraduate-Sociology example. Each row corresponds to one model and, from left to right, displays the generated image, its segmentation map, the reconstructed knowledge graph, the extracted entity and dependency lists, and finally the overall `MMMG-Score` along with its component sub-scores.

## F.5.8 History

Question: Create an educational poster on the significance of the Trail of Tears in U.S. history, emphasizing its impact on Native American communities.

Figure 49: `MMMG` Benchmark visualization for seven representative models on a Undergraduate-History example. Each row corresponds to one model and, from left to right, displays the generated image, its segmentation map, the reconstructed knowledge graph, the extracted entity and dependency lists, and finally the overall `MMMG-Score` along with its component sub-scores.

## F.5.9 Philosophy

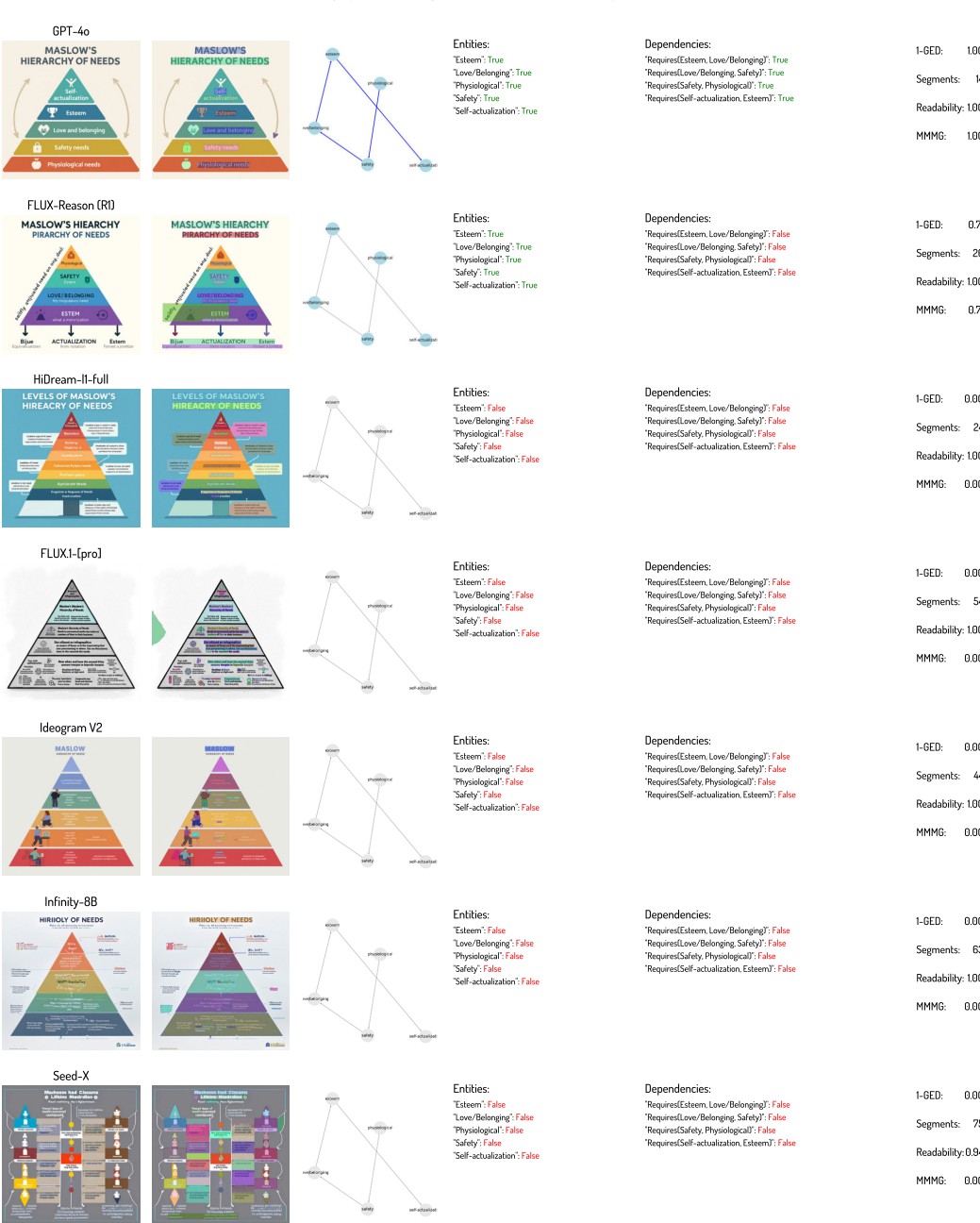

Figure 50: `MMMG` Benchmark visualization for seven representative models on a Undergraduate-Philosophy example. Each row corresponds to one model and, from left to right, displays the generated image, its segmentation map, the reconstructed knowledge graph, the extracted entity and dependency lists, and finally the overall `MMMG-Score` along with its component sub-scores.

## F.5.10 Literature

Question: Design an educational poster that illustrates the key elements of dystopian fiction using examples from literature.

Figure 51: `MMMG` Benchmark visualization for seven representative models on a Undergraduate-Literature example. Each row corresponds to one model and, from left to right, displays the generated image, its segmentation map, the reconstructed knowledge graph, the extracted entity and dependency lists, and finally the overall `MMMG-Score` along with its component sub-scores.

## F.6 PhD

### F.6.1 Biology

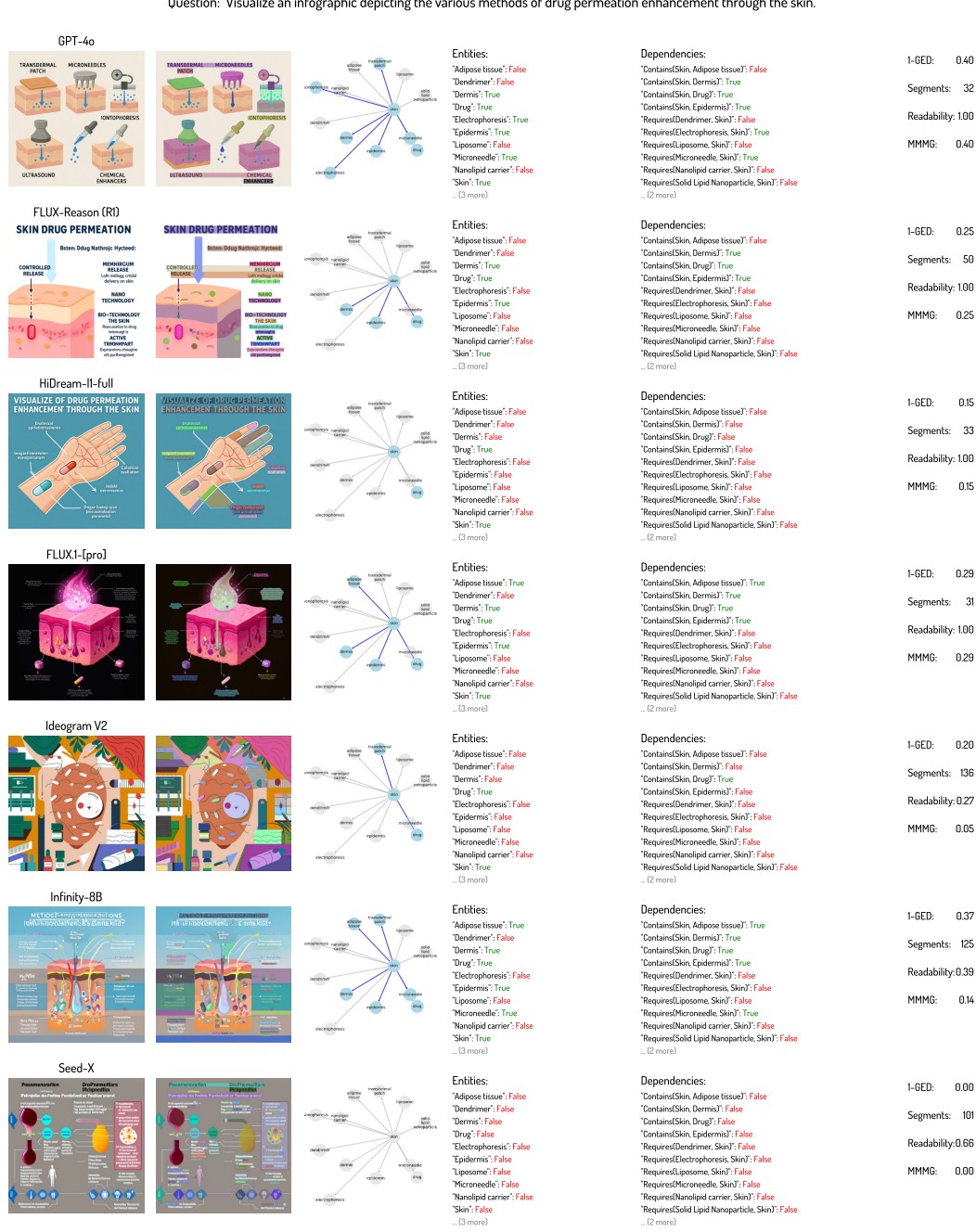

Figure 52: MMMG Benchmark visualization for seven representative models on a PhD-Biology example. Each row corresponds to one model and, from left to right, displays the generated image, its segmentation map, the reconstructed knowledge graph, the extracted entity and dependency lists, and finally the overall MMMG-Score along with its component sub-scores.

## F.6.2 Chemistry

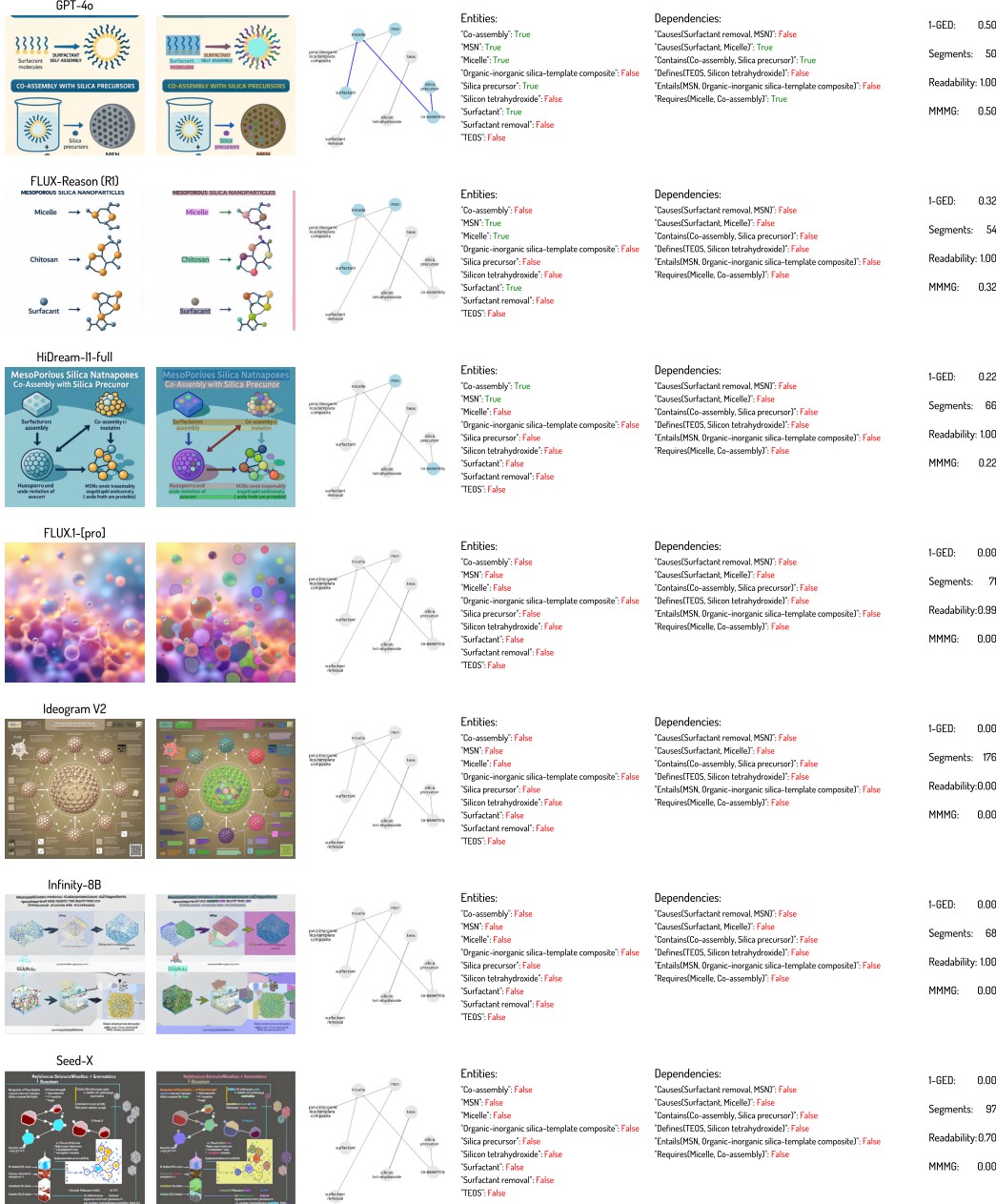

Figure 53: `MMMG` Benchmark visualization for seven representative models on a PhD-Chemistry example. Each row corresponds to one model and, from left to right, displays the generated image, its segmentation map, the reconstructed knowledge graph, the extracted entity and dependency lists, and finally the overall `MMMG-Score` along with its component sub-scores.

## F.6.3 Mathematics

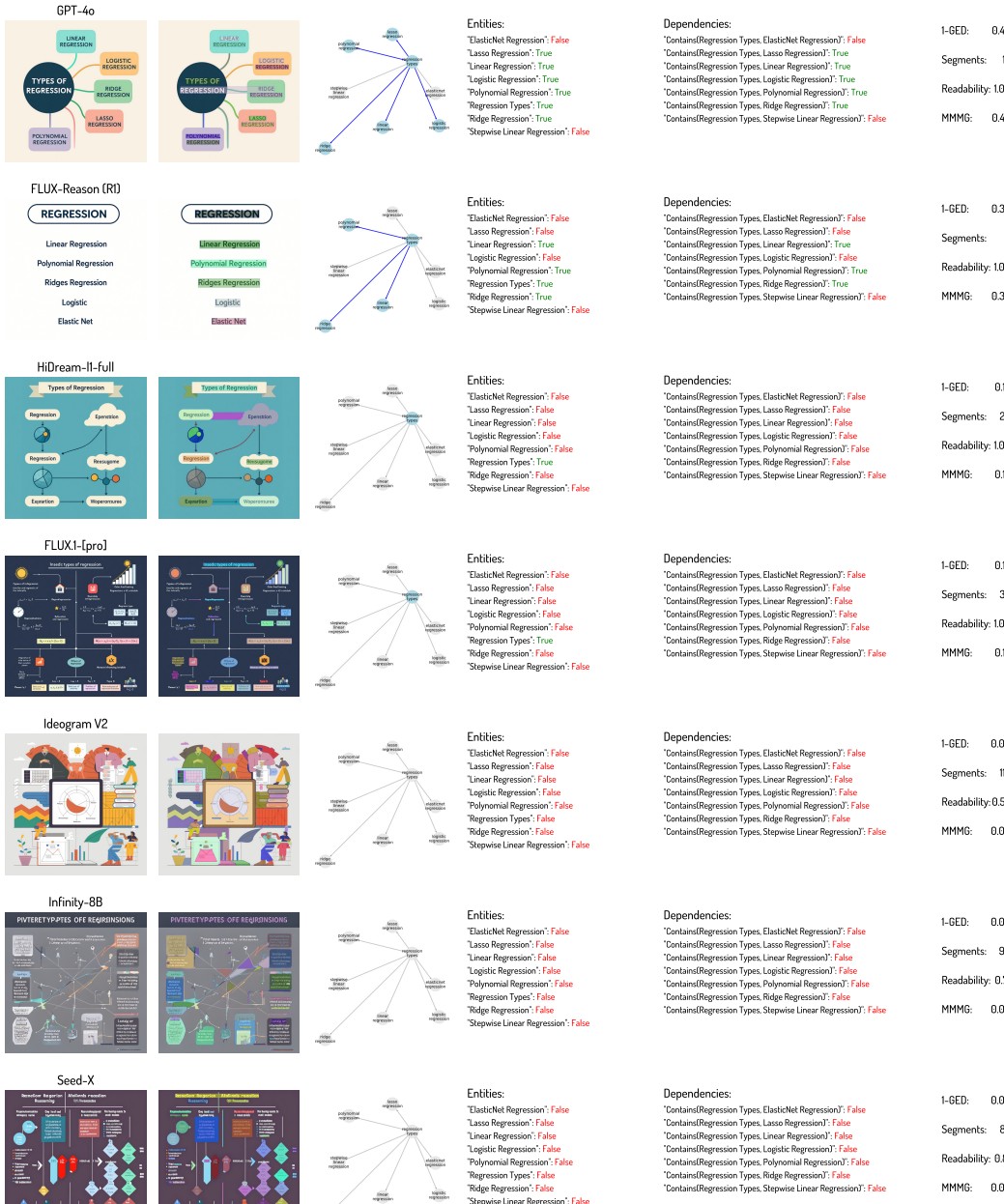

Figure 54: `MMMG` Benchmark visualization for seven representative models on a PhD-Mathematics example. Each row corresponds to one model and, from left to right, displays the generated image, its segmentation map, the reconstructed knowledge graph, the extracted entity and dependency lists, and finally the overall `MMMG-Score` along with its component sub-scores.

## F.6.4 Engineering

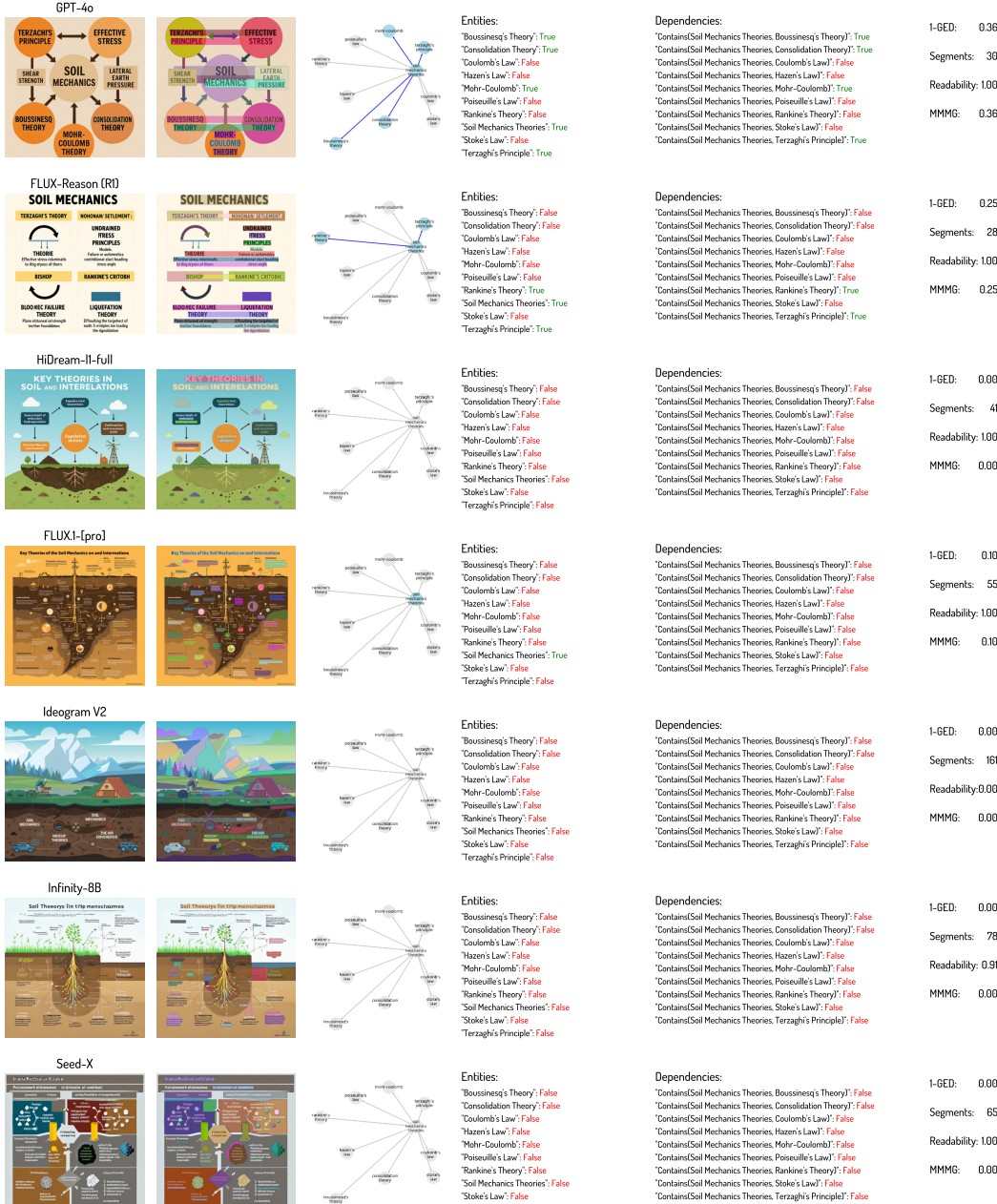

Figure 55: `MMMG` Benchmark visualization for seven representative models on a PhD-Engineering example. Each row corresponds to one model and, from left to right, displays the generated image, its segmentation map, the reconstructed knowledge graph, the extracted entity and dependency lists, and finally the overall `MMMG-Score` along with its component sub-scores.

## F.6.5 Geography

Question: Visualize an infographic illustrating the various methods of detecting exoplanets and their comparisons.

Figure 56: `MMMG` Benchmark visualization for seven representative models on a PhD-Geography example. Each row corresponds to one model and, from left to right, displays the generated image, its segmentation map, the reconstructed knowledge graph, the extracted entity and dependency lists, and finally the overall `MMMG-Score` along with its component sub-scores.

## F.6.6 Economics

Question: Visualize an infographic analyzing the effects of a price ceiling on market equilibrium and surpluses.

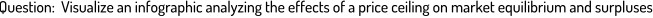

**GPT-4o**

Entities:
"Consumer surplus": True
"Deadweight loss": True
"Demand curve": True
"Equilibrium quantity": True
"Free market equilibrium": True
"Market price at equilibrium": True
"Price ceiling": True
"Producer surplus": True
"Supply curve": True

Dependencies:
"Causes(Price ceiling, Deadweight loss)": False
"Contains(Demand curve, Consumer surplus)": False
"Contains(Supply curve, Producer surplus)": False
"Defines(Free market equilibrium, Market price at equilibrium)": True
"Entails(Free market equilibrium, Equilibrium quantity)": True
"Requires(Price ceiling, Free market equilibrium)": False

1-GED:       0.86
Segments:    19
Readability: 1.00
MMMG:        0.86

**FLUX-Reason (R1)**

Entities:
"Consumer surplus": False
"Deadweight loss": False
"Demand curve": True
"Equilibrium quantity": False
"Free market equilibrium": False
"Market price at equilibrium": False
"Price ceiling": False
"Producer surplus": False
"Supply curve": True

Dependencies:
"Causes(Price ceiling, Deadweight loss)": False
"Contains(Demand curve, Consumer surplus)": False
"Contains(Supply curve, Producer surplus)": False
"Defines(Free market equilibrium, Market price at equilibrium)": False
"Entails(Free market equilibrium, Equilibrium quantity)": False
"Requires(Price ceiling, Free market equilibrium)": False

1-GED:       0.32
Segments:    39
Readability: 1.00
MMMG:        0.32

**HiDream-I1-full**

Entities:
"Consumer surplus": False
"Deadweight loss": False
"Demand curve": True
"Equilibrium quantity": True
"Free market equilibrium": True
"Market price at equilibrium": True
"Price ceiling": True
"Producer surplus": False
"Supply curve": True

Dependencies:
"Causes(Price ceiling, Deadweight loss)": False
"Contains(Demand curve, Consumer surplus)": False
"Contains(Supply curve, Producer surplus)": False
"Defines(Free market equilibrium, Market price at equilibrium)": True
"Entails(Free market equilibrium, Equilibrium quantity)": True
"Requires(Price ceiling, Free market equilibrium)": False

1-GED:       0.48
Segments:    29
Readability: 1.00
MMMG:        0.48

**FLUX.1-[pro]**

Entities:
"Consumer surplus": False
"Deadweight loss": False
"Demand curve": False
"Equilibrium quantity": False
"Free market equilibrium": False
"Market price at equilibrium": False
"Price ceiling": False
"Producer surplus": False
"Supply curve": False

Dependencies:
"Causes(Price ceiling, Deadweight loss)": False
"Contains(Demand curve, Consumer surplus)": False
"Contains(Supply curve, Producer surplus)": False
"Defines(Free market equilibrium, Market price at equilibrium)": False
"Entails(Free market equilibrium, Equilibrium quantity)": False
"Requires(Price ceiling, Free market equilibrium)": False

1-GED:       0.00
Segments:    56
Readability: 1.00
MMMG:        0.00

**Ideogram V2**

Entities:
"Consumer surplus": False
"Deadweight loss": False
"Demand curve": False
"Equilibrium quantity": False
"Free market equilibrium": False
"Market price at equilibrium": False
"Price ceiling": False
"Producer surplus": False
"Supply curve": False

Dependencies:
"Causes(Price ceiling, Deadweight loss)": False
"Contains(Demand curve, Consumer surplus)": False
"Contains(Supply curve, Producer surplus)": False
"Defines(Free market equilibrium, Market price at equilibrium)": False
"Entails(Free market equilibrium, Equilibrium quantity)": False
"Requires(Price ceiling, Free market equilibrium)": False

1-GED:       0.00
Segments:    165
Readability: 0.00
MMMG:        0.00

**Infinity-8B**

Entities:
"Consumer surplus": False
"Deadweight loss": False
"Demand curve": True
"Equilibrium quantity": False
"Free market equilibrium": False
"Market price at equilibrium": False
"Price ceiling": False
"Producer surplus": False
"Supply curve": True

Dependencies:
"Causes(Price ceiling, Deadweight loss)": False
"Contains(Demand curve, Consumer surplus)": False
"Contains(Supply curve, Producer surplus)": False
"Defines(Free market equilibrium, Market price at equilibrium)": False
"Entails(Free market equilibrium, Equilibrium quantity)": False
"Requires(Price ceiling, Free market equilibrium)": False

1-GED:       0.32
Segments:    145
Readability: 0.17
MMMG:        0.05

**Seed-X**

Entities:
"Consumer surplus": False
"Deadweight loss": False
"Demand curve": False
"Equilibrium quantity": False
"Free market equilibrium": False
"Market price at equilibrium": False
"Price ceiling": False
"Producer surplus": False
"Supply curve": False

Dependencies:
"Causes(Price ceiling, Deadweight loss)": False
"Contains(Demand curve, Consumer surplus)": False
"Contains(Supply curve, Producer surplus)": False
"Defines(Free market equilibrium, Market price at equilibrium)": False
"Entails(Free market equilibrium, Equilibrium quantity)": False
"Requires(Price ceiling, Free market equilibrium)": False

1-GED:       0.00
Segments:    81
Readability: 0.88
MMMG:        0.00

Figure 57: `MMMG` Benchmark visualization for seven representative models on a PhD-Economics example. Each row corresponds to one model and, from left to right, displays the generated image, its segmentation map, the reconstructed knowledge graph, the extracted entity and dependency lists, and finally the overall `MMMG-Score` along with its component sub-scores.

## F.6.7 Sociology

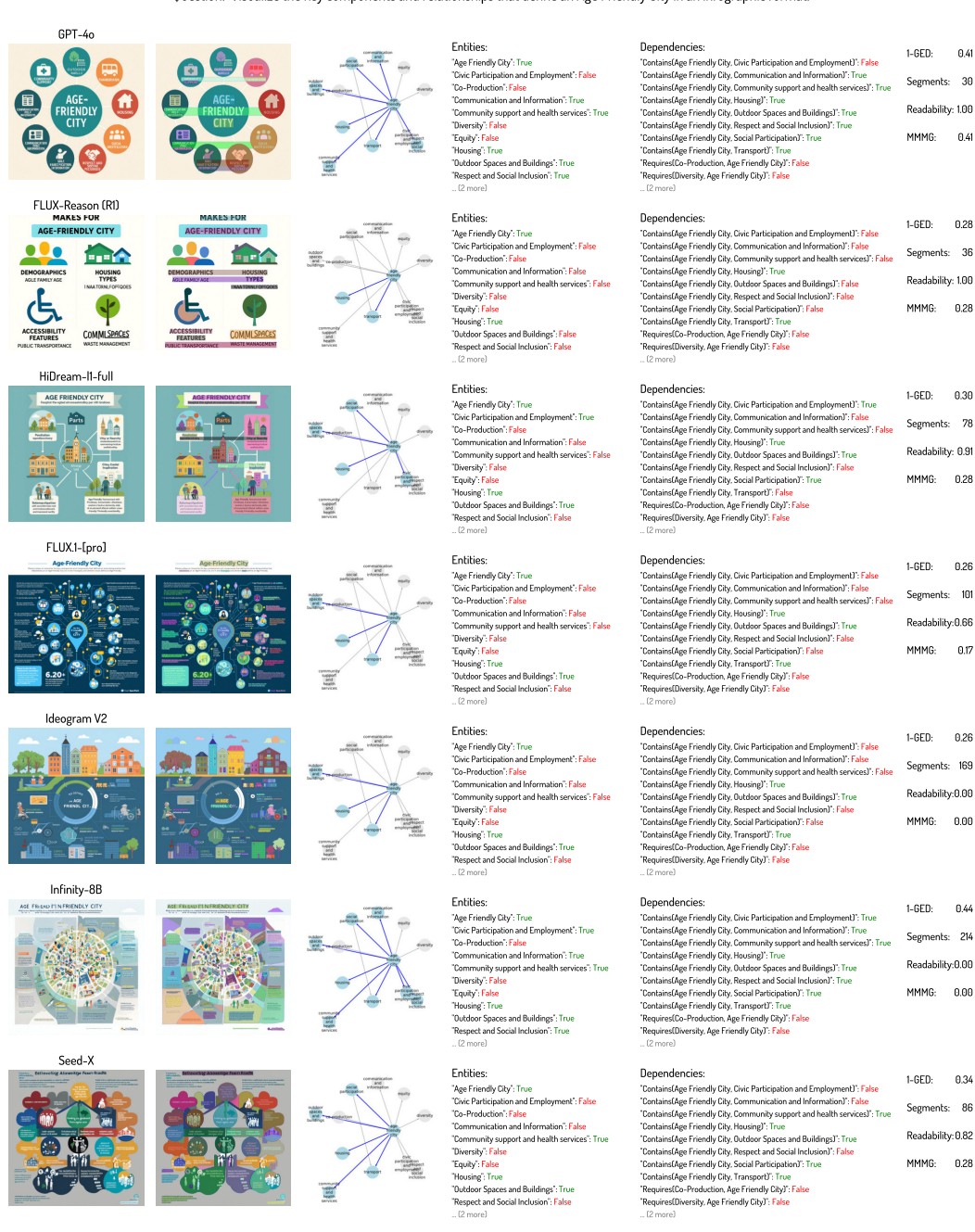

Figure 58: `MMMG` Benchmark visualization for seven representative models on a PhD-Sociology example. Each row corresponds to one model and, from left to right, displays the generated image, its segmentation map, the reconstructed knowledge graph, the extracted entity and dependency lists, and finally the overall `MMMG-Score` along with its component sub-scores.

## F.6.8 History

Question: Design a comparative chart of ancient Greek and Roman political systems, focusing on governance, citizenship, and legal frameworks.

Figure 59: MMMG Benchmark visualization for seven representative models on a PhD-History example. Each row corresponds to one model and, from left to right, displays the generated image, its segmentation map, the reconstructed knowledge graph, the extracted entity and dependency lists, and finally the overall MMMG-Score along with its component sub-scores.

## F.6.9 Philosophy

Question: Visualize a mind map that explores the relationship between philosophy and science, highlighting key debates and figures in the philosophy of science.

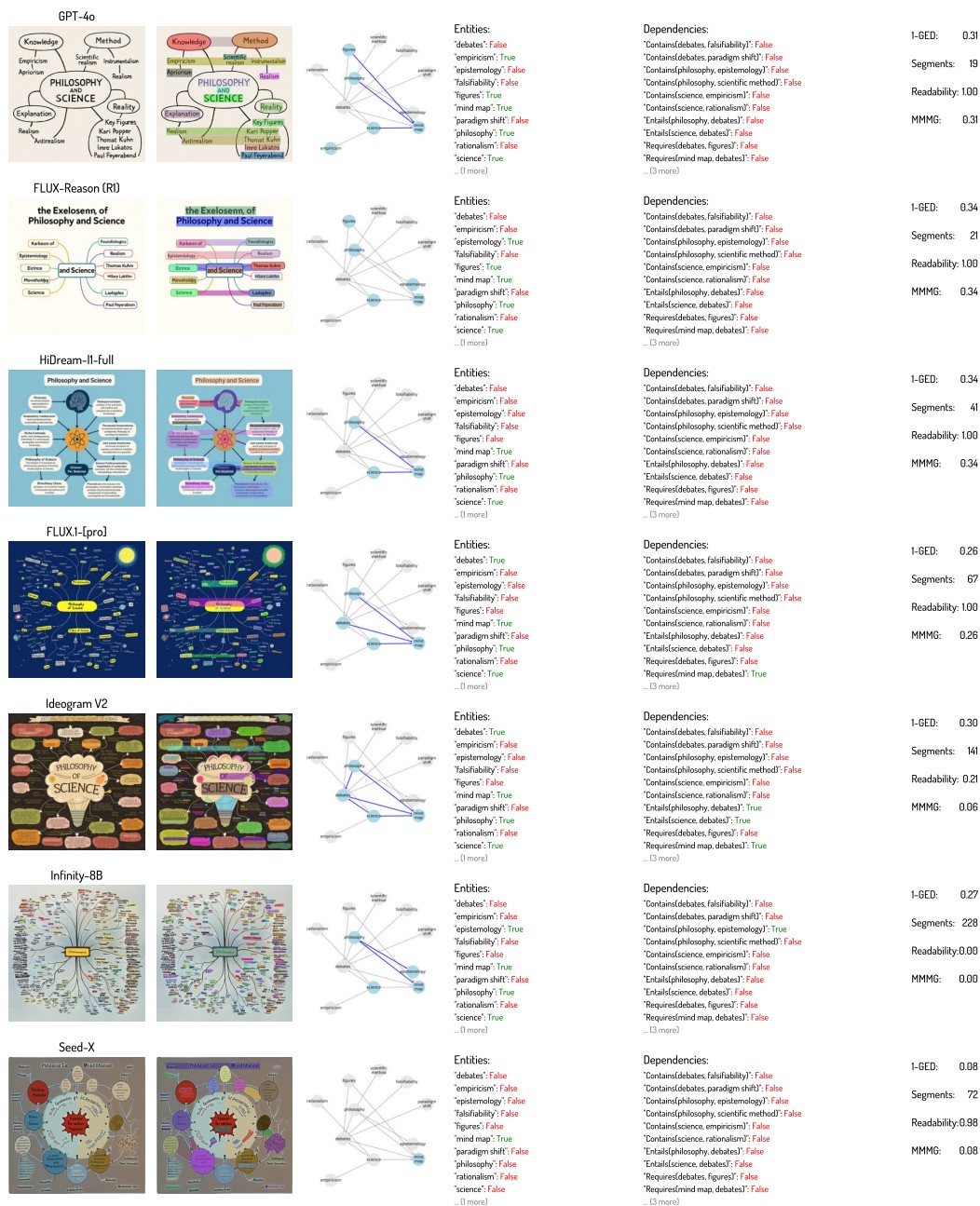

Figure 60: `MMMG` Benchmark visualization for seven representative models on a PhD-Philosophy example. Each row corresponds to one model and, from left to right, displays the generated image, its segmentation map, the reconstructed knowledge graph, the extracted entity and dependency lists, and finally the overall `MMMG-Score` along with its component sub-scores.

### F.6.10 Literature

Question: Create an infographic detailing the use of symbolism in modern literature, featuring examples from key authors.

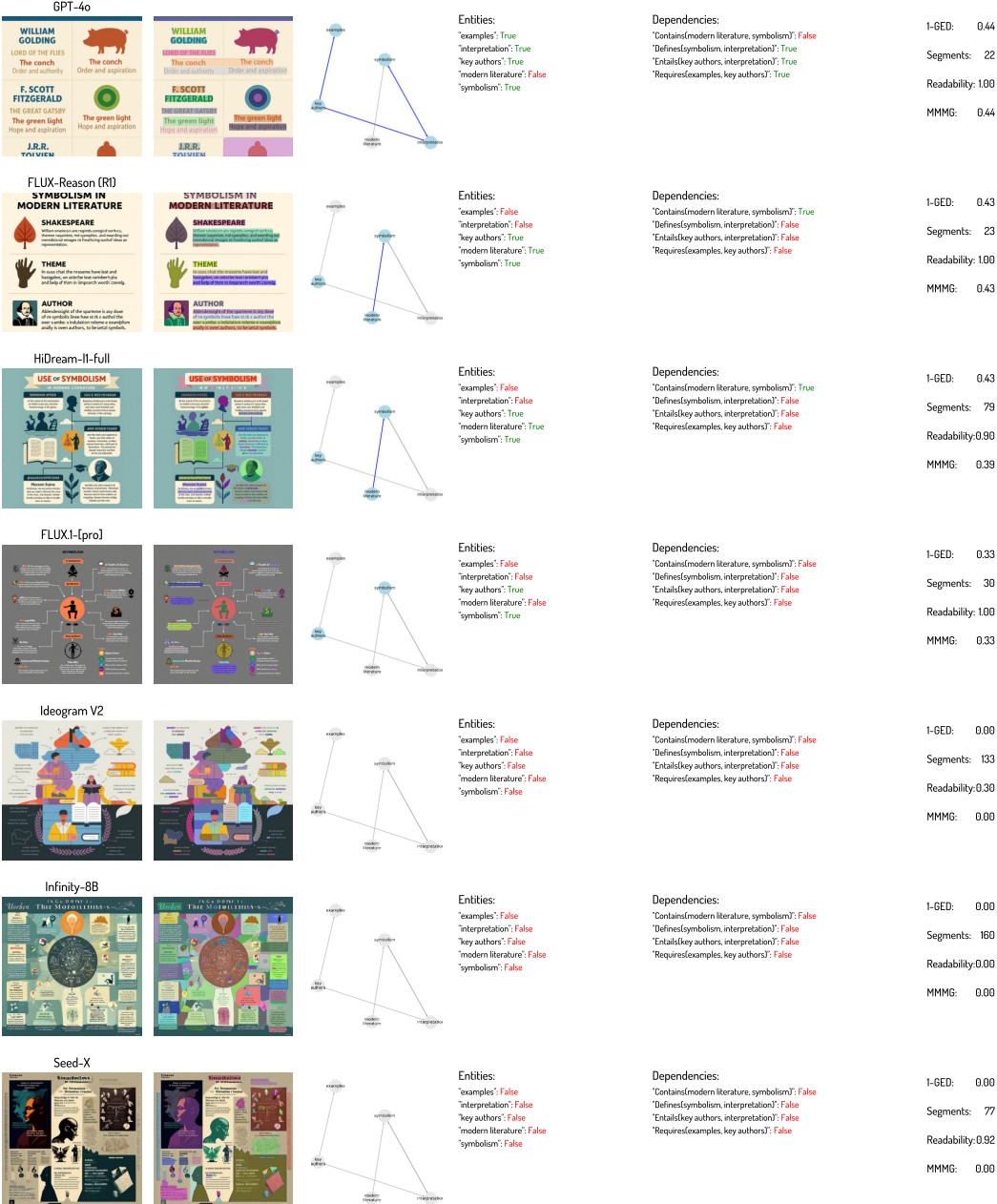

Figure 61: `MMMG` Benchmark visualization for seven representative models on a PhD-Literature example. Each row corresponds to one model and, from left to right, displays the generated image, its segmentation map, the reconstructed knowledge graph, the extracted entity and dependency lists, and finally the overall `MMMG-Score` along with its component sub-scores.

## References