# OpenReview forum: "MMMG: A Massive, Multidisciplinary, Multi-Tier Generation Benchmark for Text-to-Image Reasoning"
_NeurIPS.cc/2025/Datasets_and_Benchmarks_Track — NeurIPS 2025 Datasets and Benchmarks Track poster_

### Official Review · Reviewer_B2Gs · 2025-06-19

**Rating:** 5
**Confidence:** 3

**Summary:**

This paper introduces knowledge image generation as a new task that requires generating visually clear, explanatory images grounded in world knowledge. To evaluate this task, the authors propose MMMG, which contains expert-validated image–prompt pairs spanning diverse disciplines, educational levels, and formats. The paper employs a unified Knowledge Graph representation to define core entities and their relationships in each image, and introduces MMMG-Score, a metric that measures both visual fidelity and clarity. To enhance visual reasoning, the authors release FLUX-Reason, an open baseline combining a reasoning LLM with a diffusion model.

**Additional Feedback:**

Please refer to the weakness section

**Dataset Code Accessibility:**

Yes

**Dataset Code Comments:**

They provide clear documentation for new assets.

**Ethical Considerations:**

No, there are no or only very minor ethics concerns

**Final Justification:**

During the rebuttal, the authors actively addressed my concerns regarding the generalizability of the proposed method, as well as the overstatement of the "hyperparameter-free" claim. While the paper still has some limitations, I believe this benchmark is meaningful and will be valuable for future research. Therefore, I have decided to raise my score.

**Limitations Weaknesses:**

- In the left plots of the third row in Figure 3, there appear to be no data points corresponding to the 25–29 through 50–54 entity and relationship bins. It would be helpful if the authors clarify whether data for these bins exist. If not, excluding these bins from the distribution plots would prevent potential misinterpretation by readers.
- In Section 3.2, it would be beneficial to provide more details on how the knowledge text prompts were divided into the two groups (i.e., image generation and web crawling).
- The Knowledge Fidelity Score and Visual Readability Score evaluate fundamentally different aspects of the generated outputs. Therefore, the rationale for combining them multiplicatively into a single MMMG score is unclear. Given that the benchmark’s primary focus is on visual reasoning, it might be more appropriate to weight the Knowledge Fidelity Score more heavily. The current multiplicative formulation can yield a lower overall MMMG score for outputs with high fidelity but lower readability compared to those with the opposite characteristics, which may not align with the benchmark’s intended objectives. I believe a more detailed justification for this combination and for using a single aggregated score rather than reporting the two metrics separately would be better.
- The FLUX-Reason variants reported in Table 1 utilize the Flux-dev generative model and show improvement over using Flux-dev alone. However, since the proposed reasoning pipeline appears to be generative-model agnostic, it would strengthen the paper to demonstrate its effectiveness combined with multiple generative models (e.g., SEED-X, SDXL, Flux-pro). This would better showcase the generality and robustness of the reasoning approach.
- The error analysis in Section 4.3 provides a useful characterization of common failure modes. Nevertheless, an ablation study or detailed analysis of the FLUX-Reason pipeline’s contribution in mitigating specific failure cases would be valuable. Such an analysis could highlight which failure types are effectively addressed by the proposed reasoning pipeline and which remain challenging, thereby offering clearer insights and future directions for improving reasoning in knowledge image generation.

**Strengths Contributions:**

- The paper is well-written and easy to follow. In particular, the use of illustrative examples and pipeline overviews through figures effectively enhances clarity and improves readability.
- Great motivation. Evaluating the reasoning ability of image generation models is a challenging and realistic problem. This paper takes a promising approach by introducing a novel benchmark (MMMG) to assess such capabilities.
- The paper presents detailed statistics and breakdowns of the MMMG benchmark (Fig. 3), helping readers understand its scale and structure.
- The proposed FLUX-Reason model is compared against a wide range of generative baselines, and it consistently achieves higher MMMG scores (Table 1), demonstrating its effectiveness in knowledge-grounded image generation.

---

> ### Author Rebuttal · Authors · 2025-07-30
>
> ### **Response to Reviewer B2Gs:**
> We thank the reviewer for the careful review and constructive suggestions. We address the raised concerns as follows:
> >**Q1 In the left plots of the third row in Figure 3, there appear to be no data points corresponding to the 25–29 through 50–54 entity and relationship bins. It would be helpful if the authors clarify whether data for these bins exist.**
>
> **Response:**
>
> The gaps in Figure 3 are due to the long-tail distribution of entity and relation counts in our dataset; only a few samples fall within those higher-count bins. Data for these bins does exist, as shown in the detailed counts in **Tables S8 and S9**. We will revise Figure 3 in future versions to include zoomed-in views of these regions, preventing misinterpretation and providing a clearer representation of the data distribution.
>
> **Table S8: Entity count distribution.**
> |Range|0–4|5–9|10–14|15–19|20–24|25–29|30–34|35–39|40–44|
> |-|:-:|:-:|:-:|:-:|:-:|:-:|:-:|:-:|:-:|
> |Count|672|2906|628|162|72|6|2|3|2|
>
> **Table S9: Relation count distribution.**
> |Range|0–4|5–9|10–14|15–19|20–24|25–29|30–34|35–39|40–44|
> |-|:-:|:-:|:-:|:-:|:-:|:-:|:-:|:-:|:-:|
> |Count|1293|2312|650|152|35|4|2|2|1|
>
> >**Q2 In Section 3.2, it would be beneficial to provide more details on how the knowledge text prompts were divided into the two groups (i.e., image generation and web crawling).**
>
> **Response:**
> We divided the knowledge text prompts into two groups, a process guided by semantic clustering to support distinct objectives:
> - **Image Generation Prompts**: Selected from larger semantic clusters, these prompts represent more common concepts. This ensures the prompts align well within GPT-4o's knowledge scope, reducing the likelihood of generating out-of-distribution visuals.
> - **Web Crawling Prompts**: Chosen from smaller, more unique semantic clusters (each containing fewer than three prompts), these prompts emphasize diversity and specificity, making them particularly suitable for web-based data retrieval.
>
> Technically, we embedded 56,830 text prompts using SentenceTransformer and applied DBSCAN for clustering. With `eps=7.5` and `min_samples=1`, we obtained 11,732 clusters.
>
> >**Q3 The Knowledge Fidelity Score and Visual Readability Score evaluate fundamentally different aspects of the generated outputs. Therefore, the rationale for combining them multiplicatively into a single MMMG score is unclear. I believe a more detailed justification for this combination and for using a single aggregated score rather than reporting the two metrics separately would be better.**
>
> **Response:**
>
> We appreciate your observation that Knowledge Fidelity and Visual Readability measure distinct dimensions. Our rationale for combining them into a single score is that **both aspects are indispensable for a truly effective knowledge image**. If either fidelity or readability is lacking, the image fails in its purpose. The multiplicative combination reflects this “AND” relationship, ensuring that the overall score is low if either component is low.
>
> We initially adopted the multiplicative MMMG-Score to avoid introducing extra hyperparameters. In response to your comment, we tested two alternatives:
> - Arithmetic Mean: $\alpha \cdot K_{\text{score}} + (1 - \alpha) \cdot R_{\text{score}}$
> - Geometric Mean: $K_{\text{score}}^\alpha \cdot R_{\text{score}}^{(1 - \alpha)}$,
>
> Results with varying $\alpha$ values (0.5, 0.75, 1.0) are presented in **Tables S10 and S11**. We have discovered that:
> - **Both alternatives are highly sensitive to $\alpha$ choices**, especially the arithmetic mean. An inappropriate choice (e.g., $\alpha=0.5$) reduces the benchmark's differentiation, contradicting the goal of evaluating knowledge-grounded images.
> - Although the geometric mean is somewhat less sensitive, the **rationale for selecting any specific hyperparameter remains unclear** and lacks interpretability.
>
> **Table S10: Average MMMG-Score (arithmetic mean) across different aggregation settings.**
> |$\alpha$|0.5|0.75|1|
> |-|:-:|:-:|:-:|
> |GPT-4o|68.23|55.94|46.94|
> |FLUX-Reason (R1-7B)|63.27|46.03|31.56|
> |FLUX-Pro|55.43|40.05|27.17|
> |SDXL|49.51|37.99|26.47|
> |CogView4|42.18|29.52|18.5|
>
> **Table S11: Average MMMG-Score (geometric mean) across different aggregation settings.**
> |$\alpha$|0.5|0.75|1|
> |:-:|:-:|:-:|:-:|
> |GPT-4o|59.78|50.71|46.94|
> |FLUX-Reason (R1-7B)|45.74|36.01|31.56|
> |FLUX-Pro|39.26|29.45|27.17|
> |SDXL|30.92|26.84|26.47|
> |CogView4|22.97|18.64|18.5|
>
> >**Q4 However, since the proposed reasoning pipeline appears to be generative-model agnostic, it would strengthen the paper to demonstrate its effectiveness combined with multiple generative models (e.g., SEED-X, SDXL, Flux-pro).**
>
> **Response:**
>
> We agree that demonstrating our reasoning-guided approach across multiple backbones would strengthen the contribution. Our focus on FLUX-dev is because it outperforms SEED-X and SDXL, making it a stronger foundation. While FLUX-Pro achieves higher scores, it is a closed-source proprietary model. We would like to extend FLUX-Reason to support other stronger open-source models as future work.
>
> >**Q5 The error analysis in Section 4.3 provides a useful characterization of common failure modes. Nevertheless, an ablation study or detailed analysis of the FLUX-Reason pipeline’s contribution in mitigating specific failure cases would be valuable. Such an analysis could highlight which failure types are effectively addressed by the proposed reasoning pipeline and which remain challenging, thereby offering clearer insights and future directions for improving reasoning in knowledge image generation.**
>
> **Response:**
>
> To evaluate the effectiveness of FLUX-Reason in addressing specific failure cases, we conducted a detailed ablation study comparing three models: FLUX-dev, FLUX-dev augmented with CoT-style reasoning prompts, and FLUX-Reason (R1-Full). We systematically measured readability, entity consistency, and relation depiction accuracy, aligning with the criteria established in Section 4.3.
>
> As illustrated in **Table S12**, CoT-style reasoning prompts significantly improve entity accuracy and moderately improve relation accuracy, demonstrating that explicit textual reasoning cues help guide entity inclusion and relational coherence to some extent. However, readability notably decreased due to overly crowded visuals. We will release the results of "FLUX-dev+Reasoning CoT" on Hugging Face if necessary.
>
> In contrast, FLUX-Reason (R1-Full) substantially mitigates the readability issue and further improves entity consistency, indicating that the training stage effectively enables better spatial planning. Specifically, FLUX-Reason learns to organize and place visual elements while avoids overcrowding, maintaining semantic alignment and clarity.
>
> Nonetheless, the modest gain in relation accuracy indicates that capturing complex relational semantics remains challenging. This inherently demands higher-order multimodal reasoning, continuing to pose significant difficulties. Future research could potentially integrate more sophisticated multimodal reasoning mechanisms or leverage advanced semantic parsing techniques to address and alleviate these persistent issues.
>
> **Table S12: Ablative analysis across FLUX-Reason components.**
> |Model|Readable(%)|Entity(%)|Relation(%)|
> |-|:-:|:-:|:-:|
> |FLUX-Reason(R1-Full)|99.35|58.17|22.35|
> |FLUX-dev+Reasoning CoT|76.01|49.89|19.97|
> |FLUX-dev|91.37|41.82|16.21|

---

> > ### Comment · Reviewer_B2Gs · 2025-08-02
> > **Response to Rebuttal**
> >
> > > **Response to Q1**
> >
> > $\to$ Thank you for the clarification. In such cases, it may be better to use an axis break to truncate the middle part and emphasize that the long tail occupies only a small portion of the overall distribution.
> >
> > > **Response to Q2**
> >
> > $\to$ I respectfully disagree with the authors' explanation. It is not very convincing that the choice between web-scraped and image-generated data should be determined solely by cluster size. Wouldn’t it be more reasonable to try both approaches for the same prompt and ensemble the results? Moreover, it's widely known that web-scraped data already contains a large portion of image-generated content [1, 2]. So, dividing the two methods instead of applying them jointly to the same prompts seems difficult to justify.
> >
> > > **Response to Q3**
> >
> > $\to$ Thank you for sharing the alternative results. However, I believe your current MMMG-Score calculation ($K_\text{score} \cdot R_\text{score}$) might also suffer from a similar issue. Specifically, your equation can be interpreted as a special case where $\alpha=1$ from $K_\text{score} \cdot \frac{R_\text{score}}{\alpha}$, indicating that your equation is not hyperparameter-free. Since the two scores are not on the same scale, the result is still dependent on $\alpha$. I expect that, like the alternative methods you mentioned, your score would also be sensitive to the choice of $\alpha$. If possible, could you also report results using $\alpha=0.5$, $\alpha=0.75$,  $\alpha=1.25$, and $\alpha=1.5$   in your equation?
> > Rather than claiming it is hyperparameter-free, I suggest focusing on what Tables S10 and S11 already show: although absolute values vary depending on $\alpha$, the relative ranking across baselines remains consistent (i.e., the trend is preserved). This allows users to adjust $\alpha$ according to whether they want to prioritize Knowledge Fidelity or Visual Readability. You could state that you adopted a moderate value for $\alpha$ (e.g., $\alpha = 1.0$) that balances both.
> >
> > > **Response to Q4**
> >
> > $\to$ Thank you for the clarification. However, contrary to your claim that Flux-dev outperforms SEED-X and SDXL, the results show that SEED-X outperforms Flux-dev in 'Preschool' and 'Primary', and SDXL also outperforms Flux-dev in 'Preschool'. Even if Flux were a superior model overall, applying your proposed method to only one model risks making the conclusion model-specific. To enhance generalizability, it would be better to demonstrate effectiveness across multiple models.
> >
> > > **Response to Q5**
> >
> > $\to$ Thank you for the detailed ablation studies. However, for the same reasons mentioned in Q4, it is unclear whether the trends observed in Table S12 (based solely on Flux-dev) would hold for other models as well. Without such validation, it's difficult to conclude whether the effect is generalizable.
> >
> > [1] Shumailov et al., AI models collapse when trained on recursively generated data, Nature 2024
> >
> > [2] Hong et al., A Common Pool of Privacy Problems: Legal and Technical Lessons from a Large-Scale Web-Scraped Machine Learning Dataset, arXIv 2025.06

---

> > ### Author Response · Authors · 2025-08-03
> >
> > Thank you for your prompt and insightful reply. We clarify the further raised concerns as follows:
> >
> > >  Q1 In such cases, it may be better to use an axis break to truncate the middle part and emphasize that the long tail occupies only a small portion of the overall distribution.
> >
> > A: Good suggestion! We will follow it to revise the Figure 3 accordingly in the final revision.
> >
> > >  Q2 I respectfully disagree with the authors' explanation. It is not very convincing that the choice between web-scraped and image-generated data should be determined solely by cluster size. Wouldn’t it be more reasonable to try both approaches for the same prompt and ensemble the results? Moreover, it's widely known that web-scraped data already contains a large portion of image-generated content [1, 2]. So, dividing the two methods instead of applying them jointly to the same prompts seems difficult to justify.
> >
> > A: First, we clarify that the key reasons behind the current design are threefold:
> >
> > (i) High-quality, visually appealing knowledge images are scarce online and difficult to collect at scale. Using large-cluster prompts often leads to heavy duplication, as similar knowledge keywords frequently retrieve the same images.
> >
> > (ii) We observe that GPT-4o performs relatively worse on rare knowledge prompts that are distributed mainly within smaller semantic clusters; to ensure higher success rates, we choose semantic prompts from larger semantic clusters.
> >
> > (iii) We need to ensure minimal overlap between the image generation prompts and the web crawling prompts, as we construct the evaluation benchmark MMMG consisting of 4,456 samples, mainly from web crawling prompts, because their original images have higher knowledge fidelity.
> >
> > Second, we clarify that there is no image-generated content within the knowledge images from our web-scraped data. Such high-quality knowledge images are scarce due to inherent significant challenges. We ensure to release the URLs to all crawled knowledge images in the final revision.
> >
> > Third, the reasons we do not apply them jointly are threefold: (i) the challenge of crawling high-quality (visual appealing and standard-resolution) knowledge images from the internet for training, (ii) GPT-4o's poor performance when required to generate rare knowledge concepts, and (iii) the difficulty of ensembling generated images with retrieved images for the same prompts due to their significant gap in dataset distribution. Therefore, we argue that ensembling both image generation approaches and web-crawling remains a challenging task for future exploration.
> >
> > We also welcome any further valuable suggestions and provide corresponding explanations.
> >
> > >  Q3 Further concerns on the current MMMG-Score calculation might also suffer from a similar issue ...
> >
> > A: Above all, we respectfully clarify that the suggested new interpretation cannot support the claim that our approach is not hyperparameter-free. We can reformulate the suggested equation as follows:
> >
> > $K_{\text{score}} \cdot (\frac{R_{\text{score}}}{\alpha}) =(\frac{1}{\alpha})\cdot(K_{\text{score}} \cdot R_{\text{score}})$.
> >
> > Therefore, we can see that the additional $\alpha$ as a divisor for $R_{\text{score}}$ is equivalent to scaling the entire combined score by the constant factor $\frac{1}{\alpha}$. Thus, the hyperparameter $\alpha$ cannot serve as a true weighting mechanism to balance the importance between the knowledge fidelity and readability scores.
> >
> > Second, as we described in the paper, both $K_{\text{score}}$ and $R_{\text{score}}$ are initially in the range [0,1]; thus, the multiplicative MMMG score also lies within [0,1]. However, applying the mentioned scalar factor $\frac{1}{\alpha}$ to the overall score simply rescales all values uniformly, without altering their relative ranking or effectively weighting one component over the other.
> >
> > Third, we argue that such a simple design can avoid arbitrary weighting parameters while preserving the meaningful "AND" logic: if either knowledge fidelity or readability is low, the overall score is penalized accordingly.
> >
> > >  Q4 Applying your proposed method to only one model risks making the conclusion model-specific. To enhance generalizability, it would be better to demonstrate effectiveness across multiple models.
> >
> > A: We will add the experiment results based on SDXL to demonstrate the effectiveness of our approach across multiple models.
> >
> > >  Q5 However, for the same reasons mentioned in Q4, it is unclear whether the trends observed in Table S12 (based solely on Flux-dev) would hold for other models as well. Without such validation, it's difficult to conclude whether the effect is generalizable.
> >
> > A: We will add the experiment results based on SDXL to demonstrate that the effect of each component is generalizable.

---

> > > ### Author Response · Authors · 2025-08-03
> > > **Further Response to Q3: Score Stability under α-Variation**
> > >
> > > To further address Q3, we conduct a stability analysis using a parameterized scoring function:
> > >
> > > $\text{MMMG-Score}(\alpha) = K_{\text{score}} \cdot R_{\text{score}}^{\alpha}$,
> > >
> > > which interpolates between convex and concave regimes. As shown below, model rankings remain largely stable across $\alpha \in [0.5, 1.5]$,  indicating that our metric is robust to reasonable variations in weighting under this design. This supports the reliability of MMMG-Score under mild hyperparameter shifts.
> > >
> > > This experiment supplements our main findings by validating the scoring function’s stability under mild convex/concave perturbations, further supporting its use in multimodal benchmark settings.
> > >
> > > |alpha|0.5|0.75|1|1.25|1.5|
> > > |-|-|-|-|-|-|
> > > |GPT-4o|46.74|46.71|46.66|46.63|46.61|
> > > |FLUX-Reason(R1-7B)|28.50|28.41|28.36|28.25|28.19|
> > > |FLUX-Pro|24.82|24.68|24.66|24.23|24.09|
> > > |SDXL|17.93|16.83|16.58|15.24|14.64|
> > > |CogView4|14.66|14.02|13.99|13.25|12.84|

---

> > ### Comment · Reviewer_B2Gs · 2025-08-03
> > **Official comment by Reviewer B2Gs**
> >
> > Thank you to the authors for their detailed clarifications. First, I appreciate the correction regarding the scoring term. As pointed out, my previous interpretation of the scoring formulation, $Score = K_{\text{score}} \cdot \frac{R_{\text{score}}}{\alpha}$, was incorrect. To properly normalize scores with different magnitudes in a production setting, I agree that the correct formulation should be $Score = K_{\text{score}} \cdot R_{\text{score}}^\alpha$, as the authors suggested. I appreciate the authors' correction of this misunderstanding.
> >
> > However, I still disagree with the claim that the method is entirely "hyperparameter-free," which I find to be an overstatement. For example, combining two loss terms, $L = L_A + L_B$, implicitly assumes equal weighting. In practice, this is equivalent to setting a mixing coefficient $\lambda = 0.5$ in a more general form: $L = (1 - \lambda) L_A + \lambda L_B$. Similarly, the scores that the paper used are also a special case of $\alpha=1$. Therefore, I believe the revision should qualify the claim of being hyperparameter-free.
> >
> > Overall, several concerns have been addressed through the rebuttal, which I appreciate. However, some issues remain, such as regarding generalizability, given the lack of experiments beyond Flux-dev. Taking both sides into account and considering other reviewers' comments, I will maintain my current score. Thank you again for the thoughtful responses.

---

> > > ### Author Response · Authors · 2025-08-04
> > > **Addendum to Reviewer B2Gs**
> > >
> > > Thank you for the thoughtful follow-up. We acknowledge your point on “hyperparameter-free” and will revise the wording accordingly.
> > >
> > > The SDXL experiments are still underway due to ongoing code modifications and compute constraints. We kindly ask for ~2 days to complete them, and hope you may consider the results before finalizing your score.

---

> > ### Comment · Reviewer_B2Gs · 2025-08-04
> > **Response to Authors**
> >
> > I sincerely appreciate your efforts during the rebuttal period.
> >
> > Regarding generalizability, I would like to emphasize that including only SDXL does not sufficiently demonstrate the general applicability of the proposed method. To more convincingly support the plug-and-play nature of your approach, I believe it should also show results on other open-source models used in Table 2, such as SEED-X and CogView-4.
> >
> > While I understand that not everything can be completed within the rebuttal period, applying the method to only a limited number of baselines weakens the generalizability claim.
> >
> > Thank you again for your efforts!

---

> > > ### Author Response · Authors · 2025-08-07
> > > **Response to Reviewer B2Gs**
> > >
> > > Thank you very much for the suggestion. We agree that evaluating the results of diverse reasoning-guided baselines would further demonstrate the generality of our design. However, integrating each new baseline requires three substantial modifications:
> > >
> > > - Adapting to different tokenizers and extending their input length
> > > - Fine-tuning the image generation backbone
> > > - Running on large-scale inference and scoring.
> > >
> > > Given our rebuttal deadline, we selected SDXL as the second experimental baseline. We reported the ablation study of SDXL-Reason, whose performance mirrors the trends observed with FLUX-dev.
> > >
> > > |Model|Readable(%)|Entity(%)|Relation(%)|
> > > |-|:-:|:-:|:-:|
> > > |SDXL-Reason|86.24|42.51|18.42|
> > > |SDXL+CoT|59.53|39.76|17.39|
> > > |SDXL (base)|76.68|37.12|16.40|
> > >
> > > For multimodal models like SEED-X, we plan to include experiments with a reasoning-guided version in future work. In the future, we will also explore an end-to-end design that enables multimodal large models (MLLMs) to jointly generate both native reasoning processes and image outputs. This approach may foster a tighter integration of knowledge and visual representation, ultimately delivering greater value to the community.
> > >
> > > Once again, we sincerely thank you for your valuable suggestion!

---

> > > > ### Comment · Reviewer_B2Gs · 2025-08-07
> > > > **Response to Authors**
> > > >
> > > > I sincerely appreciate the authors' efforts to improve the quality of the paper. During the rebuttal, the authors actively addressed my concerns regarding the generalizability of the proposed method, as well as the overstatement of the "hyperparameter-free" claim. While the paper still has some limitations, I believe this benchmark is meaningful and will be valuable for future research. Therefore, I have decided to raise my score. I thank the authors once again — good work!

---

### Official Review · Reviewer_sFpJ · 2025-07-02

**Rating:** 5
**Confidence:** 4

**Summary:**

This paper introduces knowledge image generation, a novel task focused on generating explanatory visuals (e.g., diagrams, charts) from textual prompts. To support this task, the authors propose the MMMG Benchmark—a dataset comprising 4,000 expert-validated prompt-image pairs spanning 10 disciplines (e.g., Biology, Economics) and six educational levels (from preschool to PhD). Each pair is accompanied by a structured knowledge graph (KG) that defines core entities and relationships. The authors also introduce MMMG-Score, an evaluation metric that combines factual fidelity with visual clarity. Additionally, they present a baseline model, FLUX-Reason, which integrates a reasoning language model with diffusion-based image generation. This model is trained on 16,000 knowledge-image pairs.

**Dataset Code Accessibility:**

Yes

**Ethical Considerations:**

No, there are no or only very minor ethics concerns

**Final Justification:**

The authors have addressed my main concerns regarding the evaluation, as shown in Table S7. They have also promised to test on more models, such as Qwen. I request that the authors include these new results in the revision.

**Limitations Weaknesses:**

1. The proposed FLUX-Reason framework has several limitations. For instance, it lags behind GPT-4o by approximately 16 points—a significant performance gap. Furthermore, the use of GPT-4o outputs during training raises concerns about the potential inheritance of biases and errors from closed-source models.

2. The entire knowledge graph extraction and evaluation pipeline depends on OpenAI’s o3 API, which is proprietary and non-reproducible. This introduces potential risks: any updates to the API may result in performance drift, and existing biases in o3’s knowledge grounding could propagate into the MMMG-Score.

**Strengths Contributions:**

1. Novel Task Definition: The paper addresses a critical gap in text-to-image research by emphasizing explanatory reasoning over aesthetics or compositional accuracy.

2. Benchmark Design: The benchmark provides broad coverage across disciplines and educational levels. The use of structured knowledge graphs enables objective and interpretable evaluation of factual grounding.

---

> ### Author Rebuttal · Authors · 2025-07-30
>
> ### **Response to Reviewer sFpJ:**
> We thank the reviewer for the careful review and constructive suggestions. Below, we offer clarifications and respond to the concerns raised:
>
> > **Q1-1 The proposed FLUX-Reason framework has several limitations. For instance, it lags behind GPT-4o by approximately 16 points—a significant performance gap.**
>
> **Response:**
>
> We explain the main reasons behind FLUX-Reason’s ~16-point performance gap compared to GPT-4o, highlighting the difficulty of the MMMG task. This difference stems from several key aspects:
> - The underlying FLUX-dev model shows a significant performance gap compared to the proprietary models. Notably, FLUX-Reason improves upon its backbone by 12 points.
> - While our current 15k training data is practical, GPT-4o leverages substantially larger data volumes; exploring data scaling is a key future direction for bridging this gap.
> - It is probable that GPT-4o's novel architectural designs and RL-based post-training contribute to its significant performance gains. We plan to investigate these fundamental advancements further in the future.
> FLUX-Reason serves as an initial open-source baseline, demonstrating how reasoning supervision improves factual alignment. We hope it inspires more capable open-source alternatives.
>
> >**Q1-2 Furthermore, the use of GPT-4o outputs during training raises concerns about the potential inheritance of biases and errors from closed-source models.**
>
> **Response:**
>
> We selected GPT-4o due to its strength in producing knowledge-rich, detailed images. However, during experiments, we observed certain GPT-4o-specific behaviors, such as generating cropped images. To minimize potential biases and reduce errors, we conducted thorough inspections and filtered the data sources. This is an intermediate step towards achieving robust data quality. We plan to diversify our data sources by incorporating models like Gemini 2.0, HiDream, and other competitive models.
>
> >**Q2 The entire knowledge graph extraction and evaluation pipeline depends on OpenAI’s o3 API, which is proprietary and non-reproducible. This introduces potential risks: any updates to the API may result in performance drift, and existing biases in o3’s knowledge grounding could propagate into the MMMG-Score.**
>
> **Response:**
>
> We used OpenAI’s o3 model following common practice for leveraging state-of-the-art models in evaluation. To address concerns about API-induced performance drift, we tested MMMG-Score using multiple API variants (o1, o3, and o3-high). As shown in **Table S7**, despite minor shifts in absolute scores, relative model rankings remain consistent. Moving forward, we will further enhance reproducibility and robustness by evaluating additional proprietary (e.g., GPT-5) and open-source models (e.g., Qwen), and updating our results accordingly.
>
> **Table S7: Robustness of MMMG-Score across OpenAI evaluator versions.**
> |Evaluator|GPT-4o|FLUX-Reason (R1-7B)|FLUX.1-[pro]|Infinity-8B|SEED-X|
> |-|:-:|:-:|:-:|:-:|:-:|
> |OpenAI-o3-high|45.87|28.36|23.40|16.99|16.11|
> |OpenAI-o3|46.66|30.52|24.66|18.16|17.38|
> |OpenAI-o1|47.21|36.65|29.52|24.37|21.26|

---

> > ### Comment · Reviewer_sFpJ · 2025-08-09
> >
> > Thank you for the extensive response. My main concerns, especially regarding the evaluation, have been addressed. I would like to maintain my positive rating and encourage the authors to include those details in the revision.

---

### Official Review · Reviewer_vto3 · 2025-07-02

**Rating:** 5
**Confidence:** 5

**Summary:**

The paper introduces knowledge image generation as a new task to probe the reasoning capability of image generation models. The released benchmark spans over 10 disciplines, 6 educational levels, and 10 diverse knowledge formats such as charts, diagrams, and mind maps.  They propose a novel metric based on KG representations to evaluate knowledge images using factual fidelity and graph edit distance. They further release FLUX-Reason trained on 16000 knowledge image-prompt pairs. Overall this is a solid paper that proposes a new task and backs it with metrics and improvement.

**Additional Feedback:**

The references misses several Text-to-Image benchmarks. Some additional details have to be added to the paper for the benefit of the readers.

**Dataset Code Accessibility:**

Yes

**Dataset Code Comments:**

Dataset and Code released

**Ethical Comments:**

No concern

**Ethical Considerations:**

No, there are no or only very minor ethics concerns

**Limitations Weaknesses:**

1) The Knowledge Graph Extraction details are not clear in the paper
2)  Not enough reasoning for why GPT-4o image generation (30K samples) and for web crawling (26K samples) were used as sources for image collection. Why not just one?
3) Details about the architectural changes in Flux-Reason is missing. Such as how is the T5 encoder's context length changed
4) Lack of Gemini 2.5 in comparison as it is widely regarded as the strongest multimodal model with native image generation capabilities.
5) Several text-to-image benchmarks are missing from Table 1. See missing references below.

References missing:

* Beyond Aesthetics: Cultural Competence in Text-to-Image Models: https://arxiv.org/pdf/2407.06863  (tests for world knowledge and introduces a new diversity T2I metric)
* GenAI-Bench: Evaluating and Improving Compositional Text-to-Visual Generation: https://arxiv.org/abs/2406.13743 (compositional reasoning)

**Strengths Contributions:**

Overall this is a very sound paper with thorough experimentation, a nice new idea, and a timely benchmark. The paper also proposes a new finetuning approach to improve and significantly beat the base model on their benchmark.

1) The paper deals with a timely problem of knowledge image generation in the form of charts, infographs and other forms of information-rich photographs
2) The paper introduces 2 useful metrics and combines them to propose a novel score
3)  Thorough experimentation with SOTA text-image models on the proposed benchmark
4) Correlation of MMMG-Score with human scores beats existing metrics.
5) Flux-reason is an interesting approach to use text CoT to improve reasoning in image generation.

---

> ### Author Rebuttal · Authors · 2025-07-30
>
> ### **Response to Reviewer vto3**
>
> We thank the reviewer for the careful review and constructive suggestions. We address the raised concerns as follows:
>
> > **Q1 The Knowledge Graph Extraction details are not clear in the paper.**
>
> **Response:**
>
> Knowledge graphs (KGs) are generated in a two-step process: first, an automated extraction using OpenAI-o3, and then meticulous expert revision. Technically, OpenAI-o3 processes each <prompt, image> pair to produce a structured KG following a defined schema (Supplemental C.3, Page 6). Subsequently, four expert annotators manually verify and refine these generated KGs, cross-checking against the original image and prompt. This hybrid approach ensures high-quality, verifiable KGs, balancing scalability with fidelity. The final annotation format is as follows:
> ```json
> { "Visual Components":
> { "Entities": ["entity_1", "entity_2", "..."],
> "Relations": ["relation_1(e_i, e_j)", "relation_2(e_k, e_l,)", "..."]
> },
> "Key Knowledge":
> { "Definitions": "Elaborating the key knowledge concept, including the above entities and relations.",
>  "Entity Explanation": ["Entity 1 explanation", "Entity 2 explanation", "..."],
> "Relation Explanation": ["Relation 1 explanation", "Relation 2 explanation","..."],
> },
> }
> ```
>
> > **Q2 Not enough reasoning for why GPT-4o image generation (30K samples) and for web crawling (26K samples) were used as sources for image collection. Why not just one?**
>
> **Response:**
>
> Our data collection combined GPT-4o generation and web crawling to leverage the unique strengths of both sources for our distinct purposes:
> - **GPT-4o images** (initially 30k raw samples) provide controllable generation and enhanced visual aesthetics crucial for scalable training (~16k filtered samples), though this source necessitated human inspection for knowledge accuracy.
> - **Web-crawled images** (initially 26k raw samples) offer diverse, accurate, knowledge-rich real-world content, making them ideal for benchmark evaluation (~4k filtered samples); however, their variable resolution and low aesthetics made them unsuitable for training.
>
> This dual approach effectively balances authentic evaluation with scalable training data.
>
> > **Q3 Details about the architectural changes in Flux-Reason is missing. Such as how is the T5 encoder's context length changed.**
>
> **Response:**
>
> We appreciate your interest in FLUX-Reason. Our integration of Chain-of-Thought (CoT) reasoning follows a minimalist design. To accommodate longer CoT traces, we extended the T5 encoder’s default context length from 256 to 1024 tokens by setting `max_length=2048` in the tokenizer (see Section 3.4, lines 214–217). This straightforward modification makes FLUX-Reason an easily understandable baseline for leveraging high-level reasoning.
>
> > **Q4 Lack of Gemini 2.5 in comparison, as it is widely regarded as the strongest multimodal model with native image generation capabilities.**
>
> **Response:**
>
> Thanks for your advice. Since Gemini 2.5 does not support native image generation [1], we instead benchmarked Gemini 2.0, a multimodal generation model, with results shown in **Table S6**.
>
> **TABLE S6: MMMG-Score (×100) of sGemini-2.0 Model.**
> |Education|Preschool|Primary|Secondary|High|Undergrad|PhD|Avg.|
> |-|:-:|:-:|:-:|:-:|:-:|:-:|:-:|
> |Gemini-2.0|50.19|47.14|46.78|46.29|35.42|34.81|43.44|
>
> [1] Gemini 2.5 official model card: https://cloud.google.com/vertex-ai/generative-ai/docs/models/gemini/2-5-pro
>
> > **Q5 Several text-to-image benchmarks are missing from Table 1. See missing references below.**
>
> **Response:**
>
> We appreciate your valuable suggestion to include these important benchmarks. We have updated **Table 1** to include "Beyond Aesthetics" and "GenAI-Bench".
>
> **Table 1: Comparison with prior Text‑to‑Image (T2I) benchmarks.**
> |Benchmark|Scale|Focus|Domains|Metrics|World Knowledge|Explanatory|
> |--|--|--|--|--|--|--|
> |**GenAI-Bench**|1,600|Compositionality|Scenes, objects, attributes, relations, counting, comparison, etc.|VQAScore|❌|❌|
> |**CUBE**|1,000|Cultural Competence|Cuisine, Landmarks, Art spanning 8 countries |Cultural Awareness, Diversity via Vendi Scores|✅|❌|
> |**MMMG (Ours)**|4,456|Disciplinary Knowledge|10+ Academic Fields|Knowledge Graph Edit Distance + Readability|✅|✅|

---

> > ### Comment · Reviewer_vto3 · 2025-08-08
> > **Thanks to the authors**
> >
> > Thanks for providing a comprehensive response to my queries. I retain my positive evaluation of the paper and request the authors to provide the additional details and add the missing citations in the paper.

---

### Official Review · Reviewer_TDyw · 2025-07-04

**Rating:** 4
**Confidence:** 3

**Summary:**

The authors introduce MMMG, a new benchmark and dataset for knowledge-driven text-to-image generation (a.k.a., knowledge image generation). Unlike conventional text-to-image tasks that focus on simple prompt fidelity or aesthetic quality, MMMG targets multimodal reasoning that given a concise, academic-style prompt, models must autonomously infer relevant entities and their relationships and produce a coherent explanatory visual. The MMMG benchmark consists of 4,456 expertly curated prompt–image pairs spanning 10 diverse disciplines (e.g., Biology, Economics, Engineering, History, etc.) and 6 educational levels. Each sample is annotated with a ground-truth knowledge graph detailing the key concepts (nodes) and relationships (edges) that the image should convey. This structured annotation enables format-agnostic evaluation of factual content.

**Dataset Code Accessibility:**

Yes

**Ethical Considerations:**

Yes, there are significant ethics concerns that require review by an ethics expert

**Final Justification:**

I have thoroughly reviewed the issues put forward by the other reviewers as well as the rebuttal provided. All of the questions (1. Details of MMMG-Score; 2. Fairness in evaluation; 3. Open-source problem; 4. Ethics ) have been well resolved. Consequently, I maintain my original rating of “borderline accept”.

**Limitations Weaknesses:**

1. The MMMG-Score, while innovative, has an inherent limitation: it depends on the accuracy of the LLM in extracting the knowledge graph from the generated image. If the evaluator (GPT-4o) fails to detect an entity or relation that is actually depicted, the metric will falsely penalize the model’s output. The authors also acknowledge this issue: when the reference graph is complex, “OpenAI-o3 may fail to detect all entities or relations; we default missing items to false”. Furthermore, the hyper-parameters $n_{min}$ and $n_{max}$ in the Readability Score lack adequate empirical evidence for support, and no sensitivity analysis has been provided either.

2. The task defined by MMMG is extremely challenging in part because it requires drawing diagrams with legible text and clear structure, which current generative models notoriously struggle with. Many state-of-the-art image models (especially diffusion models) cannot reliably produce readable text or avoid visual clutter. This means that a model’s MMMG-Score can be heavily impacted by low-level rendering flaws, not just a lack of reasoning or knowledge. For instance, several failure cases highlighted involve “visually cluttered and unreadable text” in the output. In fact, the design of MMMG-Score explicitly zeros out the score if the image is too fragmented. This also raises the question of whether a model with an external text-overlay mechanism (placing text via post-processing) could unfairly gain score.

3. While the authors have released the dataset and the evaluation script, the trained models (baselines) and certain details are not yet publicly accessible.

**Strengths Contributions:**

1. It addresses a clear gap in the literature by focusing on reasoning-oriented image generation, as opposed to prior benchmarks that mainly test style, prompt adherence or simple compositionality.

2. The MMMG dataset is carefully curated and diverse in both content and difficulty. It covers ten different knowledge domains and progressively harder educational levels, which is far more comprehensive than prior efforts like WISE (1,000 images focused on commonsense/cultural knowledge) or GenEval (553 images of object attributes).

3. The proposed evaluation metric MMMG-Score  is a notable contribution on its own. It directly tackles the core challenge of assessing semantic fidelity in generated images.

---

> ### Author Rebuttal · Authors · 2025-07-30
>
> ### **Response to Reviewer TDyw**
>
> We thank the reviewer for the careful review and constructive suggestions. **As each raised concern consists of multiple questions, we decompose each question into multiple sub-questions** and address them as follows:
>
> > **Q1-1 The MMMG-Score, while innovative, has an inherent limitation: it depends on the accuracy of the LLM in extracting the knowledge graph from the generated image. If the evaluator (GPT-4o) fails to detect an entity or relation that is depicted, the metric will falsely penalize the model’s output. The authors also acknowledge this issue: when the reference graph is complex, “OpenAI-o3 may fail to detect all entities or relations; we default missing items to false”.**
>
> **Response:**
>
> Above all, we admit that OpenAI-o3 may fail to extract extremely complex knowledge graphs, but **we need to clarify that OpenAI-o3 generally performs well on the constructed benchmark, that the impact of failure cases is limited**, and that we provide the following key statistics to address this concern:
>
> 👉  To directly assess the accuracy and reliability of the evaluator (OpenAI-o3), we conducted a human vs. model comparison study. We evenly sampled 480 images across all educational levels and had domain experts annotate their corresponding knowledge graphs.
>
> As shown in **Table S1**, the evaluator demonstrates high alignment with human annotations, with an average recall exceeding 94% across models, indicating that missed detections and false penalties are rare.
>
> 👉  Furthermore, **Table S2** shows that MMMG-Scores computed by the LLM-based evaluator closely match those derived from expert annotations, with an average deviation of less than 1.5 points across all disciplines and difficulty levels. This confirms that, despite the theoretical limitations of using LLMs for graph extraction, the scoring metric remains reliable, consistent, and aligned with human judgment in practice.
>
> 👉 We also clarify that highly complex graphs are rare: only 13 of the 4,456 samples contain more than 25 entities or relations (see Fig. 3). In a manual audit of these high-complexity cases across three models (GPT-4o, FLUX-Reason-7B, and SDXL), only two instances exhibited severe entity-level omissions by the evaluator. These rare cases can be excluded or flagged in the future.
>
> We will include the above discussion in the final revision to avoid any confusion.
>
> **Table S1: Average Precision (P) and Recall (R) between OpenAI-o3 and expert evaluators.**
> |Model|P(%)|R(%)|
> |-|:-:|:-:|
> |GPT-4o|96.52|96.48|
> |FLUX-Reason|94.27|94.13|
> |SDXL|92.74|94.70|
>
> **Table S2: Average MMMG-Score comparison between OpenAI-o3 and expert scores (on 480 samples).**
> |Model|Evaluator|Preschool|Primary|Secondary|High|Undergrad|PhD|
> |-|:-:|:-:|:-:|:-:|:-:|:-:|:-:|
> |GPT-4o|o3|62.84|59.73|52.53|50.47|42.61|40.14|
> |GPT-4o|Experts|62.88|58.17|51.01|50.63|42.25|40.28|
> |FLUX-Reason|o3|48.16|42.29|38.10|34.58|28.37|27.06|
> |FLUX-Reason|Experts|48.33|41.91|37.72|34.79|28.14|27.43|
> |SDXL|o3|40.63|33.06|25.82|24.94|16.69|15.35|
> |SDXL|Experts|40.88|32.98|25.27|25.04|17.28|15.01|
>
> > **Q1-2 The hyper-parameters $n_{\text{min}}$ and $n_{\text{max}}$ in the Readability Score lack adequate empirical evidence for support, and no sensitivity analysis has been provided either.**
>
> **Response:**
>
> We empirically selected the hyperparameters $n_{\text{min}}$ and $n_{\text{max}}$​ based on segment statistics (Appendix D.2), where the majority of generated images fall between 65 and 155 segments. Our default values ($n_{\text{min}}=70$ and $n_{\text{max}}=160$) define a readability range that avoids overly sparse or cluttered images.
>
> Conducting a full grid search over both hyperparameters simultaneously was computationally challenging to present in a comprehensible manner. Thus, we examined sensitivity of $n_{\text{min}}$ and $n_{\text{max}}$ independently across a wide range (**Tables S3, S4**):
> - $n_{\text{min}}\in\{40, 50, 60, 70, 80, 90\}$
> - $n_{\text{max}}\in\{130, 140, 150, 160, 170, 180\}$.
>
> We tested models of varying generation quality. The key findings are:
> - Relative model rankings remain stable across all tested hyperparameter values.
> - Absolute changes in MMMG-Score were minimal (~1 point).
>
> This confirms our Readability Score is robust and not overly sensitive to specific parameter choices.
>
> **Table S3: Average MMMG-Score sensitivity to $n_{\text{min}}$.**
> |n_min|40|50|60|70|80|90|
> |-|:-:|:-:|:-:|:-:|:-:|:-:|
> |GPT-4o|45.5|46.13|46.48|46.66|46.77|46.84|
> |FLUX-Reason (R1-7B)|27.24|27.68|28.12|28.36|28.51|28.62|
> |FLUX-Pro|23.52|24.13|24.41|24.66|24.85|25.09|
> |SDXL|14.17|15.24|16.39|16.58|17.13|17.75|
> |CogView4|12.27|12.62|13.21|13.99|14.43|14.93|
> |Janus-pro-7B|8.84|9.01|9.44|9.76|10.07|10.30|
> |LlamaGen|2.57|2.63|2.69|2.72|2.73|2.74|
>
> **Table S4: Average MMMG-Score sensitivity to $n_{\text{max}}$.**
> |n_max|130|140|150|160|170|180|
> |-|:-:|:-:|:-:|:-:|:-:|:-:|
> |GPT-4o|46.53|46.58|46.63|46.66|46.69|46.71|
> |FLUX-Reason (R1-7B)|27.86|28.01|28.14|28.36|28.43|28.49|
> |FLUX-Pro|23.85|24.17|24.43|24.66|24.82|25.03|
> |SDXL|14.22|15.04|16.16|16.58|16.88|17.58|
> |CogView4|12.32|13.17|13.75|13.99|14.16|14.32|
> |Janus-pro-7B|9.71|9.73|9.74|9.76|9.78|9.79|
> |LlamaGen|2.68|2.69|2.71|2.72|2.73|2.73|
>
>
> > **Q2-1 The task defined by MMMG is extremely challenging in part because it requires drawing diagrams with legible text and clear structure, which current generative models notoriously struggle with. Many state-of-the-art image models (especially diffusion models) cannot reliably produce readable text or avoid visual clutter.**
>
> **Response:**
>
> We agree that MMMG essentially involves generating diagrams with legible text and clear structure, which is a key reason why we designed our benchmark to test models’ abilities in both knowledge fidelity and visual readability.
>
> While earlier models exhibited limitations in legibility and layout quality, recent state-of-the-art models—such as GPT-4o, FLUX-Pro/dev, and Ideogram—have shown marked improvements in both text clarity and visual organization. GPT-4o has already achieved 84% precision on dense-text poster rendering, making concerns about text legibility secondary.
>
> > **Q2-2 This means that a model’s MMMG-Score can be heavily impacted by low-level rendering flaws, not just a lack of reasoning or knowledge. For instance, several failure cases highlighted involve “visually cluttered and unreadable text” in the output.**
>
> **Response:**
>
> We would like to clarify that MMMG is not primarily affected by low-level rendering flaws. Our analysis shows that top models like GPT-4o and FLUX generate visually clear, well-organized text and layouts under identical prompts. The issue of "visually cluttered and unreadable text" primarily occurs in baseline models such as LlamaGen and Infinity, which lack strong vision-language alignment and planning capabilities.
>
> This discrepancy suggests that such visual issues are less a result of low-level rendering and more a reflection of high-level failures in reasoning and layout planning. Since MMMG evaluates models’ ability to infer semantic structures and organize entities from abstract prompts, poor visual output typically reflects a model’s difficulty in semantic inference and planning. Such limitations likely stem from differences in model scale, training data, or architecture design.
>
> > **Q2-3 In fact, the design of MMMG-Score explicitly zeros out the score if the image is too fragmented.**
>
> **Response:**
>
> We clarify our rationale as follows: the score is zeroed for overly fragmented or unreadable outputs for two key reasons:
> 1. Such outputs are empirically unusable for assessing any meaningful reasoning.
> 2. Certain models (e.g., Infinity) may exploit this by over-generating dense but illegible outputs to mislead the evaluator.
>
> This scoring rule strongly penalizes poor readability, thereby ensuring generated images maintain sufficient legibility for meaningful knowledge evaluation. We have also included Infinity’s generated outputs in the released dataset for reference.
>
> > **Q2-4 This also raises the question of whether a model with an external text-overlay mechanism (placing text via post-processing) could unfairly gain score.**
>
> **Response:**
>
> The concern is understandable, but how to achieve the text-overlay is non-trivial and even impossible as accurately detecting the regions to put text, determining appropriate text arrangement, and ensuring visually coherent text overlays is itself highly challenging. Accordingly, we performed an initial simulation by first applying PP-OCR for text-region detection, followed by inpainting using the average edge color, and then re-pasting entity texts selected to closely match the original character length. As shown in **Table S5**, this manipulation did not improve performance, clearly indicating that MMMG-Score does not heavily rely on superficial text presence.
>
> **Table S5: Impact of External Text Overlays on MMMG-Score.**
> |Education|Preschool|Primary|Secondary|High|Undergrad|PhD|
> |-|:-:|:-:|:-:|:-:|:-:|:-:|
> |SDXL|25.22|19.37|16.62|16.38|11.72|10.21|
> |SDXL-edit|24.16|19.81|17.79|17.4|11.11|10.45|
> |FLUX-Reason|35.66|32.49|31.15|27.52|22.29|21.06|
> |FLUX-Reason-edit|35.43|32.16|30.38|27.71|22.48|21.66|
>
> > **Q3 While the authors have released the dataset and the evaluation script, the trained models (baselines) and certain details are not yet publicly accessible.**
>
> **Response:**
>
> We have released the MMMG benchmark, training data, and evaluation scripts. FLUX-Reason’s model weights and code are being finalized and will be available soon.
>
> > **Q4 Ethical Considerations: Yes, there are significant ethical concerns that require review by an ethics expert.**
>
> **Response:**
>
> We assume this may have been an unintentional selection. If not, we’re happy to clarify our safeguards: our data curation excludes any potential harmful images, ensures compliance with copyright-safe datasets, and removes sensitive entities to ensure ethical integrity.

---

> > ### Comment · Reviewer_TDyw · 2025-08-09
> >
> > Thank you for your comprehensive response. The issues raised have been effectively resolved, and I have no more questions.

---

### Decision · Program_Chairs · 2025-09-18

**Decision:**

Accept (poster)

**Comment:**

The paper introduces a new task for probing the reasoning capability of image generation models. The work is thorough and timely. The authors carefully engage with the reviewers during discussion, and all reviewers are in favor of acceptance (one reviewer with a score of 4 albeit noting that there are no remaining issues, and the rest 5). The new task would be of broad interest to the community.

===== FINAL UPDATE FROM DB Track PCs ====

The final decision for this paper has been taken by the program chairs after consultation with the SACs. All Senior Area Chairs have ranked papers according to the feedback from the AC during the review process. We decided to leave the original meta-review to reflect the opinion of the AC in light of the initial discussions with reviewers and SAC.